# Partial Optimal Transport for Support Subset Selection

**Bilal Riaz**                                                              *bilalria@udel.edu*
*Department of Electrical & Computer Engineering*
*University of Delaware*

**Yüksel Karahan**                                                          *ykarahan@udel.edu*
*Department of Electrical & Computer Engineering*
*University of Delaware*

**Austin J. Brockmeier**                                                    *ajbrock@udel.edu*
*Department of Electrical & Computer Engineering*
*Department of Computer & Information Sciences*
*University of Delaware*

**Reviewed on OpenReview:** *https://openreview.net/forum?id=75CcopPxIr*

## Abstract

In probabilistic terms, optimal transport aims to find a joint distribution that couples two distributions and minimizes the cost of transforming one distribution to another. Any feasible coupling necessarily maintains the support of both distributions. However, maintaining the entire support is not ideal when only a subset of one of the distributions, namely the source, is assumed to align with the other target distribution. For these cases, which are common in machine learning applications, we study the semi-relaxed partial optimal transport problem that relaxes the constraints on the joint distribution allowing it to under-represent a subset of the source by over-representing other subsets of the source by a constant factor. In the discrete distribution case, such as in the case of two samples from continuous random variables, optimal transport with the relaxed constraints is a linear program. When sufficiently relaxed, the solution has a source marginal with only a subset of its original support. We investigate the scaling path of solutions, specifically the relaxed marginal distribution for the source, across different relaxations and show that it is distinct from the solutions from penalty-based semi-relaxed unbalanced optimal transport problems and fully-relaxed partial optimal transport, which have previously been explored. We demonstrate the usefulness of this support subset selection in applications such as color transfer, partial point cloud alignment, and semisupervised machine learning, where a part of data is curated to have reliable labels and another part is unlabeled or has unreliable labels. Our experiments show that optimal transport under the relaxed constraint can improve the performance of these applications by allowing for more flexible alignment between distributions.

## 1 Introduction

Measuring, and subsequently minimizing, the dissimilarity between two distributions or data samples are ubiquitous tasks in machine learning. Recently, the theory of optimal transport and the family of Wasserstein distances have seen applications across the spectrum of machine learning problems including computer vision (Solomon et al., 2014; Kolouri et al., 2017; Rabin et al., 2011; Garg et al., 2020), generative modeling (Arjovsky et al., 2017; Gulrajani et al., 2017; Salimans et al., 2018; Genevay et al., 2018; Tolstikhin et al., 2018; Kolouri et al., 2018; Deshpande et al., 2018; Rout et al., 2022; Korotin et al., 2023; Mokrov et al., 2021), natural language processing (Xu et al., 2018), and domain adaptation (Kirchmeyer et al., 2022). Optimal transport is widely applicable since it combines the statistical and geometric aspects of data and provides a correspondence to couple two samples or distributions.

However, a limitation of standard optimal transport is the strict constraint on the complete transfer of mass between the two distributions being compared (Frogner et al., 2015). This can be problematic in cases where such a transfer is not necessary or desirable, such as when dealing with distributions with different support, over- or under-representation, or the presence of outliers in a portion of the data. In order to deal with these scenarios partial optimal transport problems (Rubner et al., 1997; Figalli, 2010; Caffarelli & McCann, 2010; Bonneel & Coeurjolly, 2019; Chapel et al., 2020) have been proposed. Partial optimal transport relaxes the marginal constraints on the transport plan to inequalities, allowing transportation plans that cover only a fraction of the total mass. Similarly, unbalanced optimal transport corresponds to the case of unbalanced masses, where only a portion of the mass is transported. One or both of the marginal constraints can also be relaxed by divergence-based regularizations, such as Kullback-Leibler divergence, total-variation distance (equivalent to the $\ell_1$-norm of the difference in mass vectors in the discrete case), or squared $\ell_2$-norm (Benamou, 2003; Chizat et al., 2018; Blondel et al., 2018; Peyré & Cuturi, 2019; Séjourné et al., 2019; Chapel et al., 2021). The solution of partial or unbalanced optimal transport often selects a subset (also known as an active region) of the support. This property is exploited in machine learning (Chapel et al., 2020; 2021; Phatak et al., 2023), including the partial Wasserstein covering problem, which has applications in active learning (Kawano et al., 2022).

In practice, a key limiting factor is the scalability of solving the linear programs corresponding to partial, unbalanced, or standard optimal transport for large data sets. While efficient gradient based methods cannot be applied to linear programs directly, a number of regularization based remedies, including entropic (Cuturi, 2013; Cuturi & Peyré, 2018), quadratic, and group-LASSO regularization (Flamary et al., 2016; Blondel et al., 2018) have been shown to give approximate solutions. In particular, the Sinkhorn algorithm provides a solution to the entropically regularized standard optimal transport problem. The Sinkhorn algorithm has been widely applied due to its simple implementation consisting of alternating projections to the feasible sets of marginal constraints. In the work by Chizat et al. (2018), Dykstra's algorithm[1] is used to solve the entropically regularized partial optimal transport problem. Similarly, Sinkhorn-like iterations are used for entropically regularized unbalanced optimal transport in the work by Frogner et al. (2015), and the work by Séjourné et al. (2019) adapts the Sinkhorn algorithm via asymmetric proximal operators to handle a variety of divergence-based relaxations. Scalability is even more important in cases where the solution at different levels of relaxation are sought. Recent works have proposed algorithms for computing the entire scaling or regularization path of solution for fully-relaxed partial optimal transport (Phatak et al., 2023) and unbalanced optimal transport problems with fully or semi-relaxed marginal constraints (Chapel et al., 2021).

In this paper, we study a discrete case of partial optimal transport (Figalli, 2010; Caffarelli & McCann, 2010), which is posed as a linear program, where the solution is a joint distribution with one fixed marginal and one marginal that is constrained to be pointwise less than or equal to a constant factor $c \geq 1$ of the original, typically uniform, marginal. The generalized form of this constraint where each point has its own capacity factor was proposed in the work by Rabin et al. (2014). Initially, we propose an entropically regularized version to efficiently find an approximate solution, which is equivalent to a specific case within the framework covered in the work of Séjourné et al. (2019), that we solve with an algorithm that combines Sinkhorn-like projections with an accelerated proximal gradient method.

As the regularized solution is not sparse, and sparsity of transport map is essential for support subset selection, we adopt an inexact Bregman proximal point method (Xie et al., 2020) to yield solutions closer to the original, unregularized, linear program. The work by Xie et al. (2020) is based on the observation that solving the entropically regularized optimal transport problem is equivalent to a Bregman proximal point evaluation with Kullback-Leilber divergence as proximal function. As a proximal point method that uses exact proximal point evaluations is computationally expensive, an inexact proximal point evaluation, where only a few inner Sinkhorn iterations are used, is used to make the algorithm efficient. In our case, the computational complexity is on the same order as the accelerated proximal gradient method, but it returns solution much closer to the linear program for the semi-relaxed partial optimal transport problem. While other regularization-based methods may induce sparse support, the regularized solution will generally be distinct from the linear program's solutions.

---

[1]Dykstra's algorithm can find solutions at the intersection of convex, not necessarily affine, sets.

The contributions of this paper are the following:

- We motivate and study support susbset selection (SS) a specific formulation of a semi-relaxed partial optimal transport problem for selecting a subset of a source distribution, parameterized in terms of a single scalar $c \geq 1$ for a fixed target distribution.

- We study the solutions obtained along the scaling path for various values of $c$, and compare the solution path to fully-relaxed partial and divergence-based semi-relaxed unbalanced optimal transport problems.

- We develop an accelerated proximal gradient method-based algorithm to solve the entropically regularized version and adapt the inexact Bregman proximal method-based approach for optimal transport (POT) (Xie et al., 2020) to mitigate the effects of entropic regularization detailing an algorithm `SS-Bregman` that yields solutions close to the linear program formulation for subset selection and is as scalable as the Sinkhorn algorithm.

- We apply `SS-Bregman` to applications including color adaptation, partial distribution alignment, partial point cloud registration problems, and positive-unlabeled learning (Bekker & Davis, 2020).

- We incorporate the subset selection-based approach into a semi-supervised loss function for training a neural network-based classifier, which computes the optimal transport based on the learning representation.

## 2  Methodology

In Section 2.1, relevant preliminaries related to discrete optimal transport along with formulation of subset selection problem are discussed. In Section 2.3, the entropically regularized support subset selection problem and an algorithm to solve it are discussed. In Section 2.4, an inexact Bregman proximal point method to better approximate the solution of the unregularized support selection problem is detailed.

**Notation**: The set of the first $n$ natural numbers $\{1, 2, \ldots, n\}$, is denoted by $[n]$. The set of integers is denoted by $\mathbb{Z}$. The ceiling function defined on real numbers $x \in \mathbb{R}$ is $\lceil x \rceil = \min\{n \in \mathbb{Z} : n \geq x\}$. The floor function defined on real numbers $x \in \mathbb{R}$ is $\lfloor x \rfloor = \max\{n \in \mathbb{Z} : n \leq x\}$. The $n$-dimensional real vector space is denoted by $\mathbb{R}^n$. Vectors are typeset in lowercase bold ($\boldsymbol{x}$); matrices are in uppercase bold ($\boldsymbol{X}$); and bold is dropped when an element are referenced by subscripts ($x_i, X_{ij}$). When needed for clarity, elements will be referenced by subscripts on square brackets ($[\boldsymbol{x}_1]_i, [\boldsymbol{X}_2]_{ij}$). The set of non-negative vectors in $\mathbb{R}^n$, known as the non-negative orthant, is denoted by $\mathbb{R}^n_+$. The $n$-dimensional vector with all elements equal to unity is denoted by $\mathbf{1}_n$ and the $m$-by-$n$ matrix with all unity elements is denoted by $\mathbf{1}_{m \times n}$. For vectors and matrices, the symbol $\preccurlyeq$ denotes element-wise less than or equal to, and $\succcurlyeq$ denotes element-wise greater than or equal to. The set denoted by $\boldsymbol{\Delta}_n = \{\boldsymbol{x} \in \mathbb{R}^n_+ : \sum_{i=1}^n x_i = 1\}$ is the probability simplex. The element-wise product for vectors and matrices is denoted by the $\odot$ symbol. The element-wise division for vectors and matrices is denoted by the $\oslash$ symbol. The diagonal operator is a matrix valued map $\boldsymbol{D} : \mathbb{R}^n \to \mathbb{R}^{n \times n}$, such that $[\boldsymbol{D}(\boldsymbol{x})]_{ii} = x_i \; \forall \; i \in [n]$ and $[\boldsymbol{D}(\boldsymbol{x})]_{ij} = 0 \; \forall \; i \neq j \in [n]$. For $\boldsymbol{x} \in \mathbb{R}^n$, the $\ell_1$, $\ell_2$ and $\ell_\infty$ norms are given by $\|\boldsymbol{x}\|_1 = \sum_i |x_i|$, $\|\boldsymbol{x}\|_2 = (\sum_i |x_i|^2)^{\frac{1}{2}}$ and $\|\boldsymbol{x}\|_\infty = \max_i |x_i|$, respectively. Both, the Euclidean inner-product for $\boldsymbol{x}, \boldsymbol{y} \in \mathbb{R}^n$ given by $\sum_i x_i y_i$, and the Frobenius inner-product for $\boldsymbol{X}, \boldsymbol{Y} \in \mathbb{R}^{m \times n}$ given by $\sum_{i=1}^m \sum_{j=1}^n X_{ij} Y_{ij}$, are denoted by $\langle \cdot, \cdot \rangle$. The element-wise exponent of a vector or a matrix is denoted by $\mathbf{exp}(\cdot)$ and the element-wise logarithm of a vector or a matrix is denoted by $\mathbf{log}(\cdot)$. For a matrix $\boldsymbol{X} \in \mathbb{R}^{m \times n}$, its non-negative part is represented by $\boldsymbol{X}_+$ or $[\boldsymbol{X}]_+$, with matrix components given by $[\boldsymbol{X}_+]_{ij} = \max\{X_{ij}, 0\} \; \forall \; i \in [m], \; j \in [n]$. Similarly, the non-positive part of matrix $\boldsymbol{X} \in \mathbb{R}^{m \times n}$ is denoted by $\boldsymbol{X}_-$ or $[\boldsymbol{X}]_-$ and contains matrix elements $[\boldsymbol{X}_-]_{ij} = \min\{X_{ij}, 0\} \; \forall \; i \in [m]$ and $j \in [n]$. For a vector $\boldsymbol{x} \in \mathbb{R}^n$, both $\boldsymbol{x}_+$ and $\boldsymbol{x}_-$ denote the non-negative and non-positive parts respectively. We denote the indicator function of the singleton set $\{\boldsymbol{z}\}$ as $\delta_{\boldsymbol{z}}(\boldsymbol{x}) = \begin{cases} 1, & \boldsymbol{x} = \boldsymbol{z} \\ 0, & \boldsymbol{x} \neq \boldsymbol{z} \end{cases}$. The set of vectors $\{\boldsymbol{e}_i\}_{i=1}^n$ form the standard basis for $\mathbb{R}^n$, where $[\boldsymbol{e}_i]_i = 1$ and $[\boldsymbol{e}_i]_j = 0$ for $i \neq j$.

## 2.1 Problem Formulation

We consider the discrete optimal transport between two weighted samples of size $m$ and $n$ corresponding to random variables $X \sim \mu$ defined on $\{\boldsymbol{x}^{(i)}\}_{i=1}^m \subset \mathbb{R}^d$ and $Y \sim \nu$ defined on $\{\boldsymbol{y}^{(j)}\}_{j=1}^n \subset \mathbb{R}^d$, with probability measures $\mu = \sum_{i=1}^m \mu_i \delta_{\boldsymbol{x}^{(i)}}$ and $\nu = \sum_{j=1}^n \nu_j \delta_{\boldsymbol{y}^{(j)}}$ for probability masses $\boldsymbol{\mu} \in \boldsymbol{\Delta}_m$ (with $\{\mu_i = \mu(\boldsymbol{x}^{(i)})\}_{i=1}^m$) and $\boldsymbol{\nu} \in \boldsymbol{\Delta}_n$ (with $\{\nu_j = \nu(\boldsymbol{y}^{(j)})\}_{j=1}^n$), respectively. Let $\mathrm{d} : \mathbb{R}^d \times \mathbb{R}^d \to \mathbb{R}_+$ denote a distance function. In practice, this is often the Euclidean distance metric between the points $\mathrm{d}(\boldsymbol{x}, \boldsymbol{y}) = \|\boldsymbol{x} - \boldsymbol{y}\|_2$. Given $1 \le p \le \infty$, the $p$-Wasserstein distance (to the $p$-power) between $\mu$ and $\nu$ is expressed in terms of the cost matrix $\boldsymbol{M}$, where $M_{ij} = \mathrm{d}^p(\boldsymbol{x}^{(i)}, \boldsymbol{y}^{(j)}) \; \forall i \in [m], j \in [n]$ is the cost associated with transporting $\boldsymbol{x}^{(i)}$ to $\boldsymbol{y}^{(j)}$, as

$$\mathcal{W}_p^p(\mu, \nu) := \min_{\boldsymbol{P} \succcurlyeq 0} \quad \langle \boldsymbol{P}, \boldsymbol{M} \rangle \quad \text{s.t.} \quad \boldsymbol{P}\mathbf{1}_n = \boldsymbol{\mu}, \; \boldsymbol{P}^\top \mathbf{1}_m = \boldsymbol{\nu}, \tag{1}$$

where $\boldsymbol{P}$ is the transport map and $\mathbf{1}_m^\top \boldsymbol{P} \mathbf{1}_n = \mathbf{1}_m^\top \boldsymbol{\mu} = \boldsymbol{\nu}^\top \mathbf{1}_n = 1$. The constraints ensure that any solution $\boldsymbol{P}^*$ is a joint distribution that couples the target marginal $\boldsymbol{\mu}$ and the source marginal $\boldsymbol{\nu}$. (In the computational optimal transport literature, $\mu$ is referred to as the target and $\nu$ as the source.) Therefore, any Wasserstein distance requires the complete mass transfer between a fixed source and target.

In partial optimal transport (Figalli, 2010), the marginal equality constraints are replaced by the inequalities $\boldsymbol{P}\mathbf{1}_n \preccurlyeq \boldsymbol{\mu}, \; \boldsymbol{P}^\top \mathbf{1}_m \preccurlyeq \boldsymbol{\nu}$, and the equality $\mathbf{1}_m^\top \boldsymbol{P} \mathbf{1}_n = s$; $\boldsymbol{\mu} \in \mathbb{R}_+^m$ and $\boldsymbol{\nu} \in \mathbb{R}_+^n$ may not have equal mass; and the transport map need only transport a fraction of the total mass $s \in [0, \min\{\|\boldsymbol{\mu}\|_1, \|\boldsymbol{\nu}\|_1\}]$.

Motivated by machine learning scenarios with a trusted target sample of data and an additional source of data which cannot be assumed to be of uniform quality, we focus on the semi-relaxed case, where the constraint on the target is fixed $\boldsymbol{P}\mathbf{1}_n = \boldsymbol{\mu}$ with $\|\boldsymbol{\mu}\|_1 = 1$, ensuring the total mass constraint, but relax the constraint on the source, allowing mass to redistribute among the source points $\boldsymbol{\nu}^* = \boldsymbol{P}^\top \mathbf{1}_m \le c\boldsymbol{\nu}$, where $c \ge 1$ is a scaling factor and $\|\boldsymbol{\nu}\|_1 = 1$. The resulting partial optimal transport problem,[2] which we refer to as subset selection (SS), is

$$\min_{\boldsymbol{P} \succcurlyeq 0} \quad \langle \boldsymbol{P}, \boldsymbol{M} \rangle \quad \text{s.t.} \quad \boldsymbol{P}\mathbf{1}_n = \boldsymbol{\mu}, \; \boldsymbol{P}^\top \mathbf{1}_m \preccurlyeq c\boldsymbol{\nu}. \tag{2}$$

Let $\boldsymbol{P}_c^*$ denote an optimal solution, then the source's new mass is $\boldsymbol{\nu}_c^* = \boldsymbol{P}_c^{*\top} \mathbf{1}_m$. Since $\mathbf{1}_m^\top \boldsymbol{\mu} = \mathbf{1}_m^\top \boldsymbol{P}_c^* \mathbf{1}_n = 1$, $\|\boldsymbol{\nu}_c^*\|_1 = 1$. Intuitively, this problem allows the new mass of some source points that have relatively lower cost to increase by a factor of $c$ of the original mass, which enables higher cost source points to have less or even zero mass. In other words, due to total unit mass constraint, the mass increment at one source point results in its decrement at other source points. The subset of the source points selected is $\mathrm{supp}(\boldsymbol{\nu}_c^*)$, where $\mathrm{supp}(\cdot)$ indicates the support of a vector, i.e., the indices of the points with non-zero mass.

To explore the relaxed constraint set, we consider the case of a uniformly distributed mass $\boldsymbol{\nu} = \frac{1}{n}\mathbf{1}_n$ and express the constraint as $\boldsymbol{P}^\top \mathbf{1}_m \preccurlyeq \frac{1}{L}\mathbf{1}_n$, where $0 < L \le n$ and $c = \frac{n}{L}$. For a fixed value of $L$, the set of feasible source marginal distributions form a polyhedral set $\boldsymbol{\Xi}_n^{(L)} \subseteq \boldsymbol{\Delta}_n$ bounded by linear inequalities parameterized by $L$. The set of feasible source marginals $\boldsymbol{\Xi}_n^{(L)}$ is defined as $\boldsymbol{\Xi}_n^{(L)} = \{\boldsymbol{x} \in \boldsymbol{\Delta}_n : \boldsymbol{x} \preccurlyeq \frac{1}{L}\mathbf{1}_n\}$. The set $\boldsymbol{\Xi}_3^{(L)}$ for different values of $L$ are given in Figure 1. Using combinatoric reasoning we deduce the number of vertices of $\boldsymbol{\Xi}_n^{(L)}$ defining the feasible set for the source marginals.

**Remark 1.** *For the support selection partial optimal transport problem 2 with $n > 2$ and $\boldsymbol{\nu} = \frac{1}{n}\mathbf{1}_n$, by defining $L = \frac{n}{c}$, the inequality constraint can be written as $\boldsymbol{P}^\top \mathbf{1}_m \preccurlyeq \frac{1}{L}\mathbf{1}_n$. Extreme points of $\boldsymbol{\Xi}_n^{(L)}$ can be characterized as follows:*

- *For $0 < L \le 1$, the entire probability simplex $\boldsymbol{\Delta}_n$ is feasible due to the fact that in this case the vertices of the probability simplex correspond to extreme points of feasible set.*

- *For $1 < L < 2$ and $n > 2$, the feasible set $\boldsymbol{\Xi}_n^{(L)}$ has $n(n-1)$ number of vertices, which can can be written as as convex combination $\frac{1}{L}\boldsymbol{e}_i + (1 - \frac{1}{L})\boldsymbol{e}_j$, where $i, j \in [n]$ and $i \ne j$.*

- *For $L = 2$, the feasible set $\boldsymbol{\Xi}_n^{(2)}$ has $\frac{n(n-1)}{2}$ vertices given as $\frac{1}{2}(\boldsymbol{e}_i + \boldsymbol{e}_j)$ for $i \ne j$.*

---

[2]An equivalent set of constraints are $\boldsymbol{P}\mathbf{1}_n \preccurlyeq \boldsymbol{\mu}, \; \boldsymbol{P}^\top \mathbf{1}_m \preccurlyeq c\boldsymbol{\nu}, \mathbf{1}_m^\top \boldsymbol{P} \mathbf{1}_n = 1$.

- *More generally, the number of extreme points of the feasible set $\mathbf{\Xi}_n^{(L)}$ is*

$$\frac{n!}{\lfloor L \rfloor! \lceil 1 - \frac{\lfloor L \rfloor}{L} \rceil!(n - \lfloor L \rfloor - \lceil 1 - \frac{\lfloor L \rfloor}{L} \rceil)!},$$

*with the vertices given as the set of possible multi-set permutations of the vector:*

$$\left[\overbrace{\tfrac{1}{L}\ \ \tfrac{1}{L}\ \ \cdots\ \ \tfrac{1}{L}}^{\lfloor L \rfloor\ terms}\ \ 1-\tfrac{\lfloor L \rfloor}{L}\ \ \overbrace{0\ \ 0\ \ \cdots\ \ 0}^{n-1-\lfloor L \rfloor\ terms}\right]^{\top} = \left[\overbrace{\tfrac{c}{n}\ \ \tfrac{c}{n}\ \ \cdots\ \ \tfrac{c}{n}}^{\lfloor \frac{n}{c} \rfloor\ terms}\ \ 1-\tfrac{c}{n}\lfloor\tfrac{n}{c}\rfloor\ \ \overbrace{0\ \ 0\ \ \cdots\ \ 0}^{n-1-\lfloor \frac{n}{c} \rfloor\ terms}\right]^{\top}. \quad (3)$$

*Thus, when $L \geq 2$ is an integer, which corresponds to $c \leq \frac{n}{2}$ and $\frac{1}{c} \mod \frac{1}{n} = 0$ the vertices correspond to uniform distribution of mass across a support of cardinality $L = \frac{n}{c}$. However, due to the equality constraints on the other marginal the optimal solution may be a convex combination of these vertices.*

Since the amount of mass to be transported is constant, when the scaling parameter $c$ is increased (or equivalently, when the value of $L$ is decreased) starting from $c = 1$, the behavior of the coupling $\boldsymbol{P}_c^*$, and the marginal $\nu_c^* = (\boldsymbol{P}_c^*)^{\top}\mathbf{1}_n$ in the transportation problem is affected. This behavior depends on the structure of the cost matrix $\boldsymbol{M}$ and leads to a redistribution of mass, assigning more mass to certain points and less to others. At $c = 1$, corresponding to standard optimal transport, all source constraints are active, meaning that all constraints in the problem are considered. As the value of $c$ is increased, constraints become inactive, which constraints depends on the structure of the cost matrix $\boldsymbol{M}$ and the distributions. Once a point's mass goes to zero it never re-enters the support for larger values of $c$. These observations follow from the linear nature of the problem. Analogous solution paths have been studied for fully-relaxed optimal transport (Phatak et al., 2023) and fully or semi-relaxed regularized unbalanced optimal transport (Chapel et al., 2021). Once $c$ reaches $c^*$, all inequality constraints are inactive and can be discarded. After this breakpoint $c^*$, any further increments in $c$ do not affect the resulting transport plans. In other words, for $c \geq c^*$, the transport plan $\boldsymbol{P}_c^*$ remains the same as $\boldsymbol{P}_{c^*}^*$, which is a solution that can found by a greedy algorithm. The exact value of $c^*$ can be determined analytically, based on the possible greedy solutions. The analytical expression for $c^*$ depends on the properties of the cost matrix $\boldsymbol{M}$ and the constraints involved,

$$c^* = \max_{j \in [n]} \frac{\sum_i Q_{ij}}{\nu_j}, \quad (4)$$

where the matrix $\boldsymbol{Q}$ is found by nearest neighbor search,

$$Q_{ij} = \begin{cases} \mu_i, & j \in \arg\min_{k \in [n]} M_{ik} \\ 0, & \text{otherwise} \end{cases}, \quad i \in [m], j \in [n]. \quad (5)$$

$\boldsymbol{P}_{c^*}^*$ is a greedy solution with non-zero entries taken from $\boldsymbol{Q}$ by taking one non-zero element in each row, breaking ties arbitrarily. Thus, as $c$ varies from 1 to $c^*$, SS varies from standard optimal transport to a nearest-neighbor transport.

For more flexibility, a designer can provide an upper-bound on the mass assignments to source points by $\boldsymbol{\zeta} \in \mathbb{R}_+^n$ with $\|\boldsymbol{\zeta}\|_1 \geq 1$ (equality corresponds to standard optimal transport), which adds the flexibility in designing partial optimal transport problems that allow variable ranges of masses for the source points. The upper-bound can be expressed in terms of a capacity vector $\boldsymbol{\kappa} \in \mathbb{R}_+^n$ such that $\boldsymbol{\zeta} = \boldsymbol{\kappa} \odot \boldsymbol{\nu}$ as introduced in the work by Rabin et al. (2014). For target distribution $\mu$ and source upper-bounding measure $\zeta = \sum_{j=1}^m \zeta_j \delta_{\boldsymbol{y}^{(j)}}$, which is not necessarily a probability measure, the support subset selection problem can be stated as

$$\mathcal{S}_p(\mu, \zeta) := \min_{\boldsymbol{P} \succcurlyeq 0}\ \langle \boldsymbol{P}, \boldsymbol{M} \rangle \quad \text{s.t.} \quad \boldsymbol{P}\mathbf{1}_n = \boldsymbol{\mu},\ \boldsymbol{P}^{\top}\mathbf{1}_m \preccurlyeq \boldsymbol{\zeta}, \quad (6)$$

and related to the $p$-Wasserstein distance (to the $p$-power) by $\mathcal{S}_p(\mu, \zeta)\big|_{\zeta = c\nu} = \mathcal{S}_p(\mu, c\nu) \leq \mathcal{W}_p^p(\mu, \nu)$ for $c \geq 1$.

## 2.2 Relation to Prior Work

While we consider a purely linear program consisting of a semi-relaxed partial optimal transport, prior work includes fully-relaxed partial optimal transport and a variety of non-linear approaches for fully or semi-relaxed partial and unbalanced optimal transport.

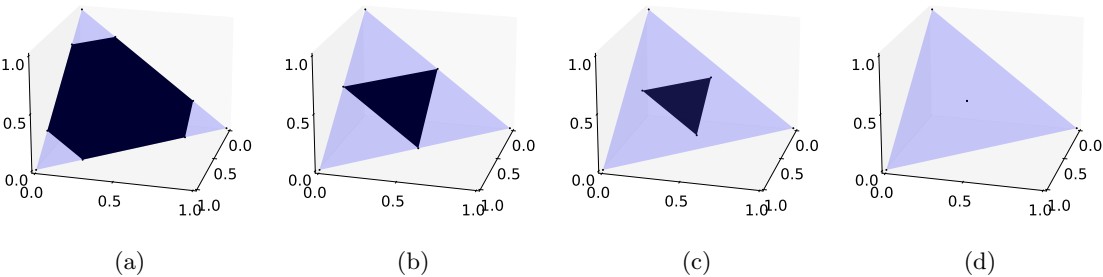

Figure 1: The feasible set $\boldsymbol{\Xi}_3^{(L)}$ of the source's new marginal distribution given a uniform distribution $\boldsymbol{\nu}$ for different values of $L$. For $0 < L \leq 1$ the whole probability simplex is feasible. (a) $L = 1.3$, six vertices will exist for $1 < L < 2$. (b) $L = 2$ yields 3 vertices. (c) $L = 2.2$, $2 < L < 3$ also give 3 vertices. (d) $L = 3$ yields a singleton set corresponding to the original uniform distribution.

### 2.2.1 Divergence-based semi-relaxed partial and unbalanced optimal transport

Many relaxed approaches for unbalanced and partial optimal transport penalize the divergence of the marginal from the source and/or target distribution (Rabin et al., 2014; Frogner et al., 2015; Chizat et al., 2018; Séjourné et al., 2019). Many of these also use entropic regularization for scalable algorithms since entropic regularization of the joint distribution defining the transport plan can be cast as the Kullback-Leibler divergence between the joint and the product of the marginals. The works by Chizat et al. (2018) and Séjourné et al. (2019) provides an extensive framework for entropic regularization and divergence-based relaxations. The latter work (Séjourné et al., 2019) mentions the adjustments necessary to perform asymmetric marginal penalty case for the semi-relaxed case.

The Kullback-Leibler divergence is a member of the wider family of $f$-divergences that have been adopted in relaxed and semi-relaxed optimal transport (Chizat et al., 2018; Séjourné et al., 2019). In the discrete distribution case $\boldsymbol{p}, \boldsymbol{q} \in \boldsymbol{\Delta}_n$, these divergences can be expressed as $\mathcal{D}_\varphi(\boldsymbol{p}\|\boldsymbol{q}) := \sum_{i=1}^n q_i \varphi(\frac{p_i}{q_i})$, where $\varphi$ is the generating function of the divergence. Notably, $\varphi(r) = \mathrm{KL}(r) := r \log r - r + 1$ is the generating function for the KL divergence, and $\varphi(r) = \mathrm{TV}(r) := \frac{1}{2}|r - 1|$ is the generating function for total variation, which in the discrete case is equivalent to an $\ell_1$-norm based distance: $\mathcal{D}_{\mathrm{TV}}(\boldsymbol{p}\|\boldsymbol{q}) = \frac{1}{2}\sum_{i=1}^n q_i |\frac{p_i}{q_i} - 1| = \frac{1}{2}\|\boldsymbol{p} - \boldsymbol{q}\|_1$.

The family of semi-relaxed optimal transport problems with divergence penalty $\frac{1}{\rho} > 0$ on the marginal is

$$\mathcal{SRW}_\varphi^{(\rho)}(\mu, \nu) := \min_{\boldsymbol{P} \geqslant 0} \quad \langle \boldsymbol{P}, \boldsymbol{M} \rangle + \frac{1}{\rho}\mathcal{D}_\varphi\big(\boldsymbol{P}^\top \mathbf{1}_m \| \boldsymbol{\nu}\big) \quad \text{s.t.} \quad \boldsymbol{P}\mathbf{1}_n = \boldsymbol{\mu}. \tag{7}$$

The choice $\varphi(r) = \imath_{[0,c]}(r) := \begin{cases} 0, & r \in [0, c], \\ \infty, & \text{otherwise,} \end{cases}$ yields a range constraint to the interval of $[0, c]$ for the ratio of the source marginal (Chizat et al., 2018; Séjourné et al., 2019), which is equivalent to $\mathcal{S}_p(\mu, c\nu)$ for any value of $\rho$ and $c \geq 1$: $\mathcal{SRW}_{\imath_{[0,c]}}(\mu, \nu) = \mathcal{S}_p^p(\mu, c\nu)$.

More generally, one can consider penalties which are not $f$-divergences such as those based on $\ell_\infty$-norm $\mathcal{D}_{\ell_\infty}(\boldsymbol{P}^\top \mathbf{1}_m \| \boldsymbol{\nu}) := \|(\boldsymbol{P}^\top \mathbf{1}_m) \oslash \boldsymbol{\nu} - \mathbf{1}\|_\infty$ and the squared $\ell_2$-norm $\mathcal{D}_{\ell_2}(\boldsymbol{P}^\top \mathbf{1}_m \| \boldsymbol{\nu}) := \|\boldsymbol{P}^\top \mathbf{1}_m - \boldsymbol{\nu}\|_2^2$ (Benamou, 2003; Blondel et al., 2018; Chapel et al., 2021). The work by Chapel et al. (2021) details a regularization path algorithm for finding the breakpoints in terms of $\rho$ of the piecewise linear related solutions along the path, and efficiently calculating the solutions via rank-1 updates of matrix inverse required to solve the sequence of non-negative, penalized linear regression problems.

Solutions to all of the penalty forms can also be found via equivalent constraint-based optimizations with the constraint $\mathcal{D}_\varphi(\boldsymbol{P}^\top \mathbf{1}_m \| \boldsymbol{\nu}) \leq a$, where $a \geq 0$. In particular, for $c \geq 2$, $\mathcal{S}_p(\mu, c\nu)$, the SS problem 2, is

equivalent to the constraint-based optimization with the $\ell_\infty$-norm based penalty by letting $a = c - 1$, since

$$\|(\boldsymbol{P}^\top \mathbf{1}_m) \oslash \boldsymbol{\nu} - \mathbf{1}_n\|_\infty \leq a \implies \forall j \in [n], \quad -a\nu_j \leq [\boldsymbol{P}^\top \mathbf{1}_m]_j - \nu_j \leq a\nu_j$$

$$\implies \forall j \in [n], \quad (1-a)\nu_j \leq [\boldsymbol{P}^\top \mathbf{1}_m]_j \leq (a+1)\nu_j \implies (2-c)\boldsymbol{\nu} \preccurlyeq \boldsymbol{P}^\top \mathbf{1}_m \preccurlyeq c\boldsymbol{\nu},$$

where the lower bound is non-positive for $c \geq 2$. However, for $1 < c < 2$ or for other choices of $\mathcal{D}_\varphi$ the feasible set of the marginal differs from the linear program.

The work by Rabin et al. (2014) presents a semi-relaxed optimal transport that combines an additional $\ell_1$-norm penalty on the capacity's deviation with an inequality constraint on the marginal, which ensures the problem in terms of the transport plan $\boldsymbol{P}$ stays linear,

$$\min_{\substack{\boldsymbol{P} \succcurlyeq 0 \\ \boldsymbol{\kappa} \in \mathbb{R}_+^n}} \quad \langle \boldsymbol{P}, \boldsymbol{M} \rangle + \frac{1}{\rho}\|\boldsymbol{\kappa} - \mathbf{1}_n\|_1 \qquad \overset{\boldsymbol{\zeta} = \boldsymbol{\kappa} \odot \boldsymbol{\nu}}{=} \qquad \min_{\substack{\boldsymbol{P} \succcurlyeq 0 \\ \boldsymbol{\zeta} \in \mathbb{R}_+^n}} \quad \langle \boldsymbol{P}, \boldsymbol{M} \rangle + \frac{1}{\rho}\|\boldsymbol{\zeta} \oslash \boldsymbol{\nu} - \mathbf{1}_n\|_1$$

$$\text{s.t.} \quad \boldsymbol{P}\mathbf{1}_n = \boldsymbol{\mu}, \ \boldsymbol{P}^\top \mathbf{1}_m \preccurlyeq \boldsymbol{\kappa} \odot \boldsymbol{\nu}, \ \langle \boldsymbol{\kappa}, \boldsymbol{\nu} \rangle \geq 1 \qquad \text{s.t.} \quad \boldsymbol{P}\mathbf{1}_n = \boldsymbol{\mu}, \ \boldsymbol{P}^\top \mathbf{1}_m \preccurlyeq \boldsymbol{\zeta}, \ \boldsymbol{\zeta}^\top \mathbf{1}_n \geq 1.$$

For a given $\rho$, there is a value of $a \geq 0$ such that the constraint-based optimization problem

$$\min_{\substack{\boldsymbol{P} \succcurlyeq 0 \\ \boldsymbol{\zeta} \in \mathbb{R}_+^n}} \quad \langle \boldsymbol{P}, \boldsymbol{M} \rangle \quad \text{s.t.} \quad \boldsymbol{P}\mathbf{1}_n = \boldsymbol{\mu}, \ \boldsymbol{P}^\top \mathbf{1}_m \preccurlyeq \boldsymbol{\zeta}, \ \boldsymbol{\zeta}^\top \mathbf{1}_n \geq 1, \ \|\boldsymbol{\zeta} \oslash \boldsymbol{\nu} - \mathbf{1}_n\|_1 \leq a, \tag{8}$$

has an equivalent solution. The $\ell_1$-norm based divergence constraint will induce sparsity in the deviations between $\boldsymbol{\zeta} \oslash \boldsymbol{\nu}$ and $\mathbf{1}_n$, such that many of the points maintain the corresponding value of $\boldsymbol{\nu}$. In the uniform distribution case $\boldsymbol{\nu} = \frac{1}{n}\mathbf{1}_n$, the solution for $\boldsymbol{\zeta}$ has a maximum value of $\frac{a}{n}$ such that $\boldsymbol{\zeta} \oslash \boldsymbol{\nu} = a$ , the vertices of the feasible set for the constraint-based formulations include permutations of the vector

$$\left[ \frac{a}{n} \quad \overbrace{\frac{1}{n} \quad \frac{1}{n} \quad \cdots \quad \frac{1}{n}}^{n - \lceil a \rceil \text{ terms}} \quad \frac{\lceil a \rceil - a}{n} \quad \overbrace{0 \quad 0 \quad \cdots \quad 0}^{\lceil a \rceil - 2 \text{ terms}} \right]^\top .$$

This can be compared to the vertices of the feasible set for the marginals of the SS problem 2 explored in Remark 1, which have more uniform distribution of mass. More uniform mass distribution is motivated by the maximum entropy principle and desirable such as machine learning tasks such as semi-supervised learning where the goal is to augment the learning based on additional diversity.

By replacing the $\ell_1$-norm with the $\ell_\infty$-norm in the capacity-based optimization problem 8 can be related to $\mathcal{S}_p(\mu, c\nu)$ for all values of $c \geq 1$ by setting $a = c - 1$ since $\zeta_j$ is only involved as an upper bound, no longer considering the lower bound of $(2-c)\nu_j$, for $[\boldsymbol{P}^\top \mathbf{1}_m]_j = \sum_{i=1}^m P_{ij} \leq \zeta_j \leq c\nu_j \quad \forall j \in [n]$, yielding the SS problem 2.

### 2.2.2 Strictly uniform, semi-relaxed partial optimal transport

The work by Chapel et al. (2020), introduced in the context of positive-unlabeled (PU) learning, further constrains the semi-relaxed partial optimal transport problem such that the non-zero source masses must be $\frac{1}{n}$, the PU Wasserstein optimal transport problem is

$$\mathcal{PUW}_p^p(\mu, \nu; s) := \min_{\boldsymbol{T} \succcurlyeq 0} \quad \langle \boldsymbol{T}, \mathbf{M} \rangle \quad \text{s.t.} \quad \mathbf{1}_m^\top \boldsymbol{T} \mathbf{1}_n = s, \ \boldsymbol{T}\mathbf{1}_n \preccurlyeq \frac{s}{m}\mathbf{1}_m, \ \boldsymbol{T}^\top \mathbf{1}_m \in \{0, \frac{1}{n}\}^n. \tag{9}$$

Due to the constraint that the marginal source masses are in the set $\{0, \frac{1}{n}\}$ this problem is not a linear program and only has a non-empty feasible set when $s \mod 1/n = 0$, since $s$ must be an integer multiple of $\frac{1}{n}$, which is an analogous condition to having a uniform distribution among the support as discussed in Remark 1. If this constraint is relaxed to $\boldsymbol{T}^\top \mathbf{1}_m \preccurlyeq \frac{1}{n}\mathbf{1}_n$, then the problem is equivalent to semi-relaxed SS problem 2, which by the linearity of the problem may result in a solution satisfying the original constraint $\boldsymbol{T}^\top \mathbf{1}_m \in \{0, \frac{1}{n}\}^n$. To solve this combinatoric problem, the work by Chapel et al. (2020) obtains the solution to a convex minimization problem involving group LASSO regularization and additional dummy points, as in problems for unbalanced optimal transport (Guittet, 2002) to account for dropped mass. The solution to this problem will create a strictly uniform distribution (after renormalization) amongst the selected subset of the source marginal. That is, the group LASSO regularization induces a solution whose marginal is a vertex of the feasible set discussed in Remark 1.

### 2.2.3 Fully-relaxed partial optimal transport

While the formulation we adopt for subset selection is only for the source marginal (a semi-relaxed formulation of optimal transport), partial optimal transport formulations can achieve support subset selection on both marginals using a fully-relaxed optimization (Figalli, 2010; Phatak et al., 2023). Adapting the notation in the work by Chapel et al. (2020), the partial optimal transport problem is

$$\mathcal{PW}_p^p(\mu, \nu; s) := \min_{\boldsymbol{T} \succcurlyeq 0} \ \langle \boldsymbol{T}, \mathbf{M} \rangle \quad \text{s.t.} \quad \boldsymbol{T}\mathbf{1}_n \preccurlyeq \boldsymbol{\mu}, \ \boldsymbol{T}^\top \mathbf{1}_m \preccurlyeq \boldsymbol{\nu}, \ \mathbf{1}_m^\top \boldsymbol{T}\mathbf{1}_n = s, \tag{10}$$

where $s \in (0, 1]$ is the fraction of the mass transported. When the source distribution is uniform $\boldsymbol{\mu} = \frac{1}{n}\mathbf{1}_n$, after renormalization $\boldsymbol{P} = \frac{1}{s}\boldsymbol{T}$, the feasible set for the source marginal $\boldsymbol{P}^\top \mathbf{1}_m$ consists of permutations of the vector

$$\Big[\overbrace{\tfrac{1}{ns} \quad \tfrac{1}{ns} \quad \cdots \quad \tfrac{1}{ns}}^{\text{ns terms}} \quad \overbrace{0 \quad 0 \quad \cdots \quad 0}^{\text{n(1-s) terms}}\Big]^\top,$$

which are also the vertices of the feasible set of the marginals for the semi-relaxed SS problem 2 explored in Remark 1 with $L = ns$. We note that solutions for the fully-relaxed problem are more likely to have marginals with this form due to the lack of the equality constraints for the target marginal, as compared to the semi-relaxed SS problem. However, the main difference of the fully-relaxed approaches compared to our semi-relaxed SS approach (or other semi-relaxed approaches) is that points in the target may lose mass or be completely dropped, which is not ideal when the goal is to filter the source distribution for any points similar to the target distribution.

The fully-relaxed optimal transport can be directly related to the divergence-based unbalanced optimal transport problem,

$$\mathcal{FRW}_\varphi^{(\rho)}(\mu, \nu) := \min_{\boldsymbol{P} \succcurlyeq 0} \ \langle \boldsymbol{P}, \boldsymbol{M} \rangle + \frac{1}{\rho} \left( \mathcal{D}_\varphi(\boldsymbol{P}\mathbf{1}_n \| \boldsymbol{\mu}) + \mathcal{D}_\varphi(\boldsymbol{P}^\top \mathbf{1}_m \| \boldsymbol{\nu}) \right). \tag{11}$$

There exists a value of $\rho \geq 0$ such that the fully-relaxed optimal transport with modified cost matrix $\mathbf{M}' = \mathbf{M} - \frac{1}{\rho}\mathbf{1}_m \mathbf{1}_n^\top$ and total variation divergence penalties on both marginals will yield the same solution as $\mathcal{PW}_p^p(\mu, \nu; s)$ Caffarelli & McCann (2010); Chizat et al. (2018); Séjourné et al. (2019). However, as discussed above, a total variation or $\ell_1$-based penalty on only the one marginal with an equality constraint on the other induces a different solution.

For $c = \frac{1}{s} \geq 1$, the fully-relaxed partial optimal transport problem is

$$
\begin{aligned}
&\min_{\boldsymbol{P} \succcurlyeq 0} \ \langle \boldsymbol{P}, \boldsymbol{M} \rangle \\
&\text{s.t.} \quad \boldsymbol{P}\mathbf{1}_n \preccurlyeq c\boldsymbol{\mu}, \ \boldsymbol{P}^\top \mathbf{1}_m \preccurlyeq c\boldsymbol{\nu}, \ \mathbf{1}_m^\top \boldsymbol{P}\mathbf{1}_n = 1,
\end{aligned}
\quad = \quad
\begin{aligned}
&\min_{\boldsymbol{T} \succcurlyeq 0} \ c\langle \boldsymbol{T}, \boldsymbol{M} \rangle \\
&\text{s.t.} \quad \boldsymbol{T}\mathbf{1}_n \preccurlyeq \boldsymbol{\mu}, \ \boldsymbol{T}^\top \mathbf{1}_m \preccurlyeq \boldsymbol{\nu}, \mathbf{1}_m^\top \boldsymbol{T}\mathbf{1}_n = \frac{1}{c},
\end{aligned}
\tag{12}
$$

where $\boldsymbol{T} = \frac{1}{c}\boldsymbol{P}$. Based on equation 12, $c \cdot \mathcal{PW}_p^p(\mu, \nu; \frac{1}{c})$ is the cost for an optimal transport plan with fully-relaxed constraints $\boldsymbol{P}\mathbf{1}_n \preccurlyeq c\boldsymbol{\mu}$ and $\boldsymbol{P}^\top \mathbf{1}_m \preccurlyeq c\boldsymbol{\nu}$.

The recent work by Phatak et al. (2023), studies the value of $\omega(s) := \mathcal{PW}_p^p(\mu, \nu; s)$ across all values of $s \in (0, 1]$, which is known as the OT-profile as introduced in the work of Figalli (2010). In the work by Phatak et al. (2023), $\omega(s)$ is shown to be a piece-wise linear convex function of $s$, and the entire profile can be computed exactly and approximated efficiently. Furthermore, derivative of $\omega$ with respect to $s$ can be used find a mass fraction where the partial transport separates inliers and outliers Phatak et al. (2023).

To relate the fully-relaxed to the semi-relaxed partial optimal transport, we consider the unbalanced partial OT-profile function to denote the case where the target marginal is also scaled down by $s$ such that it is wholly transported

$$\mathcal{PW}_p^p(s\mu, \nu; s) := \min_{\boldsymbol{T} \succcurlyeq 0} \ \langle \boldsymbol{T}, \mathbf{M} \rangle \quad \text{s.t.} \quad \boldsymbol{T}\mathbf{1}_n \preccurlyeq s\boldsymbol{\mu}, \ \boldsymbol{T}^\top \mathbf{1}_m \preccurlyeq \boldsymbol{\nu}, \ \mathbf{1}_m^\top \boldsymbol{T}\mathbf{1}_n = s.$$

Let $c = \frac{1}{s}$, then our proposed subset support cost is

$$
\mathcal{S}_p(\mu, \frac{1}{s}\nu) = \min_{\boldsymbol{P} \succcurlyeq 0} \quad \langle \boldsymbol{P}, \mathbf{M} \rangle \qquad\qquad \min_{\boldsymbol{T} \succcurlyeq 0} \quad \frac{1}{s}\langle \boldsymbol{T}, \mathbf{M} \rangle
$$
$$
\text{s.t.} \quad \mathbf{1}_m^\top \boldsymbol{P} \mathbf{1}_n = 1, \qquad = \qquad \text{s.t.} \quad \mathbf{1}_m^\top \boldsymbol{T} \mathbf{1}_n = s,
$$
$$
\boldsymbol{P} \preccurlyeq \boldsymbol{\mu}, \ \boldsymbol{P}^\top \mathbf{1}_m \preccurlyeq \frac{1}{s}\boldsymbol{\nu} \qquad\qquad \boldsymbol{T}\mathbf{1}_n \preccurlyeq s\boldsymbol{\mu}, \ \boldsymbol{T}^\top \mathbf{1}_m \preccurlyeq \boldsymbol{\nu}.
$$

Thus, $\mathcal{S}_p(\mu, \frac{1}{s}\nu) = \frac{1}{s}\mathcal{PW}_p^p(s\mu, \nu; s)$ and $\frac{1}{c}\mathcal{S}_p(\mu, c\nu) = \mathcal{PW}_p^p(\frac{1}{c}\mu, \nu; \frac{1}{c})$, with optimal solutions related by $\boldsymbol{P}^* = \frac{1}{s}\boldsymbol{T}^*$. Based on this relation we explore the adoption of the knee finding algorithms applied to selection of $s$ (Phatak et al., 2023) to optimize the selection of $c$ in cases where the source differs from the target by the presence of outliers.

## 2.3 Support Subset Selection with Entropic Regularization

The support subset selection problem 6 is a linear program, which can be exactly solved by the simplex method or interior point methods, both of which do not scale well with the dimension of transport map (Cuturi, 2013). In order to apply efficient gradient-based optimization to linear programs, entropic regularization has been added to linear objective functions (Li & Fang, 1997). In the work by Cuturi (2013), entropic regularization is added to the optimal transport problem to efficiently approximate the Wasserstein distance using Sinkhorn's matrix scaling algorithm (Cuturi, 2013; Sinkhorn, 1964). For fixed target distribution $\mu$ and upper-bounding source measure $\zeta$ with mass $\boldsymbol{\zeta} \in \mathbb{R}_+^n$, the proposed entropically regularized support subset selection problem is

$$
\mathcal{S}_p^{(\gamma)}(\mu, \zeta) := \min_{\boldsymbol{P} \succcurlyeq 0} \quad \langle \boldsymbol{P}, \boldsymbol{M} \rangle + \gamma \langle \boldsymbol{P}, \log(\boldsymbol{P}) - \mathbf{1}_{m \times n} \rangle \quad \text{s.t.} \quad \boldsymbol{P}\mathbf{1}_n = \boldsymbol{\mu}, \ \boldsymbol{P}^\top \mathbf{1}_m \preccurlyeq \boldsymbol{\zeta}, \tag{13}
$$

where $\gamma$ is the regularization parameter. In the case of a uniform distribution $\boldsymbol{\nu} = \frac{1}{n}\mathbf{1}_n$, the entropically regularization problem will be equivalent to using a Kullback-Leibler divergence between the joint and product of the given marginals and additional identity and range constraints on the solution's marginals Séjourné et al. (2019), as shown in Appendix D. It is important to mention that the regularization term $\langle \boldsymbol{P}, \log(\boldsymbol{P}) - \mathbf{1}_{m \times n} \rangle$ is negative entropy, which is 1-strongly convex with respect to the $\ell_1$ and $\ell_2$ norms in the feasible set: $\{\boldsymbol{P} : \boldsymbol{P} \succcurlyeq 0, \ \boldsymbol{P}\mathbf{1}_n = \boldsymbol{\mu}, \ \boldsymbol{P}^\top \mathbf{1}_m \preccurlyeq \boldsymbol{\zeta}\}$ (Beck, 2017). The Lagrangian of problem 13 is

$$
\mathcal{L}(\boldsymbol{P}, \boldsymbol{\alpha}, \boldsymbol{\beta}) = \left\langle \boldsymbol{P}, \ \boldsymbol{M} + \gamma(\log \boldsymbol{P} - \mathbf{1}_m \mathbf{1}_n^\top) + \boldsymbol{\alpha}\mathbf{1}_n^\top + \mathbf{1}_m \boldsymbol{\beta}^\top \right\rangle - \langle \boldsymbol{\alpha}, \boldsymbol{\mu} \rangle - \langle \boldsymbol{\beta}, \boldsymbol{\zeta} \rangle, \tag{14}
$$

where $\boldsymbol{\alpha}, \boldsymbol{\beta}$ are the Lagrange multipliers. Note that we have adopted the approach of Cuturi (2013) and do not explicitly enforce the simplex constraint on $\boldsymbol{P}$, which would lead to the log-sum-exp formulation as in the works by Cuturi & Peyré (2018); Lin et al. (2022); Guminov et al. (2021). Taking the element-wise derivative of $\mathcal{L}$ with respect to $\boldsymbol{P}$ and setting it to zero yields

$$
\tilde{\boldsymbol{P}}(\boldsymbol{\alpha}, \boldsymbol{\beta}) = \boldsymbol{D}\Big(\exp(-\boldsymbol{\alpha}/\gamma)\Big)\exp(-\boldsymbol{M}/\gamma)\boldsymbol{D}\Big(\exp(-\boldsymbol{\beta}/\gamma)\Big). \tag{15}
$$

Substituting $\tilde{\boldsymbol{P}}$ back into Lagrangian results in the dual problem

$$
\min_{\boldsymbol{\alpha}, \boldsymbol{\beta}} \quad \left\{ f(\boldsymbol{\alpha}, \boldsymbol{\beta}) := \gamma \mathbf{1}_m^\top \tilde{\boldsymbol{P}}(\boldsymbol{\alpha}, \boldsymbol{\beta})\mathbf{1}_n + \langle \boldsymbol{\alpha}, \boldsymbol{\mu} \rangle + \langle \boldsymbol{\beta}, \boldsymbol{\zeta} \rangle \right\}
$$
$$
\text{s.t.} \quad \boldsymbol{\beta} \succcurlyeq 0. \tag{16}
$$

The constraint set $\boldsymbol{\beta} \succcurlyeq 0$ is closed. The indicator function of the constraint set $\boldsymbol{\beta} \succcurlyeq 0$ is defined as

$$
\imath_+(\boldsymbol{\beta}) := \begin{cases} 0, & \text{for } \boldsymbol{\beta} \succcurlyeq 0 \\ \infty, & \text{otherwise.} \end{cases}
$$

Therefore, we can convert problem 16 into an unconstrained composite optimization problem,

$$
\min_{\boldsymbol{\alpha}, \boldsymbol{\beta}} \quad f(\boldsymbol{\alpha}, \boldsymbol{\beta}) + \imath_+(\boldsymbol{\beta}). \tag{17}
$$

Since $f(\boldsymbol{\alpha}, \boldsymbol{\beta})$ is convex and $\iota_+(\boldsymbol{\beta})$ is proper, closed, and convex, we can apply the accelerated proximal gradient algorithm to solve the composite optimization problem. Defining the Gibbs kernel $\boldsymbol{K} = \exp\left(-\frac{\boldsymbol{M}}{\gamma}\right)$, the partial gradient $\nabla_{\boldsymbol{\beta}} f(\boldsymbol{\alpha}, \boldsymbol{\beta})$ is

$$\nabla_{\boldsymbol{\beta}} f(\boldsymbol{\alpha}, \boldsymbol{\beta}) = \boldsymbol{\zeta} - \exp\left(-\boldsymbol{\beta}/\gamma\right) \odot \left(\boldsymbol{K}^\top \exp\left(-\boldsymbol{\alpha}/\gamma\right)\right). \tag{18}$$

The proximal projection for the non-negative orthant's indicator function $\iota_+$ is computed by setting any negative entries to zero.

Algorithm 1 (`SS-Entropic`) outlines our accelerated proximal gradient algorithm to solve the dual form of subset selection problem with entropic regularization. Similar to the standard entropically regularized optimal transport problem, the dual variable $\boldsymbol{\alpha}$ is updated with a Sinkhorn-like update at iteration $k$ as

$$\boldsymbol{\alpha}^{(k+1)} = \gamma \log\left(\left(\boldsymbol{K}\exp(-\boldsymbol{\beta}^{(k)}/\gamma)\right) \oslash \boldsymbol{\mu}\right). \tag{19}$$

Whereas, $\boldsymbol{\beta}$ is updated at iteration $k$ using accelerated proximal gradient based update rule (Beck, 2017; Beck & Teboulle, 2009) using the extrapolated point $\boldsymbol{\xi}$ with step size $1/\eta_s^{(k)}$

$$\boldsymbol{\beta}^{(k+1)} = \left[\boldsymbol{\xi}^{(k)} - \frac{1}{\eta_s^{(k)}}\nabla_{\boldsymbol{\xi}} f(\boldsymbol{\alpha}^{(k+1)}, \boldsymbol{\xi}^{(k)})\right]_+ = \left[\boldsymbol{\xi}^{(k)} - \frac{1}{\eta_s^{(k)}}\left(\boldsymbol{\zeta} - \exp(-\boldsymbol{\xi}^{(k)}/\gamma) \odot \boldsymbol{K}^\top \exp\left(-\boldsymbol{\alpha}^{(k+1)}/\gamma\right)\right)\right]_+ \tag{20}$$

$$\boldsymbol{\xi}^{(k+1)} = \boldsymbol{\beta}^{(k+1)} + \frac{t_k - 1}{t_{k+1}}(\boldsymbol{\beta}^{(k+1)} - \boldsymbol{\beta}^{(k)}), \text{ with } t_{k+1} = \frac{1 + \sqrt{1 + 4t_k^2}}{2}, \tag{21}$$

which uses equation 18 to compute the gradient with respect to the variable $\boldsymbol{\xi}^{(k)}$ before applying the proximal operator $[\cdot]_+$. In `SS-Entropic`, we use a constant step size $\frac{1}{\eta_s^{(k)}} = \gamma$ (since the primal problem 16 is $\gamma$-strongly convex and its semi-dual is $\frac{1}{\gamma}$-Lipschitz smooth Cuturi & Peyré (2016), see Appendix A for details), but another option is a backtracking line search (Beck, 2017).

By incorporating the update of $\boldsymbol{\alpha}^{(k+1)}$ as in equation 19 directly into the gradient $\nabla_{\boldsymbol{\beta}} f(\boldsymbol{\alpha}^{(k+1)}, \boldsymbol{\beta}^{(k)})$, the algorithm can be written entirely in terms of $\boldsymbol{\beta}^{(k)}$. This shows that `SS-Entropic` consists of standard accelerated proximal gradient updates and has a $\mathcal{O}(1/k^2)$ convergence rate. As shown in the work by Beck (2017), the required number of iterations $k_\varepsilon$ to achieve an $\varepsilon$ sub-optimal solution of the optimization problem 17 using `SS-Entropic` is upper-bounded as

$$k_\varepsilon + 1 \leq \sqrt{\frac{2}{\gamma\varepsilon}} \cdot \|\boldsymbol{\beta}^{(i)} - \boldsymbol{\beta}^*\|, \tag{22}$$

where $\boldsymbol{\beta}^{(i)}$ is the initialization and $\boldsymbol{\beta}^*$ is the optimal solution.

If `SS-Entropic` is allowed to run until its convergence, it returns the optimal coupling $\boldsymbol{P}^*$, but in practice, if `SS-Entropic` does not reach convergence, $\hat{\boldsymbol{P}}^* \in \mathbb{R}_+^{m \times n}$ may violate the primal constraints on its marginals as these are not ensured by an approximate dual solution. For some applications a projection of $\hat{\boldsymbol{P}}^*$ to satisfy one or both of the marginal constraints may be required. While not explored in this paper due to the additional computational cost, projection to the feasible set can be done by the fast dual proximal gradient (FDPG) algorithm from the works by Beck & Teboulle (2014) and Beck (2017) in conjunction with Algorithm-2 in the work of Altschuler et al. (2017).

## 2.4 Support Subset Selection with the Inexact-Bregman Proximal-point Method

Although the entropic regularization of the coupling distribution enables an efficient approximation of the support subset selection problem 6, the entropic regularization yields denser coupling distributions as compared to the unregularized problem. The denser coupling distributions result in a new marginal mass $\boldsymbol{\nu}^*$ that is also not sparse, yielding complete support rather than a subset of the source points. Different approaches

---

**Algorithm 1:** (`SS-Entropic`) Fast proximal gradient algorithm to solve the dual problem 17 of the entropically regularized support subset selection problem 13

---

      **Inputs**        : Target distribution $\boldsymbol{\mu}$, mass assignment bounding vector $\boldsymbol{\zeta}$, cost matrix $\boldsymbol{M}$, entropic regularization parameter $\gamma$, initial dual variable, $\boldsymbol{\beta}^{(\mathrm{i})} \in \mathbb{R}_+^n$, and iteration limit *max-iter*.

      **Outputs**      : $\hat{\boldsymbol{P}}^*$, which approaches the optimal coupling $\boldsymbol{P}^*$

**1** **Function** `EntropicSS`($\boldsymbol{\mu}$, $\boldsymbol{\zeta}$, $\boldsymbol{\beta}^{(\mathrm{i})}$, $\gamma$, $\boldsymbol{M}$, *max-iter*):

      **Initialization:** $t_0 \leftarrow 1$, $\boldsymbol{\beta}^{(0)} \leftarrow \boldsymbol{\beta}^{(\mathrm{i})}$, $\boldsymbol{\xi}^{(0)} \leftarrow \boldsymbol{\beta}^{(\mathrm{i})}$, $\boldsymbol{K} \leftarrow \exp(-\frac{1}{\gamma}\boldsymbol{M})$

**2**       **for** $k \leftarrow 0$ **to** *max-iter* $- 1$ **do**

**3**            $\boldsymbol{\alpha}^{(k+1)} \leftarrow \gamma \log\left( \left(\boldsymbol{K}\exp(-\frac{1}{\gamma}\boldsymbol{\beta}^{(k)})\right) \oslash \boldsymbol{\mu} \right)$

**4**            $\boldsymbol{\beta}^{(k+1)} \leftarrow \left[ \boldsymbol{\xi}^{(k)} - \gamma\nabla_{\boldsymbol{\xi}}f(\boldsymbol{\alpha}^{(k+1)}, \boldsymbol{\xi}^{(k)}) \right]_+$

**5**            $t_{k+1} \leftarrow \frac{1+\sqrt{1+4t_k^2}}{2}$

**6**            $\boldsymbol{\xi}^{(k+1)} \leftarrow \boldsymbol{\beta}^{(k+1)} + (\frac{t_k-1}{t_{k+1}})(\boldsymbol{\beta}^{(k+1)} - \boldsymbol{\beta}^{(k)})$

**7**       **end**

**8**        $\boldsymbol{\alpha}^* \leftarrow \boldsymbol{\alpha}^{(k+1)}$

**9**        $\boldsymbol{\beta}^* \leftarrow \boldsymbol{\beta}^{(k+1)}$

**10**       $\hat{\boldsymbol{P}}^* \leftarrow \tilde{\boldsymbol{P}}(\boldsymbol{\alpha}^*, \boldsymbol{\beta}^*) = \boldsymbol{D}\left(\exp\left(-\frac{1}{\gamma}\boldsymbol{\alpha}^*\right)\right)\boldsymbol{K}\boldsymbol{D}\left(\exp(-\frac{1}{\gamma}\boldsymbol{\beta}^*)\right)$

**11** **return** $\hat{\boldsymbol{P}}^*$, $\boldsymbol{\alpha}^*$, $\boldsymbol{\beta}^*$

---

have been proposed to maintain the computational benefits of entropic regularization while yielding solutions closer to the unregularized problem (Schmitzer, 2019; Xie et al., 2020).

In this paper, we follow the work by Xie et al. (2020) and adapt an inexact Bregman proximal gradient for the negative entropy function (Teboulle, 1992) to the partial optimal transport case. The Bregman proximal gradient approach uses a proximal operator where the usual Euclidean distance(Parikh & Boyd, 2014) is replaced with the Bregman divergence associated with a continuously differentiable and strictly convex function (Beck, 2017). In the case of negative entropy, the Bregman divergence is the Kullback–Leibler divergence. Let $\phi(\boldsymbol{P}) = \langle \boldsymbol{P}, \log(\boldsymbol{P}) - \mathbf{1}_{m\times n}\rangle$ denote the negative entropy of a non-negative matrix $\boldsymbol{P}$, then given a non-negative matrix $\boldsymbol{P}' \in \mathbb{R}_+^{m\times n}$, the Bregman divergence is

$$\mathcal{B}_\phi(\boldsymbol{P}||\boldsymbol{P}') := \langle \boldsymbol{P}, \log(\boldsymbol{P}\oslash\boldsymbol{P}')\rangle - \langle\boldsymbol{P}, \mathbf{1}_{m\times n}\rangle + \langle\boldsymbol{P}', \mathbf{1}_{m\times n}\rangle. \tag{23}$$

For the subset selection problem 6, the Bregman proximal point evaluated at $\boldsymbol{P}^{(t)}$, is

$$\begin{aligned} \mathbf{Breg\text{-}prox}_\phi(\boldsymbol{P}^{(t)}) = \underset{\boldsymbol{P}\succcurlyeq 0}{\arg\min} \quad &\langle\boldsymbol{P}, \boldsymbol{M}\rangle + \lambda\mathcal{B}_\phi(\boldsymbol{P}||\boldsymbol{P}^{(t)}) \\ \text{s.t.} \quad &\boldsymbol{P}\mathbf{1}_n = \boldsymbol{\mu}, \ \boldsymbol{P}^\top\mathbf{1}_m \preccurlyeq \boldsymbol{\zeta}, \end{aligned} \tag{24}$$

where $\lambda$ is positive scaling factor. By substituting $\mathcal{B}_\phi(\boldsymbol{P}||\boldsymbol{P}^{(t)})$ from equation 23 into equation 24 and ignoring the constant term $\langle\boldsymbol{P}^{(t)}, \mathbf{1}_{m\times n}\rangle$ we obtain

$$\begin{aligned} \mathbf{Breg\text{-}prox}_\phi(\boldsymbol{P}^{(t)}) = \underset{\boldsymbol{P}\succcurlyeq 0}{\arg\min} \quad &\langle\boldsymbol{P}, \boldsymbol{M} - \log(\boldsymbol{P}^{(t)})\rangle + \lambda\langle\boldsymbol{P}, \log(\boldsymbol{P}) - \mathbf{1}_{m\times n}\rangle \\ \text{s.t.} \quad &\boldsymbol{P}\mathbf{1}_n = \boldsymbol{\mu}, \ \boldsymbol{P}^\top\mathbf{1}_m \preccurlyeq \boldsymbol{\zeta}, \end{aligned} \tag{25}$$

which corresponds to the entropically regularized subset selection problem 13 with parameters $\gamma$ and $\boldsymbol{M}$ in 13 replaced by $\lambda$ and $\boldsymbol{M} - \log(\boldsymbol{P}^{(t)})$, respectively. Thus, solving the entropy-regularized support subset selection problem is required to solve an iteration of the proximal-step evaluation problem in equation 25. It has been shown in the work by Xie et al. (2020) that as $t \to \infty$, the iterations $\boldsymbol{P}^{(t+1)} = \mathbf{Breg\text{-}prox}_\phi(\boldsymbol{P}^{(t)})$ converge to an optimal solution of the original unregularized problem. Therefore, to solve 6 we can iteratively

invoke `SS-Entropic` to obtain $\boldsymbol{P}^{(t+1)} = \textbf{Breg-prox}_\phi(\boldsymbol{P}^{(t)})$, while replacing $\gamma$ and $\boldsymbol{M}$ in problem 13 by $\lambda$ and $\boldsymbol{M} - \lambda\boldsymbol{P}^{(t)}$ in problem 25, respectively.

Algorithm 2 (`SS-Bregman`) outlines the steps to solve the subset support selection using the Bregman proximal-point method, where the inner loop is solved by `SS-Entropic`. The nested loops of the exact proximal point algorithm can result in high computational costs, but this can be circumvented by choosing a lower number of iterations for the inner loop—stopping before its convergence. This is justified by the observation that the majority of the progress towards optimal solutions by gradient based methods is achieved during the first few iterations. Recently, an inertial variant of inexact-Bregman proximal point method for the optimal transport has been proposed (Yang & Toh, 2022), which may further accelerate the Bregman proximal point method, but to the best of our knowledge there are no guarantees for accelerated convergence.

---

**Algorithm 2:** (`SS-Bregman`) Inexact-Bregman Proximal Point Algorithm to approximately solve 6 via 25

> **Inputs** : Target distribution $\boldsymbol{\mu}$, mass assignment upper bounding vector $\boldsymbol{\zeta}$, cost matrix $\boldsymbol{M}$, Bregman scaling parameter $\lambda$, and initial dual variable, $\boldsymbol{\beta}^{(i)} \in \mathbb{R}_+^n$, inner-iteration limit *max-inner-iter* and outer-iteration limit *max-outer-iter*
>
> **Outputs** : $\hat{\boldsymbol{P}}^*$
>
> **Initialization:** $\boldsymbol{\beta}^{(0)} \leftarrow \boldsymbol{\beta}^{(i)}, \boldsymbol{P}^{(0)} \leftarrow \frac{1}{mn}\mathbf{1}_{m\times n}$
>
> **1 for** $t \leftarrow 0$ **to** *max-outer-iter* $- 1$ **do**
>
>     // repeatedly invoke Entropic-SS
>
> **2**    $\boldsymbol{P}^{(t+1)}, \boldsymbol{\alpha}^{(t+1)}, \boldsymbol{\beta}^{(t+1)} \leftarrow \texttt{EntropicSS}\left(\boldsymbol{\mu}, \boldsymbol{\zeta}, \boldsymbol{\beta}^{(t)}, \lambda, \boldsymbol{M} - \log(\boldsymbol{P}^{(t)}), \textit{max-inner-iter}\right)$
>
> **3 end**
>
> **4** $\hat{\boldsymbol{P}}^* = \boldsymbol{P}^{(t+1)}$

---

Due to early stopping, `SS-Bregman` can yield infeasible solutions that do not satisfy the marginal constraints. In practice, the number of iterations depends on problem in the hand. For applications related to point cloud registration, color transfer and PU learning, where number of data points in a data-batch is small, the algorithm is allowed to run with a large number of iterations yielding a highly accurate and feasible solution. Whereas, for the applications related to neural network training where training efficiency is more important than the solution accuracy, the algorithm is allowed to run for a smaller number of iterations.

## 2.5 Point Cloud Registration with Subset Selection

Point cloud registration is a well studied problem that tries to find a correspondence of points in one sample (cloud) to another sample (Zhang et al., 2021; Zang et al., 2019). Practical settings include points sampled from the boundaries of 3D images such as captured by LIDAR or the points on the edges of objects in 2D images (Xu et al., 2023). More generally, points correspond to data points in two or more samples. In both of these cases it is useful to consider the case that the two samples exist in different coordinate frames such that there is an affine transformation needed to align the samples before finding the correspondence.

Partial optimal transport for point cloud registration is motivated by cases of occlusion in 2D or 3D imagery. In the case of data, it could be that one sample has dropped modes either by the nature of the data gathering or generating process. Our proposed subset selection algorithms are applicable to cases where the source is assumed to have a complete or overcomplete representation of the target, i.e., only a subset of the target is available and all target points should be maintained.

We propose to use support subset selection as a loss function for optimizing affine transformations in partial point cloud registration. This can be posed as a bi-level optimization problem

$$\min_{\Theta} \min_{\boldsymbol{P} \succcurlyeq 0} \quad \langle \boldsymbol{P}, \hat{\boldsymbol{M}}(\Theta) \rangle \quad \text{s.t.} \quad \boldsymbol{P}\mathbf{1}_n = \boldsymbol{\mu}, \ \boldsymbol{P}^\top\mathbf{1}_m \preccurlyeq c\boldsymbol{\nu}, \tag{26}$$

where $\Theta = [\boldsymbol{A}, \boldsymbol{b}]$ are the parameters of the affine transform, the entries of the cost matrix $\hat{\boldsymbol{M}}(\Theta)$ are $[\hat{\boldsymbol{M}}(\Theta)]_{ij} = \|\boldsymbol{x}_i - \hat{\boldsymbol{y}}_j^\Theta\|_2^2 \quad i \in [m], j \in [n]$ for fixed target $\{\boldsymbol{x}_i\}_{i=1}^m$ and transformed source $\{\hat{\boldsymbol{y}}_j^\Theta = \boldsymbol{A}\boldsymbol{y}_j + \boldsymbol{b}\}_{j=1}^n$.

The standard approach to solve bi-level optimization problems in point cloud registration discussed in the works by Arun et al. (1987) and Myronenko & Song (2010), is an iterative alternating algorithm with two steps, where the sub-problem for the affine transform is solved exactly via ordinary least squares. If the coupling matrix during an iteration is given by $\boldsymbol{P}^*$, the next subproblem is to find the affine transformation parameters $\Theta = [\boldsymbol{A}, \boldsymbol{b}]$ that minimize the weighted squared errors $\sum_{i,j}[\boldsymbol{P}^*]_{ij}\|\boldsymbol{x}_i - (\boldsymbol{A}\boldsymbol{y}_j + \boldsymbol{b})\|_2^2$. The solution can be found analytically in terms of the source mass vector $\boldsymbol{\nu}^* = \boldsymbol{P}^{*\top}\mathbf{1}_m$, weighted means of the target point cloud $\boldsymbol{X} = [\boldsymbol{x}_1, \ldots, \boldsymbol{x}_m]^\top$ and the source point cloud $\boldsymbol{Y} = [\boldsymbol{y}_1, \ldots, \boldsymbol{y}_n]^\top$ as $\bar{\boldsymbol{x}} = \boldsymbol{X}^\top\boldsymbol{\mu}$ and $\bar{\boldsymbol{y}} = \boldsymbol{Y}^\top\boldsymbol{\nu}^*$, and centered point clouds $\tilde{\boldsymbol{X}} = \boldsymbol{X} - \mathbf{1}_m\bar{\boldsymbol{x}}^\top$ and $\tilde{\boldsymbol{Y}} = \boldsymbol{Y} - \mathbf{1}_n\bar{\boldsymbol{y}}^\top$, as

$$
\begin{aligned}
\boldsymbol{A} &= \big(\tilde{\boldsymbol{X}}^\top\boldsymbol{P}^*\tilde{\boldsymbol{Y}}\big)\big(\tilde{\boldsymbol{Y}}^\top\boldsymbol{D}(\boldsymbol{\nu}^*)\tilde{\boldsymbol{Y}}\big)^\dagger, \\
\boldsymbol{b} &= \bar{\boldsymbol{x}} - \boldsymbol{A}\bar{\boldsymbol{y}},
\end{aligned}
\tag{27}
$$

where $(\cdot)^\dagger$ indicates the Moore-Penrose pseudo-inverse.

To find a solution to equation 26, we also use an iterative alternating algorithm with two steps. In contrast to Myronenko & Song (2010), instead of using complete source and target point clouds to obtain affine transformations, during every iteration we draw batches from both source and target point clouds to obtain the coupling matrix $\boldsymbol{P}^*$ and update the affine transformation parameters $\Theta$ via a gradient update. The advantage of this mini-batch based approach is an implicit regularization and faster updates for affine transformation parameters. In the first step, given the affine transformation we obtain an approximate solution $\hat{\boldsymbol{P}}^*$ to the subset selection problem 6 via `SS-Bregman`. In the second step, we used automatic differentiation of the cost $\langle\hat{\boldsymbol{P}}^*, \hat{\boldsymbol{M}}(\Theta)\rangle$ and perform gradient based update for the parameters $\Theta = [\boldsymbol{A}, \boldsymbol{b}]$. It is important to mention that we follow the approach adopted by Xie et al. (2020) for gradient evaluation. Therefore, during an iteration, once the subset set selection map $\hat{\boldsymbol{P}}^*$ is obtained, it is deemed constant for the iteration in consideration, therefore the gradient is: $\nabla_\Theta\langle\hat{\boldsymbol{P}}^*, \hat{\boldsymbol{M}}(\Theta)\rangle = \sum_{i,j}[\hat{\boldsymbol{P}}^*]_{ij}\nabla_\Theta[\hat{\boldsymbol{M}}(\Theta)]_{ij}$. More specifically, we used PyTorch based automatic differentiation for gradient evaluation (Paszke et al., 2017) and the Adam optimizer (Kingma & Ba, 2014) with learning rate of 0.5 for gradient based updates of parameters $\Theta$. To initialize the affine mapping parameters we simply set $\boldsymbol{A}$ and $\boldsymbol{b}$ to the identity matrix and zero vector, respectively. However, since the bi-level optimization problem is not convex, even though the subset selection problem at each iteration is convex, in practice the algorithm could be allowed to run with multiple initialization to obtain the best fit.

## 3 Experimental Results and Discussion

In this section we discuss the application of subset selection. Subsections 3.1 and 3.2 discuss the application of subset selection in toy data sets: point-clouds in 2D and 3D with and without affine transformations and color transfer, respectively. Subsection 3.3 discusses subset selection for positive-unlabeled learning tasks. Subsection 3.4 discusses the application of subset selection for semi-supervised training of neural networks. All the experiments done in this paper use $p = 2$ and the Euclidean distance to define the cost matrix. Unless stated otherwise, experiments use $\boldsymbol{\mu} = \frac{1}{m}\mathbf{1}_m$ and $\boldsymbol{\zeta} = \frac{c}{n}\mathbf{1}_n$, where $c \geq 1$ is the scaling factor.

### 3.1 Subset Selection on Point Clouds

**Circle and Square**: In order to demonstrate the proposed algorithms and highlight the difference between regular optimal transport and subset selection, we consider a target sample of points from a circle centered at the origin and a source sample of points from a 2D uniform distribution also centered at the origin. We allow the scaling parameter $c$ to vary between 1 and 100, obtain the optimal transport plans $\boldsymbol{P}^*$ using both `SS-Entropic` and `SS-Bregman`, and evaluate the cost values $\langle\boldsymbol{P}^*, \boldsymbol{M}\rangle$. Results for this toy case are shown in Figure 2. It can be observed that as $c$ is increased the transport cost decreases until it saturates to the cost of the greedy solution $\sum_{i\in[m]}\frac{1}{m}\min_{j\in[n]}[\boldsymbol{M}]_{ij}$, which corresponds to $c = c^*$ where the transport map could be found by greedily choosing nearest source point for each target point as in equation 4. Figure 2 also illustrates the transport couplings for $c \in \{1, 1.25, 1.5, 1.75, 2, 4, 8, 16\}$. A key observation is that transport maps obtained with `SS-Bregman` are sparser as compared to the denser maps obtained using `SS-Entropic`. Additionally, they achieve smaller values of transport cost. Therefore in the subsequent sections, we focus

on results from `SS-Bregman` in the main body; results for Algorithm `SS-Entropic` are in the Appendix B and Appendix C, the latter includes comparisons with penalty-based semi-relaxed formulations.

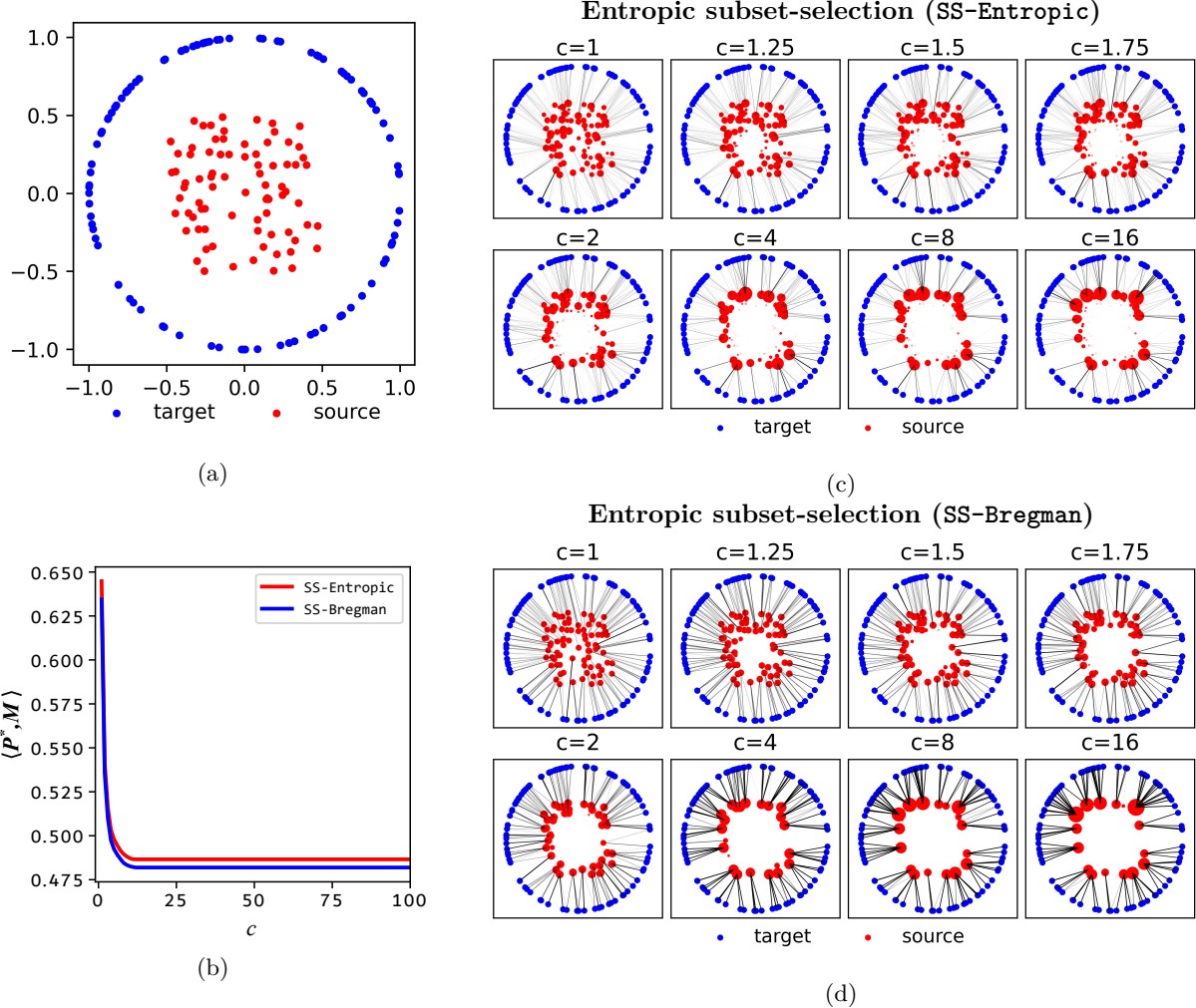

Figure 2: (a) The toy data generated by uniformly sampling $m = 100$ points from a circle centered at the origin with unit diameter as the target. The source contains $n = 80$ points generated by sampling uniformly from $[-\frac{1}{2}, \frac{1}{2}] \times [-\frac{1}{2}, \frac{1}{2}]$. (b) The optimal costs $\langle \boldsymbol{P}^*, \boldsymbol{M} \rangle$ obtained using `SS-Entropic` and `SS-Bregman` versus $c$ for $c \in [1, 100]$. `SS-Entropic` is ran for 10,000 iterations with $\gamma = 0.1$. `SS-Bregman` is ran for *max-outer-iter* = 100, *max-inner-iter* = 100 and $\lambda = 0.1$. (c) and (d) Support subset selection results obtained for $c \in \{1, 1.25, 1.5, 1.75, 2, 4, 8, 16\}$ using the Algorithms `SS-Entropic` and `SS-Bregman`, respectively.

**Fragmented Hypercube with Mode Dropping**: We demonstrate the utility of the support subset selection algorithm for partial point cloud registration on a toy case with one dropped mode and an affine transformation between the source and the target. Specifically, we consider data sampled from a uniform distribution over a hypercube (specifically, a square in 2D or a cube in 3D), which is then fragmented, where the target has one less fragment than the source. To generate the source we sample $n$ points $\{\boldsymbol{v}_i\}_{i=1}^n$ from the uniform distribution over a unit hypercube centered at the origin $[-\frac{1}{2}, \frac{1}{2}]^d$, $d \in \{2, 3\}$. These points are then fragmented into $2^d$ fragments according to their quadrant $\tilde{\boldsymbol{y}}_i = \boldsymbol{v}_i + (d-1)\operatorname{sign}(\boldsymbol{v}_i)$ and then offset to obtain the source points as $\boldsymbol{y}_i = \tilde{\boldsymbol{y}}_i + 5(d-1)$ for $i \in [n]$. The target data is generated similarly: a sample of $\hat{m} > m$ points $\{\boldsymbol{z}_i\}_{i=1}^{\hat{m}}$ is obtained from $[-\frac{1}{2}, \frac{1}{2}]^d$, then points with all negative coordinates are discarded, leaving $m$ points, which are fragmented into $2^d - 1$ fragments to obtain the target set $\{\boldsymbol{x}_i\}_{i=1}^m$ via $\boldsymbol{x}_i = \boldsymbol{z}_i + (d-1)\operatorname{sign}(\boldsymbol{z}_i)$ for $i \in [m]$. Examples of the data for 2D and 3D are shown in Figure 3(a) and Figure 4(a), respectively.

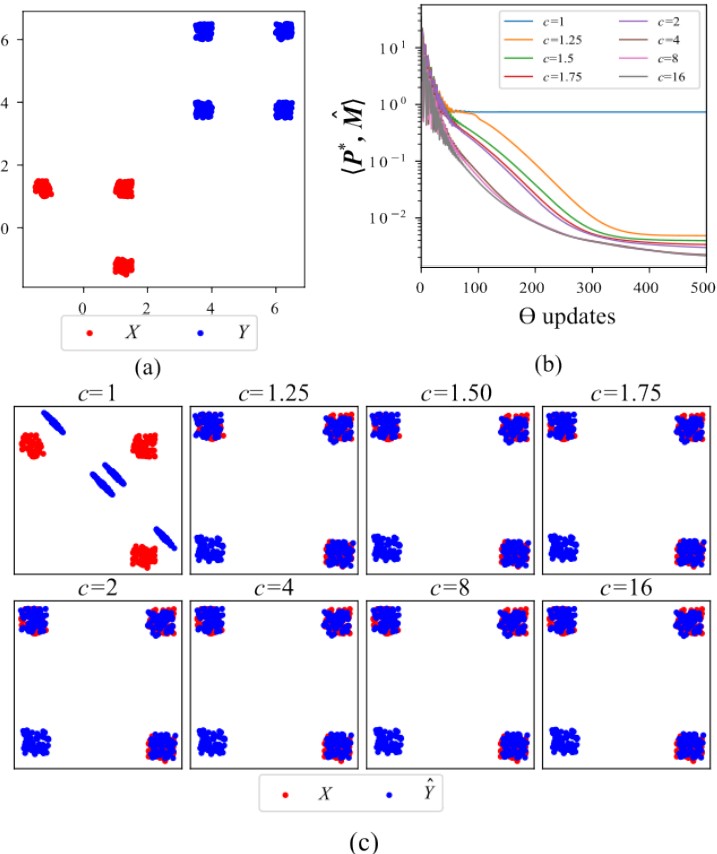

Figure 3: Results for affine transformation optimization with subset selection for partial optimal transport. Target points $X$ are sampled from a 2D fragmented hypercube centered at the origin with negative coordinates removed, whereas source points $Y$ are sampled from a translated fragmented hypercube. (a) Source and target sample points. (b) Loss function plotted against $\Theta = [A, b]$ updates for scaling parameter $c \in \{1, 1.25, 1.5, 1.75, 2, 4, 8, 16\}$. (c) Target and transformed source points after application of optimized affine transformation. Subset selection problems are solved using the $\texttt{SS-Bregman}$ with $\lambda = 0.1$, *max-outer-iter* = 100 and *max-inner-iter* = 500.

Due to the translation by $5(d-1)$ of the source point coordinates, direct application of the transport map will not yield a meaningful registration. Instead we use the bi-level optimization algorithm described in Section 2.5. The target and the transformed source after applying the affine transformation obtained using $\texttt{SS-Bregman}$ for $c \in \{1, 1.25, 1.5, 1.75, 2, 4, 8, 16\}$ are displayed in Figure 3(c) and Figure 4(c), respectively. Clearly, the $c = 1$ case corresponding to the complete optimal transport fails to identify a meaningful affine transformation, instead skewing and rotating the source fragments to minimize the Wasserstein distance to the target. The figures also display the cost $\langle P^*, \hat{M} \rangle$ across iterations. It can be observed that, like the previous toy examples as the value of scaling factor $c$ is increased, initially the value of the optimal loss $\langle P^*, \hat{M} \rangle$ decreases but after certain values of $c$, it saturates and stops decreasing and stays constant afterwards.

**Partial Point Cloud for 3D Shapes**: We further apply this form of subset selection based point cloud registration to point clouds for 3D objects when the target points are only taken from a portion of the entire 3D point cloud. Results for the Stanford bunny and armadillo (Turk & Levoy, 1994; Krishnamurthy & Levoy, 1996) are shown in Figure 5. It can be observed that for the case $c = 1$, which corresponds to complete optimal transport, the entire set of source points are coupled to the target point cloud which results in a distorted affine transform. For $c \in \{2, 5, 10, 20\}$, subset selection allows an appropriate subset of the source points to be well-fit by an affine transform to the target point cloud.

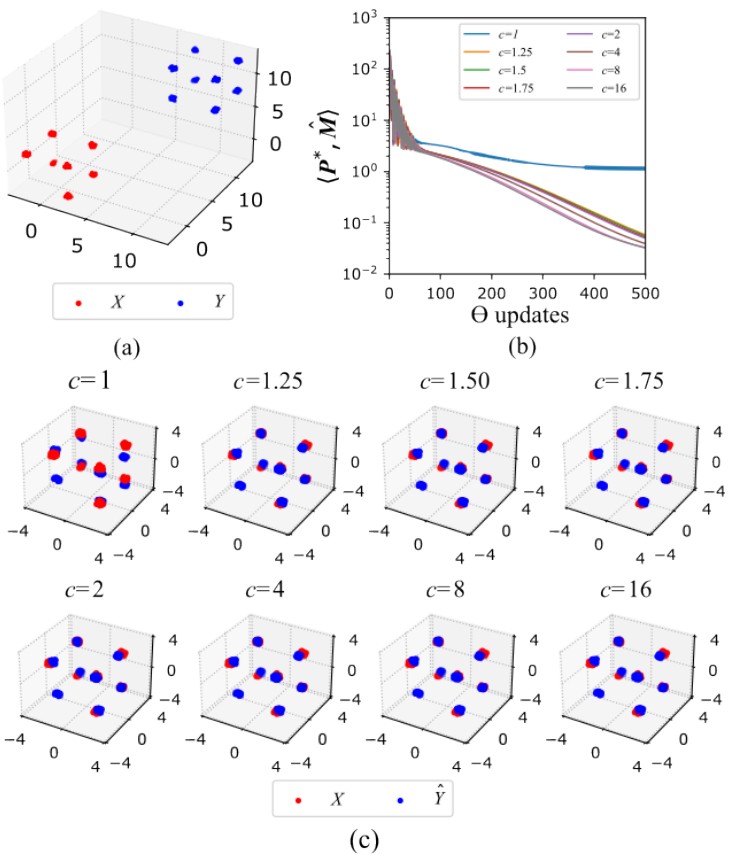

Figure 4: Results for affine transformation optimization with subset selection for partial optimal transport. Target points $\boldsymbol{X}$ are sampled from a 3D fragmented hypercube centered at the origin with negative co-ordinates removed, whereas source points $\boldsymbol{Y}$ are sampled from a translated fragmented hypercube. (a) Source and target sample points. (b) Loss function plotted against $\Theta = [\boldsymbol{A}, \boldsymbol{b}]$ updates for scaling parameter $c \in \{1, 1.25, 1.5, 1.75, 2, 4, 8, 16\}$. (c) Target and transformed source points after application of the optimized affine transformation. Subset selection problems are solved using the SS-Bregman with $\lambda = 0.1$, *max-outer-iter* $= 200$ and *max-inner-iter* $= 500$.

## 3.2 Color Transfer

Color transfer is the problem of finding a correspondence in the colors of pixels (represented as points in a 3D color space) between two images and then using this map to assign the colors of the source image to the target image (Reinhard et al., 2001). Color transfer is essentially an optimal transport problem in the color space, but with the added context that the pixels have their image coordinates, which are not used by the algorithm. For practical application to high resolution images, the pixel colors are first quantized using k-means clustering, as using partial optimal transport on the full set of pixel colors is computationally demanding. While in standard optimal transport the relative mass of each color cluster has to be preserved, here we exploit our formulation of partial optimal transport as support subset selection to allow a subset of colors to be used at a higher proportion than in the original source and allow a subset of colors to be completely discarded. For example, if a color cluster represents 1% of the original source's pixels, then it could represent up to $c\%$ of the target's pixels.

We apply k-means clustering to the set of vectors in RGB color space representing the source's $M$ pixels and the target's $N$ pixels separately to obtain $m \ll M$ color centroids $\{\boldsymbol{x}_i\}_{i=1}^m \subset \mathbb{R}_+^3$ for the target image and $n \ll N$ color centroids $\{\boldsymbol{y}_j\}_{j=1}^n \subset \mathbb{R}_+^3$ for the source image, with $\boldsymbol{\mu} \in \boldsymbol{\Delta}_m$ and $\boldsymbol{\mu} \in \boldsymbol{\Delta}_n$ being the vectors of proportion of colors in the target and source image color clusters, respectively. After that, we define the

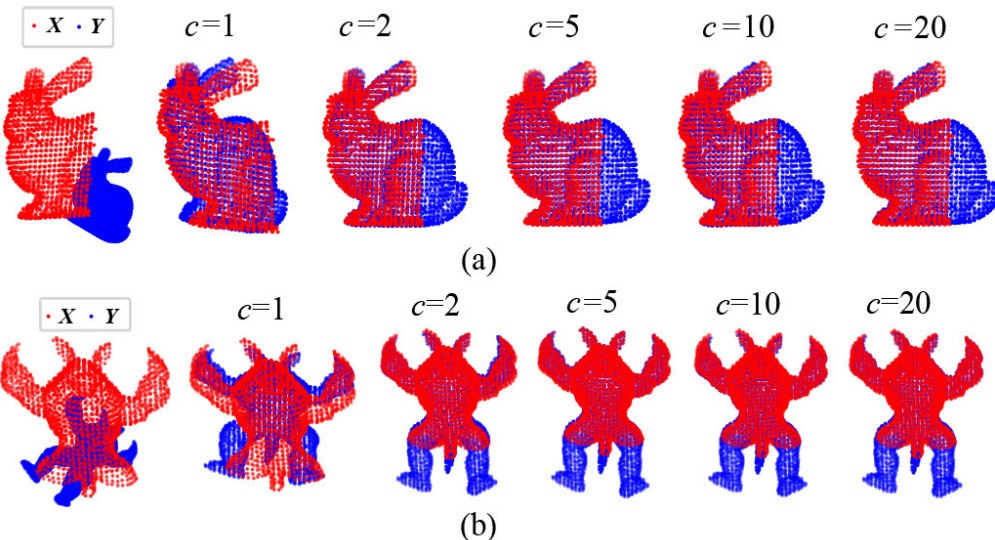

Figure 5: Affine transformation optimization for partial alignment of point clouds where a subset of the source point-cloud $\boldsymbol{Y}$ can be perfectly aligned (after rotation and scaling) with the target point cloud $\boldsymbol{X}$. We use the optimization algorithm described in Section 2.5, where SS-Bregman is employed to obtain the coupling $\boldsymbol{P}^*$ given the affine transformation parameters $\boldsymbol{A}$ and $\boldsymbol{b}$, which are updated using equation 27. (a) Stanford bunny point cloud. (b) Stanford armadillo point cloud.

cost matrix between the color centroids as $\boldsymbol{M}_{ij} = \|\boldsymbol{x}_i - \boldsymbol{y}_j\|_2^2, \forall i \in [m], j \in [n]$ and obtain the support subset selection map $\boldsymbol{P}^* \in \mathbb{R}_+^{m \times n}$ using SS-Bregman, such that $\boldsymbol{P}^* \mathbf{1}_n = \boldsymbol{\mu}$ and $\boldsymbol{P}^{*\top} \mathbf{1}_m \preccurlyeq c\boldsymbol{\nu}$. The support subset selection is then used to obtain the barycenter projections by solving (Blondel et al., 2018)

$$\hat{\boldsymbol{x}}_i = \underset{\boldsymbol{x} \in \mathbb{R}^3}{\arg\min} \sum_{j=1}^{n} P_{ij}^* \|\boldsymbol{x} - \boldsymbol{y}_j\|_2^2, \quad \forall i \in [m]. \tag{28}$$

The analytic solution of the barycenter projections can be compactly written as

$$\hat{\boldsymbol{X}} = (\boldsymbol{P}^* \oslash (\boldsymbol{\mu}\mathbf{1}_n^\top))\boldsymbol{Y} \in \mathbb{R}^{m \times 3}, \tag{29}$$

where $\boldsymbol{X} = [\boldsymbol{x}_1, \boldsymbol{x}_2, \ldots, \boldsymbol{x}_m]^\top \in \mathbb{R}^{m \times 3}$ and $\boldsymbol{Y} = [\boldsymbol{y}_1, \boldsymbol{y}_2, \ldots, \boldsymbol{y}_m]^\top \in \mathbb{R}^{n \times 3}$ are matrices of the color centroids. Each pixel in the target image is assigned the corresponding barycenter projection $\hat{\boldsymbol{x}}_{\pi(i)}$, where $\pi(i) \in [m]$ is the cluster assignment for the $i$th pixel of target image, $i \in [M]$.

We apply this color transfer scheme to images freely available though a Creative Commons licence, the "Louisiana Nature Scene Barataria Preserve" by Neil O as target and "Autumn in Toronto" by Bahman A-Mahmoodi as source. The color transfer results with $m = n = 128$ and SS-Bregman with $\lambda = 0.1$ are shown in Figure 6. It can be observed that the results for larger values of $c$ are smoother within objects or areas of similar color (e.g., the dark backdrop behind the peppers) and sharper in the color transitions between different objects (the colors in the orange versus red pepper on the right side of the photograph) as compared to the optimal transport case $c = 1$. This is due to the fact that larger values of $c$ allow certain colors to be reused more than their prevalence in the source image and allow some colors to be discarded, which enables smoother transitions in colors for areas of the target images with smooth color gradients. Similar observations can be seen in Figure 7 which uses the same settings and MATLAB test images: "peppers" as target and "corn" as source.

### 3.3 Subset Selection for Positive-Unlabeled Learning

In this section, we discuss the application of subset selection to the one-class semi-supervised classification scheme known as positive-unlabeled (PU) learning (Bekker & Davis, 2020). In PU learning, the training

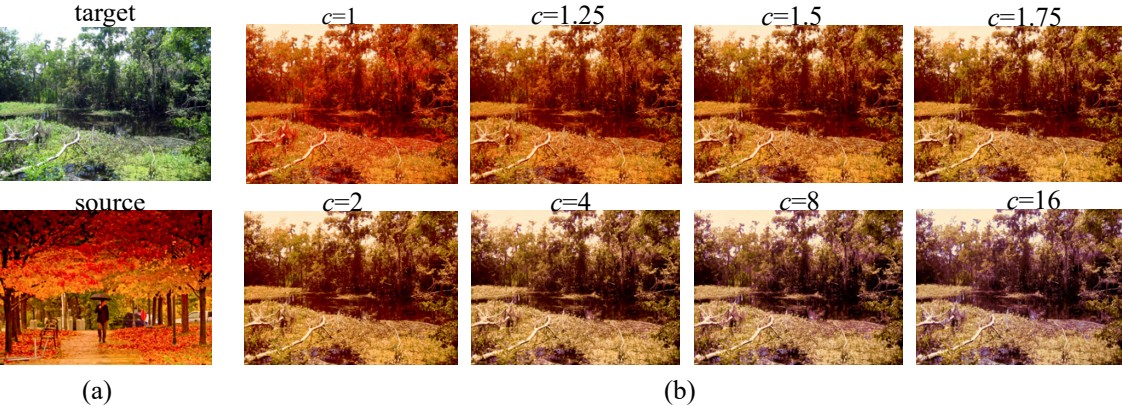

Figure 6: Color transfer results for $c \in \{1, 1.25, 1.5, 1.75, 2, 4, 8, 16\}$ for "Louisiana Nature Scene Barataria Preserve" by Neil O as target and "Autumn in Toronto" as source by Bahman A-Mahmoodi as target. The value of $c$ for each image is indicated at the top of image.

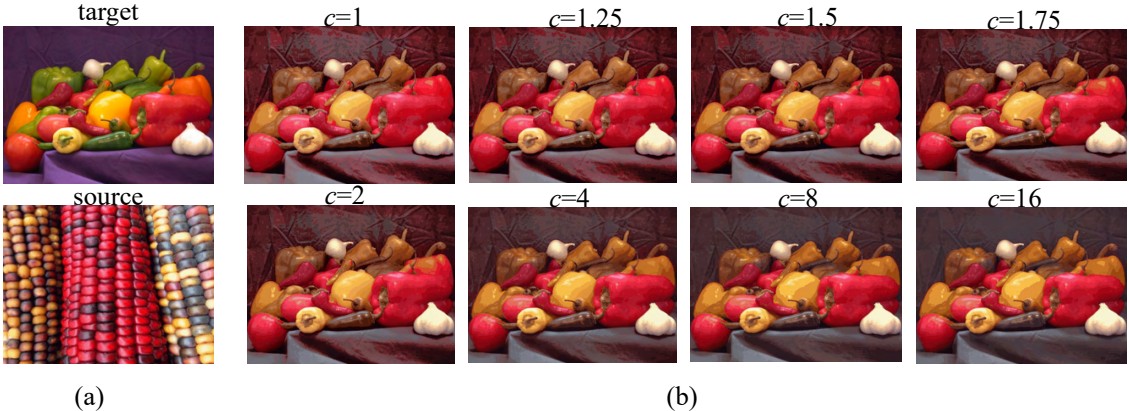

Figure 7: Color transfer results for $c \in \{1, 1.25, 1.5, 1.75, 2, 4, 8, 16\}$ for MATLAB image "peppers" as target and "corn" as source. The value of $c$ for each image is indicated at the top of image.

sample consists of purely positively labeled instances, and the unlabeled test sample consists of both positive and negative instances. Previous work often assumes that a prior on the probability of positive instances in unlabeled data is known (Kato et al., 2019; Hsieh et al., 2019; Chapel et al., 2020). Partial optimal transport is then used to find a subset with cardinality proportional to the prior of the test sample (source/unlabeled) that corresponds to all or a subset of the training sample (target/positive). We argue that all of the target mass should be preserved in cases of a relatively small and curated positive training sample. This motivates the application of our proposed subset selection approach to find the subset of the source that covers the positive target, compared to fully-relaxed approaches.

To illustrate the difference between fully and semi-relaxed partial optimal transport for PU learning, we consider two-dimensional random variables for the positive and unlabeled points where the support of positive random variable is a subset of the support of unlabeled random variable and both have long-tailed distributions, as shown in Figure 8. The lower accuracy for the fully-relaxed solution (79.5% versus subset selection's 84.1%) can be accounted for by the increased distances between source and target points away from the origin, which causes the target points to be dropped, preventing true positives in the source from being selected. This result holds for more general settings discussed in Appendix E.

We applied subset selection to PU learning using the experimental settings adapted from the work of Chapel et al. (2020), who explored using the PU-Wasserstein (PUW) 9 and the partial Gromov-Wasserstein distance (PGW) on various UCI, MNIST, colored-MNIST, and Caltech-office data sets. For the UCI, MNIST, and

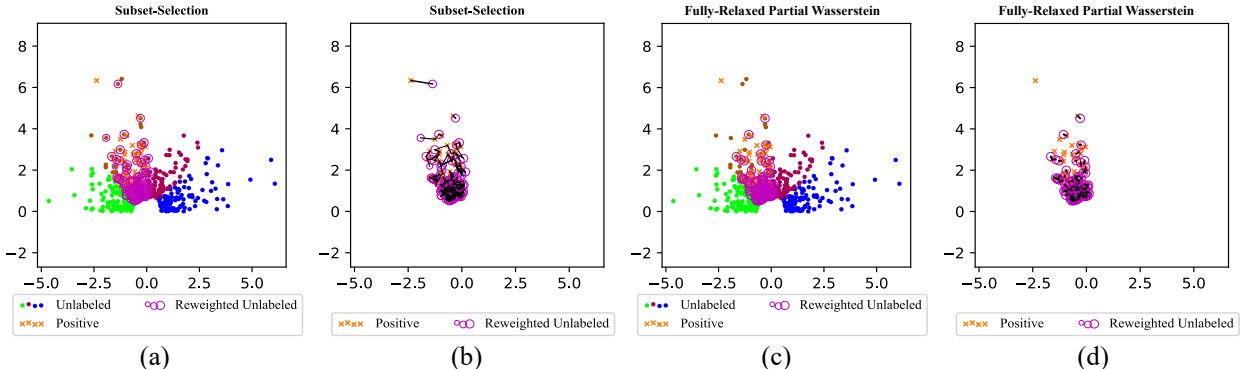

Figure 8: Results of PU learning on toy data using subset selection (accuracy 84.1%) and fully-relaxed partial optimal transport (accuracy 79.5%). Data is generated as $[r\cos(\psi),\ r\sin(\psi)]$ where the radius $r$ is drawn from a truncated exponential distribution with density $2\exp(-r)\imath_{[\log 2,\infty)}(r)$ and $\psi$ is uniform over a subset of angles $[0,\pi]$. The points belong to four classes corresponding to the angle falling in one of the intervals $[0,\ \frac{\pi}{4}]$, $(\frac{\pi}{4},\ \frac{\pi}{2}]$, $(\frac{\pi}{2},\ \frac{3\pi}{4}]$, or $(\frac{3\pi}{4},\ \pi]$. The $m=100$ target/positive points are all from the third class. The solutions are obtained using the known number of positives $n_+$ out of the $n=400$ source points, setting $c=\frac{n}{n_+}$ for our semi-relaxed approach and $s=\frac{n_+}{n}$ for the fully-relaxed partial optimal transport.

colored-MNIST data sets, we randomly draw $m=400$ positive points and $n=800$ unlabeled points. For Caltech-office data sets, we randomly sampled $m=100$ positive points from the first domain and $n=100$ unlabeled data points from the second domain. Following the experiments from the works by Chapel et al. (2020) and Kato et al. (2019), for multi-class data sets, we chose the data points from the class labeled 1 as positive and a random mixture of all classes as unlabeled, and the prior probability of positive class in the unlabeled set $\pi_+$ is set to be exactly the proportion of positives in unlabeled sample $\pi_+=\frac{n_+}{n}$, where $n_+$ is the number of true positives. This informs the PU-Wasserstein, partial Gromov-Wasserstein, and fully-relaxed partial Wasserstein optimal transport problems on the amount of mass to be transported as $s=\pi_+$, whereas for subset selection we set the scaling parameter to be $c=\frac{n}{n_+}$. Classification accuracy is evaluated by assigning positive predictions to the $n_+$ largest source mass assignments and negative predictions to the remaining source points. We also compute the ROC curve by using the source mass assignment $\boldsymbol{\nu}^*$ to rank the unlabeled source points. We ran the experiment 10 times and report the average of classification accuracy and the area under the ROC curve (ROC-AUC) in Table 1. It can be observed that the proposed subset-selection performs better than PU Wasserstein in terms of accuracy in 8 out of the 10 data sets with same domain (UCI, MNIST, and Caltech-office with same domains). Subset selection does best overall on 6 out of these 10, with fully-relaxed partial optimal transport performing better on 4 datasets. For the Caltech-office data sets with domain transfer, the partial Gromov-Wasserstein optimal transport does best. In terms of ROC-AUC, subset selection does better than PU Wasserstein in 9 out of the 10 intra-domain data sets, which is not surprising since the relative ranking is more meaningful than when the mass assignments are restricted to be binary valued $\{0,\frac{1}{n_+}\}$ as in the solutions from PU Wasserstein.

We attribute, higher AUC-ROC of subset selection as compared to fully-relaxed Wasserstein, to the fact that, subset selection problem has equality constraints on target mass which ensures coverage (lower false negatives). The differences between our solutions (SS) and those for the PU Wasserstein (PUW) are subtle, as both are semi-relaxed and maintain equality constraints on the target distribution. The small, but consistent, differences in the accuracy between our method and PUW may be due to the additional uniform mass assignment constraints in PUW, which is achieved through group-LASSO. The uniform mass assignment constraints in PUW may result into larger transport costs as compared to solutions obtained using SS with same cardinality. The largest mass assignments in the solution to SS might be more reliable than constrained solutions to PU Wasserstein.

| Dataset | $\pi_+$ | Accuracy | | | | ROC-AUC | | | |
|---|---|---|---|---|---|---|---|---|---|
| | | PUW | PGW | FR-PW | SS | PUW | PGW | FR-PW | SS |
| mushrooms | 0.518 | 95.15 | 94.85 | **99.45** | 96.63 | 0.9657 | 0.3336 | **0.9948** | 0.9883 |
| shuttle | 0.786 | 95.13 | 93.63 | **96.35** | 96.20 | 0.9321 | 0.6215 | 0.9467 | **0.9718** |
| pageblocks | 0.898 | 91.90 | 90.35 | 91.88 | **92.40** | 0.8036 | 0.7197 | 0.7817 | **0.8513** |
| usps | 0.167 | 98.28 | 95.55 | 97.08 | **98.48** | 0.9815 | 0.5096 | 0.9476 | **0.9927** |
| connect-4 | 0.658 | **60.95** | 58.05 | **60.95** | 60.73 | 0.5692 | 0.5126 | 0.5666 | **0.5871** |
| spambase | 0.394 | 78.80 | 68.40 | 69.08 | **79.28** | 0.7952 | 0.5834 | 0.6770 | **0.8369** |
| mnist | 0.1 | 99.08 | 98.23 | 99.15 | **99.18** | 0.9874 | 0.7638 | 0.9768 | **0.9971** |
| mnist-colored | 0.1 | 91.58 | 96.78 | **97.66** | 91.88 | 0.8189 | 0.6619 | 0.9360 | **0.9521** |
| surf C → surf C | 0.1 | 90.00 | 87.20 | 82.00 | **90.40** | **0.8576** | 0.4622 | 0.7469 | 0.7333 |
| surf C → surf A | 0.1 | 81.60 | **86.80** | 81.40 | 81.60 | 0.4546 | 0.4764 | **0.5337** | 0.4889 |
| surf C → surf W | 0.1 | 82.20 | **86.40** | 81.20 | 82.20 | 0.4707 | 0.4807 | 0.4451 | **0.5056** |
| surf C → surf D | 0.1 | 80.00 | **87.00** | 80.00 | 80.00 | 0.3756 | 0.4328 | 0.4056 | **0.4444** |
| decaf C → decaf C | 0.1 | 94.00 | 86.20 | 82.00 | **94.40** | 0.9498 | 0.5713 | 0.7667 | **0.9682** |
| decaf C → decaf A | 0.1 | 80.20 | **88.20** | 81.80 | 80.40 | 0.3986 | 0.5031 | **0.5349** | 0.4564 |
| decaf C → decaf W | 0.1 | 80.20 | **88.60** | 82.00 | 80.00 | 0.4299 | 0.5827 | 0.5611 | **0.5965** |
| decaf C → decaf D | 0.1 | 80.80 | **92.20** | 80.40 | 80.40 | 0.4546 | **0.5042** | 0.4617 | 0.4530 |

Table 1: PU learning on data sets as in the work by (Chapel et al., 2020). For subset selection (SS) and fully-relaxed partial-Wasserstein (FR-PW), accuracy is evaluated by assigning label 1 to $n_+$ largest mass assignments and label 0 to the remaining mass assignments. For PU Wasserstein (PUW), the mass assignment are constrained to be binary valued in the set $\{0, p\}$, the data points with mass assignments 0 are labeled 0 and the data points with mass $p = \frac{1}{n\pi_+} = \frac{1}{n_+}$ are labeled 1.

### 3.3.1 PU Learning on MNIST/EMNIST

To further illustrate how `SS-Bregman` operates on PU learning, we apply it to the case where the positive training sample (target) consists of MNIST digit images and the unlabeled test sample contains 50% points (MNIST digits) and 50% negative points (alphabetic letters from EMNIST). When $c = 1$, which is equivalent to standard optimal transport, initially all the images in the unlabeled source sample are assigned uniform masses. As $c$ is increased, we hypothesize that the true positive MNIST digits will been assigned larger mass and remain in the selected support, whereas the EMNIST letters will receive relatively lower or zero mass. Our hypothesis is confirmed by the results displayed in Figure 9(a), which displays the ROC curve across different choices of $c$, and in Figure 9(c), which displays the area under the ROC curve (AUC). As $c$ is increased, source points with largest mass assignments are mostly MNIST digits. Likewise, Figure 9(e) shows the images with highest mass for different values of $c$ which are mainly MNIST numbers or EMNIST letters with close resemblance to a number. Figure 9(b) visualizes the distribution of source point masses by graphing the sorted masses for different values of $c$. From these curves the cardinality of the subset is easily seen for different values of $c$. Notably, for values of $c \leq 4$ there are exists a subset of the selected source points with uniform mass, but for larger values of $c$, the mass is non-uniform across all instances. These changes correspond to the change in slope of the entropy of the distribution for different values of $c$ is displayed in Figure 9(d).

We further compared our approach for PU learning with semi-relaxed optimal transport approaches using the the squared $\ell_2$-norm penalty Chapel et al. (2021) and total-variation (TV) divergence Séjourné et al. (2019). The formulation of semi-relaxed problems is discussed in Appendix C. We used POT-toolbox Flamary et al. (2021) to solve the semi-relaxed optimal transport problems. For both squared $\ell_2$-norm and total-variation penalties, we varied regularization parameter $\rho$ with 32 uniform steps on logarithmic scale between $10^{-3}$ and $10^3$. For subset-selection, $c$ parameter is varied uniformly on logarithmic scale with 32 steps between 1 and 32. In order to evaluate the performance of each method, we assigned label 1 to all the selected points and zero to all the remaining points. We also computed the cardinalities and entropies of mass-assignments $\boldsymbol{\nu}^*$. In Figure 10 we compare the effect of the cardinality of the support of the mass assignment vector $\boldsymbol{\nu}^*$ on the accuracy and entropy $H(\boldsymbol{\nu}^*)$. We observe that for the mass assignments with same cardinality,

mass assignments obtained through subset selection have higher entropy as compared to both TV and $\ell_2$ penalized semi-relaxed optimal transport. For each regularization discussed above, mass-assignment paths across scaling parameter $c$ for `SS-Bregman` are discussed in the Appendix C.

We also adapt the approach proposed in the work by Phatak et al. (2023) in the context of fully-relaxed partial optimal transport to automatically select the proportion of mass to separate inliers and outliers to automatically find a choice of $c$ for subset selection for PU learning. The approach finds the knee of the smoothed version of the first derivative of $\frac{1}{c}\mathcal{S}_p(\mu, c\nu)$ as a function of $\frac{1}{c} \in (0, 1]$, using the kneedle method (Satopaa et al., 2011). The results in Figure 10(c) show that the automatically selected value of $c$ is at the highest accuracy.

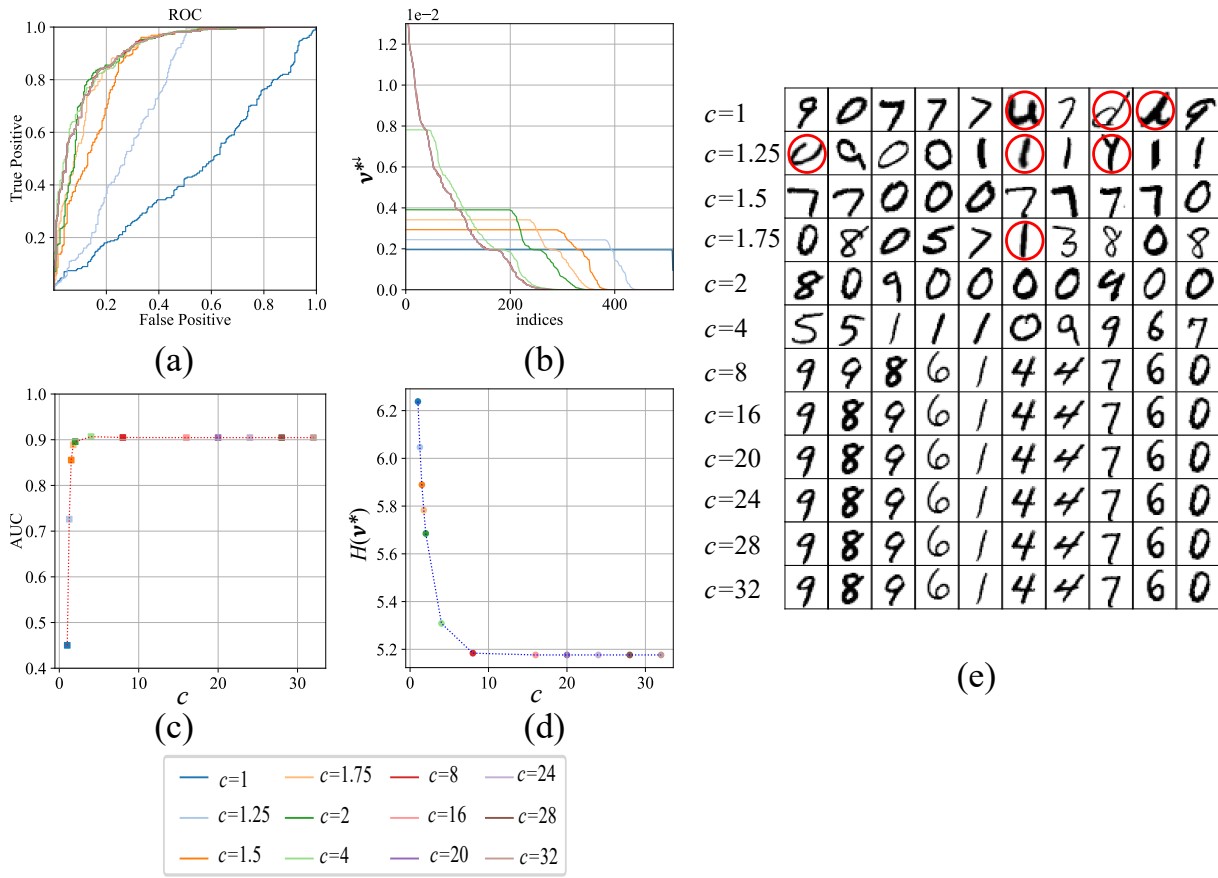

Figure 9: Subset selection results obtained using `SS-Bregman` with parameters $\lambda = 1$, *max-outer-iter* $= 250$ and *max-inner-iter* $= 20$. Source sample $n = 512$ consists of 50% digit images from MNIST and 50% letter images from EMNIST. The target sample contains $m = 512$ digit images drawn from MNIST. (a) ROC curves for different values for $c \in \{1, 1.25, 1.50, 1.75, 2, 4, 8, 16, 20, 24, 28, 32\}$. (b) Mass assignments to source images in descending order $\boldsymbol{\nu}^{*\downarrow}$. (c) AUC of ROC versus $c$. (d) Entropy $H(\boldsymbol{\nu}^*)$ of the mass assignments $\boldsymbol{\nu}^*$ versus $c$.

### 3.3.2 PU Learning for CIFAR-10 Neural-Network Representations

We now consider the proposed subset selection algorithm for PU learning on the CIFAR-10 data set, where a single class from the training set is treated as the positive target and a mixture of all classes from the test set is the unlabeled source. Fundamentally, the performance of optimal transport methods on PU learning depends on the distance metric defining the cost matrix. Thus, the method performs poorly if a Euclidean distance metric is applied to complex data such as natural images. Instead, a learning representation extracted from

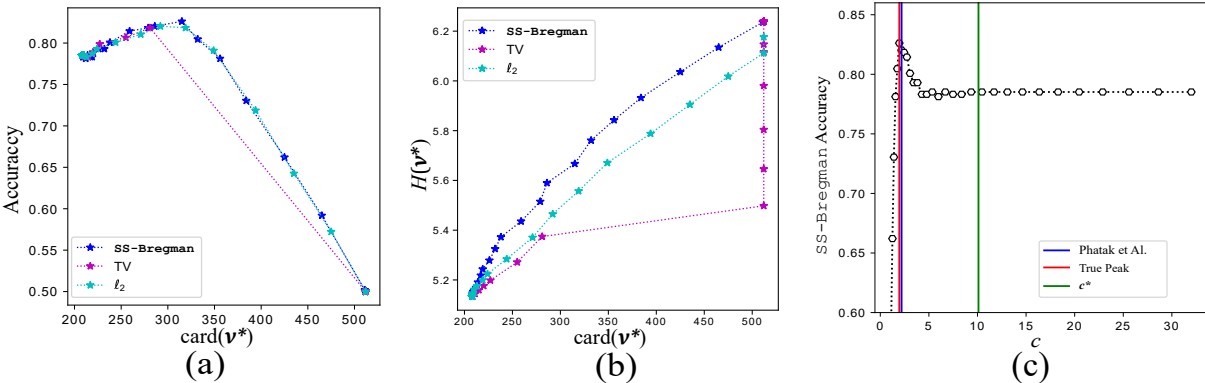

Figure 10: PU learning with semi-relaxed optimal transport approaches on MNIST/EMNIST. Solutions for different cardinalities are obtained by varying the regularization/constraint parameters across a grid, uniform on a logarithmic scale. (a) Accuracy versus cardinality of selected subset using `SS-Bregman`, semi-relaxed optimal transport with total variation (TV) and squared-$\ell_2$ penalties for PU learning on MNIST/EMNIST. (b) Entropy of mass assignments versus cardinality of selected subset using `SS-Bregman`, semi-relaxed optimal transport with total variation (TV) and squared $\ell_2$ penalties. (c) Accuracy of PU learning on MNIST/EMNIST using `SS-Bregman` versus scaling parameter $c$. Vertical lines indicate the location of the knee obtained using Phatak et al. (2023), the true peak accuracy value of $c$, and the break point $c^*$.

a pretrained neural network can be used. Here each image is represented as the vector of activations of the penultimate layer of the pre-trained ResNet-20 classifier (trained on CIFAR-10), and the Euclidean distance between the activation vectors defines the cost matrix for the transport problem. The results are given in Figure 11. The results are similar to the previous MNIST/EMNIST data set. Mass is uniformly distributed across a subset of images for values of $1 < c < 8$. When the subset is greater than the proportion of positive instances in the unlabeled source, then the relative ranking of mass is not reliable: the top instances for $c \in \{1.5, 2, 4\}$ are images from the target class, but $c \in \{1.25, 1.75\}$ have images resembling it from other classes. As $c$ is increased above $c = 8$ the mass assignment is non-uniform, but constant for further increment in $c$. Values of $c$ greater than 2 have an AUC $>90\%$.

Similar to PU learning on MNIST/EMNIST, we also compared our approach for PU learning on neural network representations of CIFAR-10 with TV and $\ell_2$ penalized semi-relaxed optimal transport. For semi-relaxed formulations ($\ell_2$ and TV penalties) we varied $\rho$ on logarithmic scale between $10^{-4}$ and $10^3$ with 32 steps. For subset selection we varied the scaling parameter $c$ logarithmically between 1 and 60 in 32 steps. We adopted the same strategy as previous case (PU learning on MNIST/EMNIST) to evaluate accuracy, cardinality, and entropy of mass assignments $\nu^*$ at different values regularization and scaling parameters. Figure 12 shows accuracy versus cardinality, and entropy versus cardinality, along with variation of PU learning accuracy across $c$ and knee-based scaling parameter selection. Our observations in this case are similar to the case for EMNIST/EMNIST: subset selection has the highest entropy at a given cardinality, its accuracy is at or above the other solutions, and the knee method selects close to optimal value of $c$.

## 3.4 Subset Selection for Semi-supervised Learning

We consider the semi-supervised training of a classifier where the training set is divided into a reliably labeled (curated) target set and an unlabeled or noisily labeled source set. We apply our proposed subset selection algorithm to perform partial optimal transport of the unlabeled source to the labeled target. The transport plan is computed without knowledge of any labels but defines how the source points will be labeled, and subset selection removes points that cannot easily be aligned to labeled training points. Additionally, the new mass assignment source points may be relatively higher for unlabeled points relatively close to labeled training points and lower for unlabeled points from existing points. Used in this way, the optimal transport with subset selection automatically tunes how far to propagate labels in a manner that takes

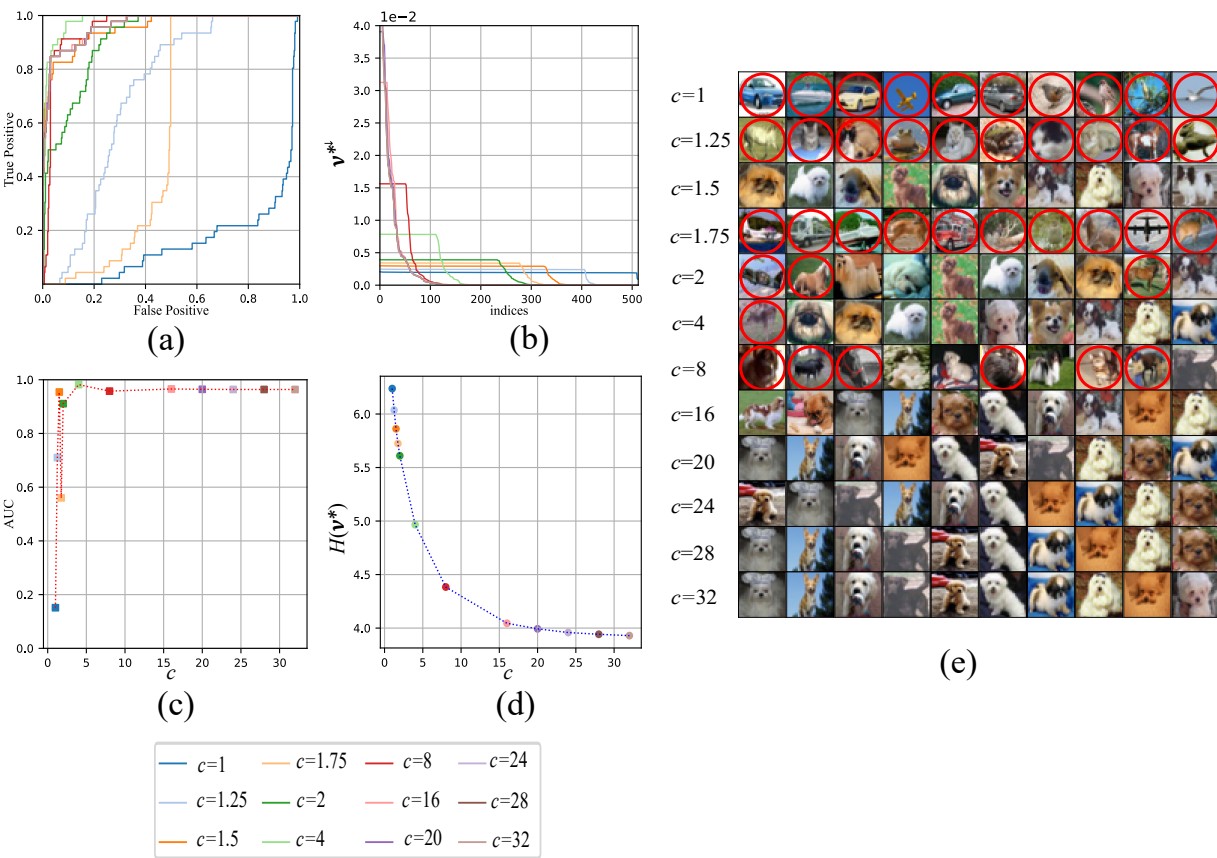

Figure 11: Subset selection results obtained using algorithm `SS-Bregman` with parameters $\lambda = 1$, *max-outer-iter* $= 250$ and *max-inner-iter* $= 20$. Target and source consist of ResNet-20 embeddings of $m = 512$ CIFAR-10 dog images and $n = 512$ randomly sampled CIFAR-10 images, respectively. (a) ROC curves for $c \in \{1, 1.25, 1.50, 1.75, 2, 4, 8, 16, 20, 24, 28, 32\}$. (b) Mass assignments to source images in descending order $\boldsymbol{\nu}^{*\downarrow}$. (c) AUC of the ROC versus $c$. (d) Entropy $H(\boldsymbol{\nu}^*)$ of mass assignments $\boldsymbol{\nu}^*$ with versus $c$. (e) Source images with 10 largest mass assignments.

into consideration the geometry and distribution of the curated target data set rather than only the local distances.

However, using the distances defined directly in the input space may not be suitable, and a pre-trained representation may not exist for various tasks. Instead, we propose to use the internal learning representation from the neural network classifier while it is being optimized with the semi-supervised loss function.

Let $\mathcal{S} = \{(\boldsymbol{x}_i, \boldsymbol{L}_i)\}_{i=1}^M$ denote the labeled portion of the training set with the input $\boldsymbol{x}_i \in \mathcal{X}$ and label encoded as a one-hot vector $\boldsymbol{L}_i \in \{0, 1\}^k \subset \boldsymbol{\Delta}_k, \|\boldsymbol{L}_i\|_1 = 1$ for $i \in [M]$, and $\mathcal{T} = \{\boldsymbol{y}_j\}_{j=1}^N$ denote the unlabeled portion, $\boldsymbol{y}_j \in \mathcal{X}$ for $j \in [N]$. We consider a neural-network classifier with soft-max activation $\boldsymbol{f}(\cdot\,;\boldsymbol{\theta}) : \mathcal{X} \to \boldsymbol{\Delta}_k$ with parameters $\boldsymbol{\theta}$ trained on data with $k$ classes. The neural network's internal representation is a function $\boldsymbol{g}(\cdot\,;\boldsymbol{\theta}) : \mathcal{X} \to \mathbb{R}^d$. The Euclidean distance between the internal representation of data points provides the distance function, $\mathrm{d}_{\boldsymbol{\theta}}(\boldsymbol{x}_i, \boldsymbol{y}_j) = \|\boldsymbol{g}(\boldsymbol{x}_i; \boldsymbol{\theta}) - \boldsymbol{g}(\boldsymbol{y}_j; \boldsymbol{\theta})\|_2$, which is parameterized by the network's parameters.

We train the neural network using mini-batches and a semi-supervised cross-entropy loss. Equal-sized batches are drawn uniformly from the pooled training data set of size $M + N$. Let $\boldsymbol{\tau}$ and $\boldsymbol{\sigma}$ denote the length-$m$ and length-$n$ vectors of indices of the labeled and unlabeled points in a given batch, respectively, where $m + n$ is the constant batch size. The $m$-by-$n$ ground cost matrix $\boldsymbol{M}(\boldsymbol{\theta})$ is defined using the squared distances among the batch's latent representations, $M_{ij}(\boldsymbol{\theta}) = \mathrm{d}_{\boldsymbol{\theta}}^2(\boldsymbol{x}_{\tau_i}, \boldsymbol{y}_{\sigma_j}) = \|\boldsymbol{g}(\boldsymbol{x}_{\tau_i}; \boldsymbol{\theta}) - \boldsymbol{g}(\boldsymbol{y}_{\sigma_j}; \boldsymbol{\theta})\|_2^2$.

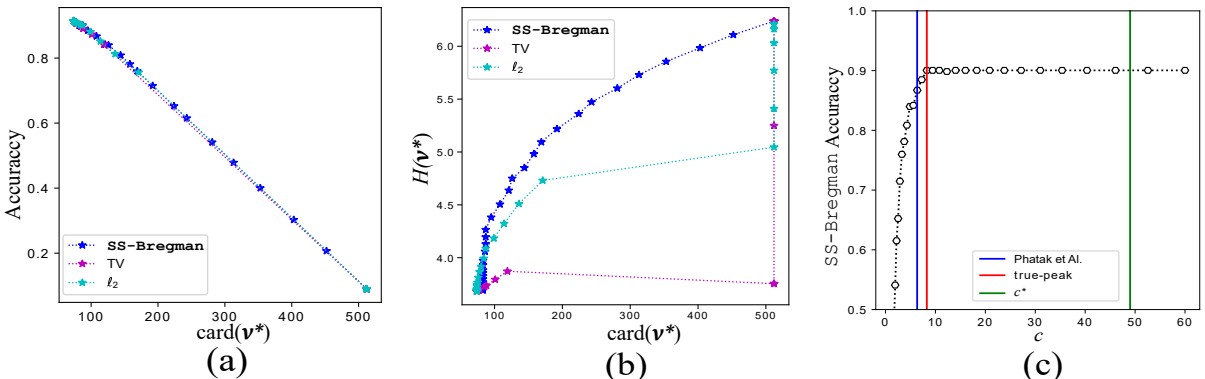

Figure 12: PU learning with semi-relaxed optimal transport approaches on CIFAR-10. Solutions for different cardinalities are obtained by varying the regularization/constraint parameters across a grid, uniform on a logarithmic scale. (a) Accuracy versus cardinality of selected subset using `SS-Bregman`, semi-relaxed optimal transport with total variation (TV), and squared $\ell_2$ for PU learning on CIFAR-10. (b) Entropy of mass assignments versus cardinality. (c) Accuracy of PU learning on CIFAR-10 using `SS-Bregman` versus scaling parameter $c$. Vertical lines indicate knee obtained using Phatak et al. (2023), true peak accuracy value of $c$, and break point $c^*$.

Given the cost matrix and hyper-parameters (including $c \leq n$), the subset selection transport plan $\boldsymbol{P}^* \in [0,1]^{m \times n}$ is obtained using `SS-Bregman`. Given the matrix of one-hot encoded labels $\boldsymbol{L} = [\boldsymbol{L}_{\tau_1}, \ldots, \boldsymbol{L}_{\tau_m}]^\top \in \{0,1\}^{m \times k}$, the matrix of pseudo-labels assigned by the algorithm of the unlabeled mini-batch points is computed $\tilde{\boldsymbol{L}} = n\boldsymbol{P}^{*\top}\boldsymbol{L} \in [0,c]^{n \times k}$, where $[\boldsymbol{P}^{*\top}\boldsymbol{L}]_{jl} = \frac{1}{n}\tilde{L}_{jl} \in [0,1]$ is the estimate of the joint probability that mini-batch unlabeled instance $j \in [n]$ belongs to class $l \in [k]$.[3] Given the pseudo-labels, the semi-supervised cross-entropy loss function for a batch is

$$\text{loss}(\boldsymbol{\theta}) = -\left[\sum_{i=1}^{m}\sum_{l=1}^{k}\frac{1}{m}\boldsymbol{L}_{il}\log(f_l(\boldsymbol{x}_{\tau_i};\boldsymbol{\theta})) + \sum_{j=1}^{n}\sum_{l=1}^{k}\frac{1}{n}\tilde{\boldsymbol{L}}_{jl}\log(f_l(\boldsymbol{y}_{\sigma_j};\boldsymbol{\theta}))\right]. \tag{30}$$

Our approach is similar to other recent work (Damodaran et al., 2020) that also employs optimal transport using a learning representation. While we address the semi-supervised case, Damodaran et al. (2020) address supervised learning in the presence of label noise and perform self optimal transport within batches to correct for label noise.

As baseline comparisons, we compare our semi-supervised approach to supervised training with either only the labeled portion or with noisy labels on the unlabeled portion. Due to the curated labeled set, the latter is not the typical label noise scenario; however, the division of a training set into a curated portion and a portion with label noise is relevant to practical scenarios. While our semi-supervised approach does not use noisy labels, future extensions could consider how to leverage the noisy labels too.

In order to evaluate our approach we used MNIST, Fashion-MNIST (FMNIST), and CIFAR-10. We split training data sets into 80/20 proportions for training and validation. We further split the training part into a reliably labeled and unreliably labeled parts. Labels for the unreliably labeled part are generated by uniformly corrupting the true labels to other classes depending on the noise level. For each of our experiments, the subset selection transport underlying the loss done is found via `SS-Bregman` with $\lambda = 0.1$, *max-outer-iter = max-inner-iter = 20* with a batch-size of 512. We used PyTorch framework for our experiments. A ResNet-18 model architecture is used on the CIFAR-10 data set. We trained the ResNet-18 for 180 epochs using Adam optimizer with an initial learning rate of 0.001, which is scheduled to be halved

---

[3]It can be seen that the total sum of this joint is 1, $\underbrace{\mathbf{1}_n^\top \boldsymbol{P}^{*\top}}_{\boldsymbol{\mu}^\top}\underbrace{\boldsymbol{L}\mathbf{1}_k}_{\mathbf{1}_m} = 1$.

after every 60 epochs. The model architectures containing two convolutional layers for MNIST and FMNIST are given in Appendix F. The neural network classification models for MNIST are trained using stochastic gradient descent with a learning rate of 0.001, whereas models for FMNIST are trained using Adam with a learning rate of 0.001 and weight decay 1e-4. Model training for MNIST and Fashion-MNIST are done on a desktop system containing Intel Core-i7 9700 CPU, with 32 GB memory and NVIDIA GeForce RTX 2070 GPU. ResNet-18 based models for CIFAR-10 are trained using Lambda-labs cloud resources with 30 vCPUs, 200 GB memory, and NVIDIA A10 GPUs.

| Dataset | Architecture | stand. | Subset Selection | | | | | | | | |
|---|---|---|---|---|---|---|---|---|---|---|---|
| | | | $c=1$ | $c=2$ | $c=3$ | $c=4$ | $c=5$ | $c=6$ | $c=7$ | $c=8$ | $c=20$ |
| **MNIST** | 2-layer conv-net | 96.57 | 97.15 | 97.39 | 97.33 | 97.40 | 97.43 | 97.32 | 97.38 | **97.44** | 97.35 |
| **FMNIST** | 2-layer conv-net | 88.68 | 90.14 | 90.30 | 90.27 | **90.51** | 90.23 | 90.38 | 90.21 | 90.37 | 90.16 |
| **CIFAR-10** | ResNet-18 | 79.10 | 87.18 | 89.45 | 89.27 | 89.21 | 89.53 | 89.54 | **89.72** | 89.47 | 89.57 |

Table 2: Validation accuracies for different values for neural network classification models trained with 50% reliably labeled points and 50% points with noisy labels (noise level 0.8). Standard training (stand.) treats them equally, but subset selection treats them as unlabeled and assign pseudo-labels. Subset selection is done using the `SS-Bregman` with $\lambda = 0.1$ and *max-outer-iter = max-inner-iter = 20*.

In the first step of experiments, we split the training set for each data set into 50/50 proportions for unreliably and unreliably labeled parts. Unreliably labeled data is generated by uniformly corrupting the labels with a 80% chance (noise level 0.8). Validation accuracies for each data set are displayed in the Table 2. Notably, the performance for $c = 1$ is higher than training with noisy labels, which shows that the semi-supervised training performs better than training with data with a high noise level. (Because the algorithm is not run to convergence, the mass assignments for unlabeled points may not be be exactly uniform in the $c = 1$ case.) The performance of subset selection is consistently higher for values of $c > 1$ compared to the $c = 1$, and the validation accuracies do not exhibit much change between $c = 2$ and $c = 20$. Therefore, we further evaluated our approach by varying both noise levels and clean and noisy proportions only for $c = 2$ and $c = 20$.

Progress of validation accuracies on CIFAR-10 are displayed in Figure 13 for clean/noisy proportions in $\{20/80, 40/60, 60/40, 80/20\}$ with noise levels $\{0.2, 0.4, 0.6, 0.8\}$. It can be observed that standard neural network training process with label noise can divided into three phases, first in which the validation accuracy increases until a peak. In the second phase, validation accuracy decreases, where the magnitude of the decrement depends on the noise level: it decreases less for low noise levels and more for higher noise levels. In the third phase, validation accuracy increases again and then oscillates around a constant value. This kind of phenomenon is more pronounced for larger noise levels (Zheng et al., 2020). In contrast, for the proposed subset selection based semi-supervised learning the validation accuracy does not go down after hitting its peak during the training process. This indicates that the transport map tend to assign correct pseudo-labels to the data points nearest the labeled data points and does not introduce label noise.

The test set accuracies are displayed in Table 3 for supervised training on only the labeled data versus the semi-supervised training. The semi-supervised training with subset selection outperforms training performs better on 2 of 3 data sets under a 20/80 split of labeled and unlabeled, but does not outperform supervised learning for the 40/60 split. Thus, the semi-supervised loss function equation 30 is most beneficial when there is a higher ratio of unlabeled to labeled points.

## 4 Discussion and Further Work

In this paper, we have focused on selecting a subset of one distribution's support as a special case of partial optimal transport Figalli (2010); Chapel et al. (2020). This is useful to find meaningful alignment when the support of the target distribution is assumed to be a subset of the source distribution. Results on the partial point cloud alignment, color transfer, PU learning, and semi-supervised learning all demonstrate the utility of this approach.

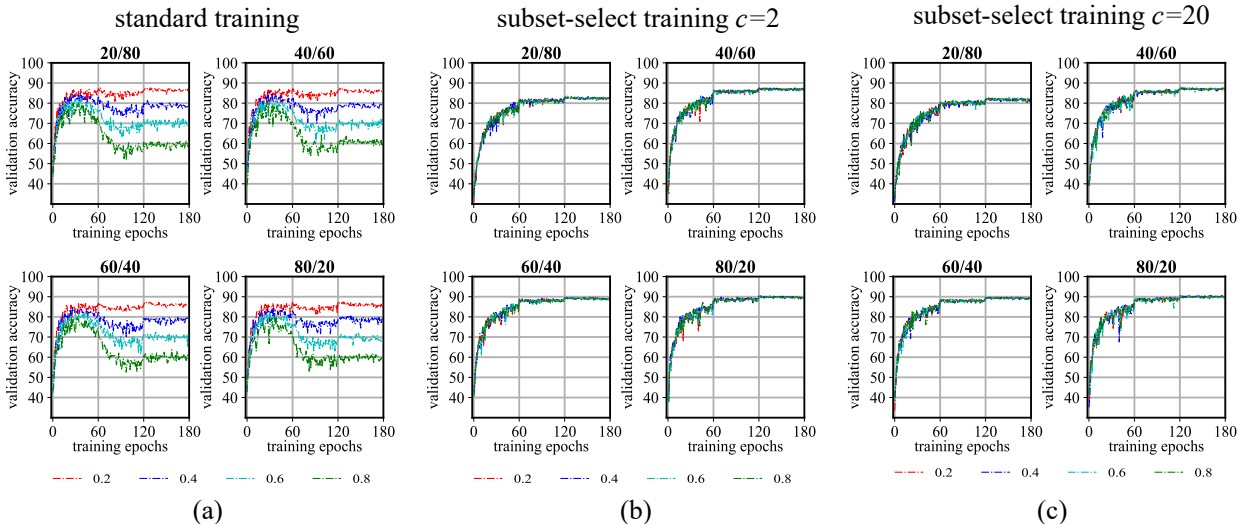

Figure 13: Progress of validation accuracies while training ResNet-18 for CIFAR-10 classification. (a) Uniform noise levels are varied between $0.2, 0.4, 0.6, 0.8$ for standard training. (b) and (c) Training with the subset selection based semi-supervised loss at different values of $c$, does not use the unreliable labels, and outperforms standard training with noisy labels when either the proportion of reliably labeled data is 40% or the noise level is 0.4 or greater. Subset selection is done using the `SS-Bregman` with $c = 20$, $\lambda = 0.1$, *max-outer-iter = max-inner-iter = 20*.

| Dataset | Architecture | Labeled/Unlabeled % | Labeled only | Semi-supervised with subset selection |
|---------|--------------|---------------------|--------------|----------------------------------------|
| **MNIST** | 2-layer conv-net | 20/80 | **98.19** | 96.09 |
| | | 40/60 | **98.30** | 97.14 |
| **F-MNIST** | 2-layer conv-net | 20/80 | 88.35 | **88.42** |
| | | 40/60 | **89.93** | 89.48 |
| **CIFAR-10** | ResNet-18 | 20/80 | 81.62 | **82.37** |
| | | 40/60 | **87.77** | 86.74 |

Table 3: Semi-supervised learning test accuracies on MNIST, Fashion-MNIST, and CIFAR-10. Subset selection is done using `SS-Bregman` with $c = 20$, $\lambda = 0.1$, *max-outer-iter = max-inner-iter = 20*.

In particular, the results from the PU learning show that the proposed subset selection is useful when there is known target distribution (an existing training or validation set) and an additional source distribution, which has additional diversity, but also outliers, compared to the target. One application of PU learning is to filter a source of new data for relevant examples for further modeling. Future work could explore subset selection approach for source distributions created from synthetic generation mechanisms. While not explored in a machine learning context, it is possible that the partial optimal transport with affine (or nonlinear) transformation can be applied to account for global covariate shift between the synthetic and real data. In this case, a user would want to balance the diversity (entropy) of the filtered source with its purity.

Interestingly, the solution for the subset selection are similar but consistently outperform those from PU Wasserstein Chapel et al. (2020), which by design of the constraints, have a maximum entropy, uniform distribution over the selected subset achieved through the group LASSO penalty. The results from subset selection show that it maintains close to maximum entropy amongst the selected support and is as accurate as other semi-relaxed penalty based approaches. Additionally, the manual choice of $c$ controlling the constraint is more intuitive than selecting a penalty parameter. Finally, the automatic selection of $c$ using the straightforward knee-based selection adapted from fully-relaxed partial optimal transport (Phatak et al., 2023) shows promising results to separate inliers and outliers.

In our experiments related to semi-supervised learning, we employed optimal transport between a labeled target and unlabeled source, to assign pseudo-labels to source points that cover the labeled data distribution, while ignoring ambiguous cases, during training. In future extension, we can consider how to use class information in the optimal transport planning, perhaps by using class conditional optimal transport, as currently the transport plan is not informed of the known target labels nor the classifier's boundaries. Another line of exploration is how to use the support subset selection to correct noisily labeled source.

Another key contribution of this work is the proposed support subset selection algorithm using the inexact Bregman proximal point algorithm (`SS-Bregman`), which as shown in Appendix B yields a solution with a sparse source marginal similar to solutions to the original linear program 6—unlike the entropically regularized solution from `SS-Entropic`. We also demonstrate that the mass assignments of the linear program solution are piece-wise linear as a function of $c$. While not fully investigated here, this behavior could be exploited to find the sequence of breakpoints where points leave the support and where points leave the active set of constraints (indicated by being on the upper diagonal).

Recently, Gromov-Wasserstein optimal transport has seen applications in graph-matching and generative modeling (Brogat-Motte et al., 2022; Li et al., 2023; Nekrashevich et al., 2023; Bunne et al., 2019; Mémoli, 2009). Due to inherent ability to match structural correspondences across spaces, partial Gromov-Wasserstein optimal transport can be used to solved robust graph-alignment problems. Recently, efficient locally convergent solutions for a relaxed Gromov-Wasserstein distance have been proposed (Peyré et al., 2016; Li et al., 2023). Future work can explore the subset selection case of the partial Gromov-Wasserstein optimal transport, where one domain is expected to have a complete or overcomplete source distribution compared to the target. This may be useful in robust domain adaptation, semi-supervised domain adaptation, and metric alignment.

### Author Contributions

Bilal Riaz and Austin Brockmeier worked on the formulation and design of experiments for subset-selection. Bilal Riaz implemented the algorithms and results. Manuscript was written and edited by all authors.

### Acknowledgments

Bilal Riaz is supported by Higher Education Commission of Pakistan and University of Delaware. We would like to express gratitude to Matthew S. Emigh for insightful discussions and the Office of Naval Research for funding this research. Research at the University of Delaware was sponsored by the Department of the Navy, Office of Naval Research under ONR award number N00014-21-1-2300.

### Code Availability

Code for the paper is available at `https://github.com/Bilal092/support-subset-selection`.

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

# A Appendix: Lipschitz Smoothness of Dual

The dual form 16 considered in this paper does not explicitly enforce the primal problem's marginal simplex constraints on the transport plan. Consequently, the dual form is not necessarily Lipschitz smooth (Lin et al., 2019; Cuturi & Peyré, 2018; Lin et al., 2022). But in the proposed algorithm, we first update the dual variable $\boldsymbol{\alpha}$ using the Sinkhorn-like update, which implicitly enforces the simplex constraint, making the semi-dual problem $\frac{1}{\gamma}$-Lipschitz smooth with respect to $\ell_1$, $\ell_2$ and $\ell_\infty$ norms. This justifies the use of $\eta_s^{(k)} = \frac{1}{\gamma}$ in the accelerated proximal-gradient based approach to solve 16.

Recall that the Lagrangian of 16 given in equation 14 is

$$\mathcal{L}(\boldsymbol{P}, \boldsymbol{\alpha}, \boldsymbol{\beta}) = \big\langle \boldsymbol{P}, \; \boldsymbol{M} + \gamma(\log(\boldsymbol{P}) - \mathbf{1}_m \mathbf{1}_n^\top) + \boldsymbol{\alpha} \mathbf{1}_n^\top + \mathbf{1}_m \boldsymbol{\beta}^\top \big\rangle - \langle \boldsymbol{\alpha}, \boldsymbol{\mu} \rangle - \langle \boldsymbol{\beta}, \boldsymbol{\zeta} \rangle. \tag{31}$$

By Slater's conditions, the problem 16 is strongly dual therefore

$$\min_{P \succcurlyeq 0} \max_{\boldsymbol{\alpha}, \boldsymbol{\beta}} \mathcal{L}(\boldsymbol{P}, \boldsymbol{\alpha}, \boldsymbol{\beta}) = \max_{\boldsymbol{\alpha}, \boldsymbol{\beta}} \min_{P \succcurlyeq 0} \mathcal{L}(\boldsymbol{P}, \boldsymbol{\alpha}, \boldsymbol{\beta}). \tag{32}$$

In order to find the minimum of the Lagrangian with respect to $\boldsymbol{P}$, one takes its element-wise derivative with respect to $\boldsymbol{P}$ and obtains $\tilde{\boldsymbol{P}}(\boldsymbol{\alpha}, \boldsymbol{\beta}) = \boldsymbol{D}\big(\exp\big(-\frac{1}{\gamma}\boldsymbol{\alpha}\big)\big) \exp\big(-\frac{1}{\gamma}\boldsymbol{M}\big) \boldsymbol{D}\big(\exp\big(-\frac{1}{\gamma}\boldsymbol{\beta}\big)\big)$, which can then be substituted back into the Lagrangian 31 to obtain the problem

$$\max_{\boldsymbol{\alpha}, \boldsymbol{\beta}} \quad \big\{ g(\boldsymbol{\alpha}, \boldsymbol{\beta}) = -\gamma \mathbf{1}_m^\top \tilde{\boldsymbol{P}}(\boldsymbol{\alpha}, \boldsymbol{\beta}) \mathbf{1}_n - \langle \boldsymbol{\alpha}, \boldsymbol{\mu} \rangle - \langle \boldsymbol{\beta}, \boldsymbol{\zeta} \rangle \big\}, \; \text{s.t. } \boldsymbol{\beta} \succcurlyeq 0, \tag{33}$$

which can be converted to the convex minimization problem 16 by defining $f(\boldsymbol{\alpha}, \boldsymbol{\beta}) = -g(\boldsymbol{\alpha}, \boldsymbol{\beta})$, as in

$$\min_{\boldsymbol{\alpha}, \boldsymbol{\beta}} \quad \big\{ f(\boldsymbol{\alpha}, \boldsymbol{\beta}) = \gamma \mathbf{1}_m^\top \tilde{\boldsymbol{P}}(\boldsymbol{\alpha}, \boldsymbol{\beta}) \mathbf{1}_n + \langle \boldsymbol{\alpha}, \boldsymbol{\mu} \rangle + \langle \boldsymbol{\beta}, \boldsymbol{\zeta} \rangle \big\}, \; \text{s.t. } \boldsymbol{\beta} \succcurlyeq 0, \tag{34}$$

The partial gradients of $f(\boldsymbol{\alpha}, \boldsymbol{\beta})$ with respect to $\boldsymbol{\alpha}$ and $\boldsymbol{\beta}$ are

$$\nabla_{\boldsymbol{\alpha}} f(\boldsymbol{\alpha}, \boldsymbol{\beta}) = \boldsymbol{\mu} - \exp\big(-\frac{\boldsymbol{\alpha}}{\gamma}\big) \odot \boldsymbol{K} \exp\big(-\frac{\boldsymbol{\beta}}{\gamma}\big), \tag{35a}$$

$$\nabla_{\boldsymbol{\beta}} f(\boldsymbol{\alpha}, \boldsymbol{\beta}) = \boldsymbol{\zeta} - \exp\big(-\frac{\boldsymbol{\beta}}{\gamma}\big) \odot \boldsymbol{K}^\top \exp\big(-\frac{\boldsymbol{\alpha}}{\gamma}\big). \tag{35b}$$

For twice continuously differentiable functions, the Lipschitz smoothness parameter is determined by the Hessian. The Hessian for $f(\boldsymbol{\alpha}, \boldsymbol{\beta})$ is

$$\boldsymbol{H}_f(\boldsymbol{\alpha}, \boldsymbol{\beta}) = \begin{bmatrix} \nabla_{\boldsymbol{\alpha}}^\top \nabla_{\boldsymbol{\alpha}} f(\boldsymbol{\alpha}, \boldsymbol{\beta}) & \nabla_{\boldsymbol{\beta}}^\top \nabla_{\boldsymbol{\alpha}} f(\boldsymbol{\alpha}, \boldsymbol{\beta}) \\ \nabla_{\boldsymbol{\alpha}}^\top \nabla_{\boldsymbol{\beta}} f(\boldsymbol{\alpha}, \boldsymbol{\beta}) & \nabla_{\boldsymbol{\beta}}^\top \nabla_{\boldsymbol{\beta}} f(\boldsymbol{\alpha}, \boldsymbol{\beta}) \end{bmatrix}, \tag{36}$$

where

$$\nabla_{\boldsymbol{\alpha}}^\top \nabla_{\boldsymbol{\alpha}} f(\boldsymbol{\alpha}, \boldsymbol{\beta}) = \frac{1}{\gamma} \boldsymbol{D}\big( \exp\big(-\frac{\boldsymbol{\alpha}}{\gamma}\big) \odot \boldsymbol{K} \exp\big(-\frac{\boldsymbol{\beta}}{\gamma}\big) \big) = \frac{1}{\gamma} \boldsymbol{D}\big(\tilde{\boldsymbol{P}}(\boldsymbol{\alpha}, \boldsymbol{\beta}) \mathbf{1}_n\big), \tag{37a}$$

$$\nabla_{\boldsymbol{\beta}}^\top \nabla_{\boldsymbol{\beta}} f(\boldsymbol{\alpha}, \boldsymbol{\beta}) = \frac{1}{\gamma} \boldsymbol{D}\big( \exp\big(-\frac{\boldsymbol{\beta}}{\gamma}\big) \odot \boldsymbol{K}^\top \exp\big(-\frac{\boldsymbol{\alpha}}{\gamma}\big) \big) = \frac{1}{\gamma} \boldsymbol{D}\big(\tilde{\boldsymbol{P}}(\boldsymbol{\alpha}, \boldsymbol{\beta})^\top \mathbf{1}_m\big), \tag{37b}$$

$$\nabla_{\boldsymbol{\beta}}^\top \nabla_{\boldsymbol{\alpha}} f(\boldsymbol{\alpha}, \boldsymbol{\beta}) = \frac{1}{\gamma} \big( \boldsymbol{K} \odot \exp\big(-\frac{\boldsymbol{\alpha}}{\gamma}\big) \exp\big(-\frac{\boldsymbol{\beta}^\top}{\gamma}\big) \big) = \frac{1}{\gamma} \tilde{\boldsymbol{P}}(\boldsymbol{\alpha}, \boldsymbol{\beta}), \tag{37c}$$

$$\nabla_{\boldsymbol{\alpha}}^\top \nabla_{\boldsymbol{\beta}} f(\boldsymbol{\alpha}, \boldsymbol{\beta}) = \frac{1}{\gamma} \big( \boldsymbol{K}^\top \odot \exp\big(-\frac{\boldsymbol{\beta}}{\gamma}\big) \exp\big(-\frac{\boldsymbol{\alpha}^\top}{\gamma}\big) \big) = \frac{1}{\gamma} \tilde{\boldsymbol{P}}(\boldsymbol{\alpha}, \boldsymbol{\beta})^\top. \tag{37d}$$

The Sinkhorn update for $\boldsymbol{\alpha}$ in equation 19 ensures that after each update of $\boldsymbol{\alpha}$, the transport plan lies on the probability simplex and matches the target marginal $\boldsymbol{\mu} = \tilde{\boldsymbol{P}}(\boldsymbol{\alpha}^{(k+1)}, \boldsymbol{\beta}^{(k)}) \mathbf{1}_n$. Defining $\tilde{\boldsymbol{\nu}} = \tilde{\boldsymbol{P}}(\boldsymbol{\alpha}^{(k+1)}, \boldsymbol{\beta}^{(k)})^\top \mathbf{1}_m$, the Hessian equation 36 at $(\boldsymbol{\alpha}^{(k+1)}, \boldsymbol{\beta}^{(k)})$ is compactly written as

$$\boldsymbol{H}_f(\boldsymbol{\alpha}^{(k+1)}, \boldsymbol{\beta}^{(k)}) = \frac{1}{\gamma} \begin{bmatrix} \boldsymbol{D}(\boldsymbol{\mu}) & \tilde{\boldsymbol{P}}(\boldsymbol{\alpha}^{(k+1)}, \boldsymbol{\beta}^{(k)}) \\ \tilde{\boldsymbol{P}}(\boldsymbol{\alpha}^{(k+1)}, \boldsymbol{\beta}^{(k)})^\top & \boldsymbol{D}(\tilde{\boldsymbol{\nu}}) \end{bmatrix}.$$

We use the induced-norms of the Hessian $\boldsymbol{H}_f(\boldsymbol{\alpha}^{(k+1)}, \boldsymbol{\beta}^{(k)})$ to characterize the smoothness at $(\boldsymbol{\alpha}^{(k+1)}, \boldsymbol{\beta}^{(k)})$. For a matrix $\boldsymbol{A} \in \mathbb{R}^{m \times n}$, the induced norm $\|\cdot\|_{p,q}$ is defined as

$$\|\boldsymbol{A}\|_{p,q} := \max_{\boldsymbol{x}:\|\boldsymbol{x}\|_p \leq 1} \|\boldsymbol{A}\boldsymbol{x}\|_q. \tag{38}$$

The twice continuously differentiable function $f(\boldsymbol{\alpha}^{(k+1)}, \boldsymbol{\beta}^{(k)})$ is $L$-Lipschitz with respect to $\ell_p$ norm, if $\|\boldsymbol{H}_f(\boldsymbol{\alpha}^{(k+1)}, \boldsymbol{\beta}^{(k)})\|_{p,q} \leq L$, where $\ell_q$ is the dual norm of the $\ell_p$ norm. Since the Hessian $\boldsymbol{H}_f(\boldsymbol{\alpha}^{(k+1)}, \boldsymbol{\beta}^{(k)})$ is a non-negative matrix, one can observe that all its matrix entries are less than $\frac{1}{\gamma} \max \{\mu_{\max}, \tilde{\nu}_{\max}\}$, where $\mu_{\max}$ and $\tilde{\nu}_{\max}$ are maximum entries of $\boldsymbol{\mu}$ and $\tilde{\boldsymbol{\nu}}$ respectively. Therefore for $p = 1$, if one can find the column index $k$ corresponding to a matrix entry with value $\frac{1}{\gamma} \max\{\mu_{\max}, \tilde{\nu}_{\max}\}$, then $\boldsymbol{x} = \boldsymbol{e}_k$ is the vertex of the $\ell_1$ norm-ball where $\|\boldsymbol{H}_f(\boldsymbol{\alpha}^{(k+1)}, \boldsymbol{\beta}^{(k)})\boldsymbol{x}\|_\infty = \frac{1}{\gamma} \max\{\mu_{\max}, \tilde{\nu}_{\max}\}$. Thus,

$$\|\boldsymbol{H}_f(\boldsymbol{\alpha}^{(k+1)}, \boldsymbol{\beta}^{(k)})\|_{1,\infty} = \frac{1}{\gamma} \max \{\mu_{\max}, \tilde{\nu}_{\max}\} \leq \frac{1}{\gamma}, \tag{39}$$

which proves the function $f(\boldsymbol{\alpha}^{(k+1)}, \boldsymbol{\beta}^{(k)})$ is $\frac{1}{\gamma}$-Lipschitz with respect to the $\ell_1$ norm. Since all the matrix entries of the Hessian $\boldsymbol{H}_f(\boldsymbol{\alpha}^{(k+1)}, \boldsymbol{\beta}^{(k)})$ are less than $\frac{1}{\gamma}$, its spectral radius is less than $\frac{1}{\gamma}$ (Horn & Johnson, 2012)(Theorem 8.1.18) and

$$\|\boldsymbol{H}_f(\boldsymbol{\alpha}^{(k+1)}, \boldsymbol{\beta}^{(k)})\|_{2,2} = \lambda_{\max}(\boldsymbol{H}_f) \leq \frac{1}{\gamma}. \tag{40}$$

Therefore, the function $f(\boldsymbol{\alpha}^{(k+1)}, \boldsymbol{\beta}^{(k)})$ is $\frac{1}{\gamma}$-Lipschitz with respect to the $\ell_2$ norm. For $p = \infty$, one can maximize the norm $\|\boldsymbol{H}_f(\boldsymbol{\alpha}^{(k+1)}, \boldsymbol{\beta}^{(k)})\boldsymbol{x}\|_1$, at the vertex of the $\ell_\infty$ ball where all entries are unit magnitude, in particular $\boldsymbol{x} = \boldsymbol{1}_{m+n}$, which results into

$$\boldsymbol{H}_f(\boldsymbol{\alpha}^{(k+1)}, \boldsymbol{\beta}^{(k)})\boldsymbol{1}_{m+n} = \frac{1}{\gamma} \begin{bmatrix} \boldsymbol{\mu} + \tilde{\boldsymbol{P}}(\boldsymbol{\alpha}^{(k+1)}, \boldsymbol{\beta}^{(k)})\boldsymbol{1}_n \\ \tilde{\boldsymbol{P}}(\boldsymbol{\alpha}^{(k+1)}, \boldsymbol{\beta}^{(k)})^\top \boldsymbol{1}_m + \tilde{\boldsymbol{\nu}} \end{bmatrix} = \frac{2}{\gamma} \begin{bmatrix} \boldsymbol{\mu} \\ \tilde{\boldsymbol{\nu}} \end{bmatrix}. \tag{41}$$

Therefore,

$$\|\boldsymbol{H}_f(\boldsymbol{\alpha}^{(k+1)}, \boldsymbol{\beta}^{(k)})\|_{\infty,1} = \frac{2}{\gamma} \left\| \begin{matrix} \boldsymbol{\mu} \\ \tilde{\boldsymbol{\nu}} \end{matrix} \right\|_1 = \frac{4}{\gamma}, \tag{42}$$

and the function $f(\boldsymbol{\alpha}^{(k+1)}, \boldsymbol{\beta}^{(k)})$ is $\frac{4}{\gamma}$-Lipschitz with respect to the $\ell_\infty$ norm. In summary, the partial gradients are all $\frac{1}{\gamma}$-Lipschitz smooth with respect to $\ell_1$, $\ell_2$ and $\ell_\infty$ norms. Additionally, considering the dual variables $\boldsymbol{\alpha}$ and $\boldsymbol{\beta}$, seperately, one can see that

$$\|\nabla_{\boldsymbol{\alpha}}^\top \nabla_{\boldsymbol{\alpha}} f(\boldsymbol{\alpha}^{(k+1)}, \boldsymbol{\beta}^{(k)})\|_{1,\infty} = \frac{\mu_{\max}}{\gamma} \leq \frac{1}{\gamma}, \quad \|\nabla_{\boldsymbol{\alpha}}^\top \nabla_{\boldsymbol{\alpha}} f(\boldsymbol{\alpha}^{(k+1)}, \boldsymbol{\beta}^{(k)})\|_{2,2} \leq \frac{1}{\gamma}, \quad \|\nabla_{\boldsymbol{\alpha}}^\top \nabla_{\boldsymbol{\alpha}} f(\boldsymbol{\alpha}^{(k+1)}, \boldsymbol{\beta}^{(k)})\|_{\infty,1} = \frac{1}{\gamma},$$

and

$$\|\nabla_{\boldsymbol{\beta}}^\top \nabla_{\boldsymbol{\beta}} f(\boldsymbol{\alpha}^{(k+1)}, \boldsymbol{\beta}^{(k)})\|_{1,\infty} = \frac{\tilde{\nu}_{\max}}{\gamma} \leq \frac{1}{\gamma}, \quad \|\nabla_{\boldsymbol{\beta}}^\top \nabla_{\boldsymbol{\beta}} f(\boldsymbol{\alpha}^{(k+1)}, \boldsymbol{\beta}^{(k)})\|_{2,2} \leq \frac{1}{\gamma}, \quad \|\nabla_{\boldsymbol{\beta}}^\top \nabla_{\boldsymbol{\beta}} f(\boldsymbol{\alpha}^{(k+1)}, \boldsymbol{\beta}^{(k)})\|_{\infty,1} = \frac{1}{\gamma}.$$

Therefore, $f(\boldsymbol{\alpha}^{(k+1)}, \boldsymbol{\beta}^{(k)})$ is separately $\frac{1}{\gamma}$-Lipschitz for both $\boldsymbol{\alpha}$ and $\boldsymbol{\beta}$ with respect to $\ell_1, \ell_2$ and $\ell_\infty$ norms.

## B    Appendix: Entropic Regularization Results

In this Appendix, results for point cloud registration and color transfer for `SS-Entropic` are displayed, which can be compared to the results for `SS-Bregman` in the main body.

**Fragmented Hypercubes**: Figure 14 shows the results for affine transformation optimization in 2D, which can be compared with the results in Figure 3(a). Visually it is clear that the alignment is much worse for values of $c \in \{1.25, 1.5\}$, but quantitatively it is worse for all values of $c > 1$ as `SS-Bregman` achieves cost below $10^{-2}$ at 500 iterations of $\boldsymbol{\Theta}$ updates. Figure 15 shows the results in 3D using `SS-Entropic`, which can be compared with the results in Figure 4(a). In this case, results are quantitatively worse for all values of $c > 1$ as `SS-Bregman` achieves cost below $10^{-1}$ at 500 iterations $\boldsymbol{\Theta}$ updates.

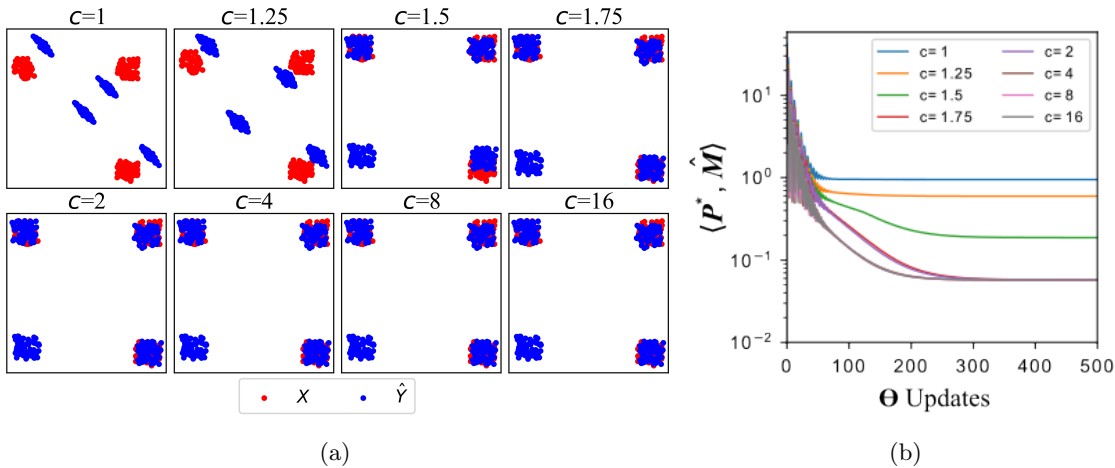

Figure 14: Results for affine transformation optimization with subset selection for partial optimal transport. Target points $X$ are sampled from a 2D fragmented hypercube centered at the origin with negative coordinates removed, whereas source points $Y$ are sampled from a translated fragmented hypercube. (a) Target and transformed source points after application of optimized affine transformation. Subset selection problems are solved using the `SS-Entropic` with $\gamma = 0.01$ with *max-iter* = 4000.(b) Loss function curves for scaling parameter $c \in \{1, 1.25, 1.5, 1.75, 2, 4, 8, 16\}$.

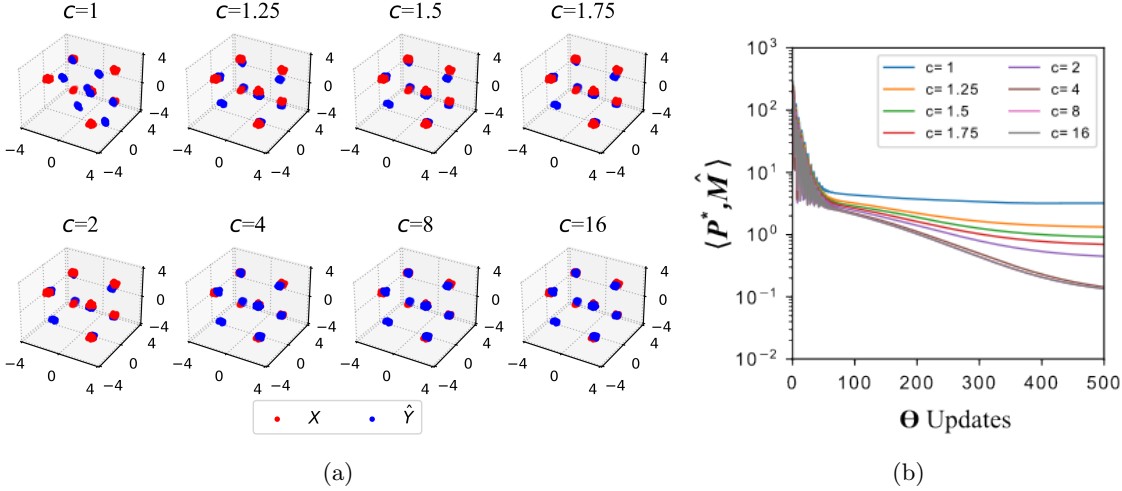

Figure 15: Results for affine transformation optimization with subset selection for partial optimal transport. Target points $X$ are sampled from a 3D fragmented hypercube centered at the origin with negative coordinates removed, whereas source points $Y$ are sampled from a translated fragmented hypercube. (a) Target and transformed source points after application of optimized affine transformation. Subset selection problems are solved using the `SS-Entropic` with $\gamma = 0.01$ with *max-iter* = 4000.(b) Loss function curves for scaling parameter $c \in \{1, 1.25, 1.5, 1.75, 2, 4, 8, 16\}$.

**Partial point cloud registration**: The results for partial point cloud registration with entropically regularized subset selection (`SS-Entropic`) for the Stanford bunny and armadillo point clouds. It is clear that the entropically regularized form alone fails to find a meaningful correspondence, transforming the source such that is completely covered by the partial point cloud.

**Color Transfer**: The results for color transfer with the entropically regularized subset selection `SS-Entropic` are shown in Figure 17, which can be compared to results from `SS-Bregman` shown in Figure 6 and Figure 7. Namely, for the first image "Louisiana Nature Scene Barataria Preserve" the entropically

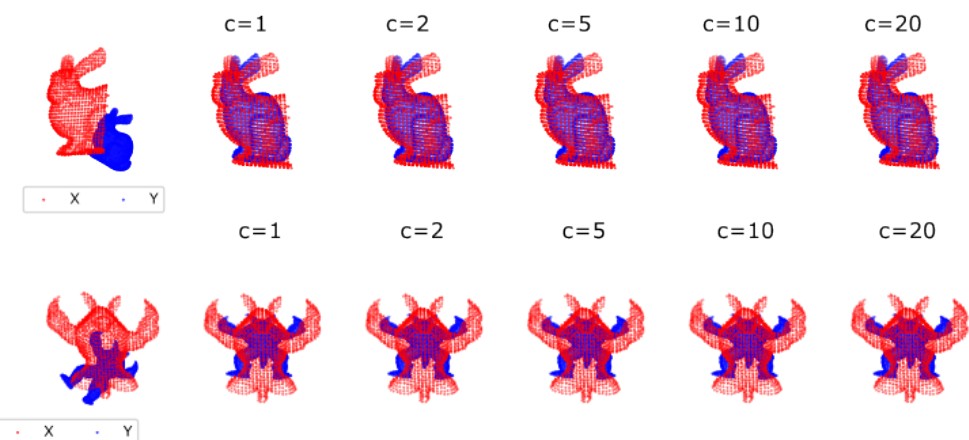

Figure 16: Bunny and Armadillo partial point cloud registration using entropically regularized subset selection SS-Entropic $\gamma = 0.05$, fails to find a accurate alignment of the source with the partially occluded target.

regularized results appear more monochromatic with less distinct colors. In the second set of images, there is no visual difference between the outputs of SS-Entropic and SS-Bregman.

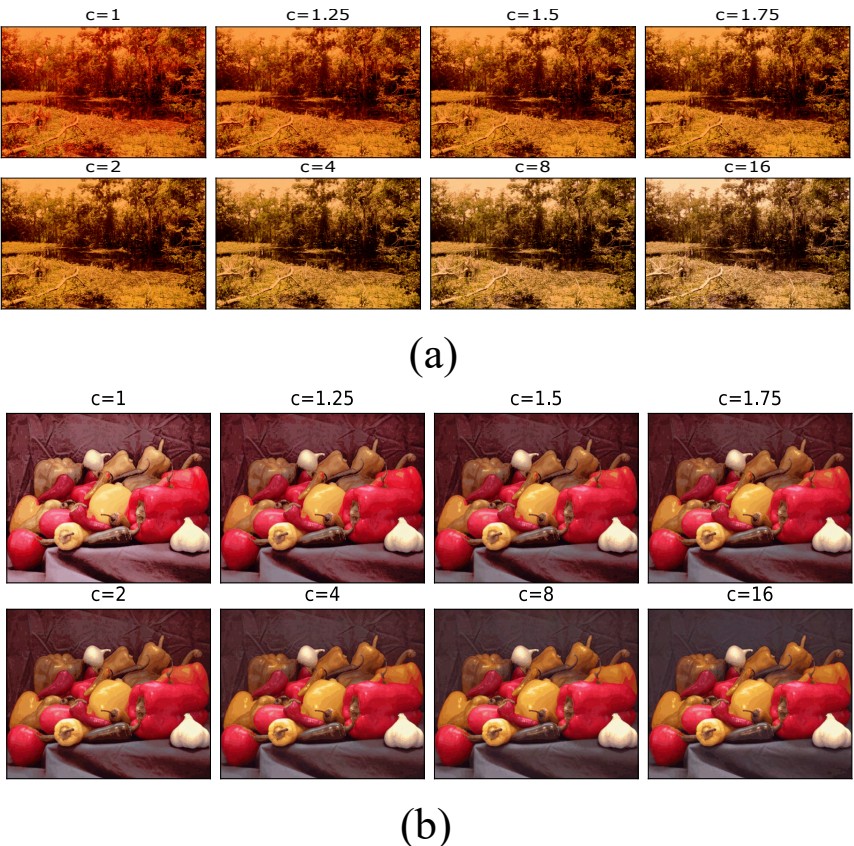

Figure 17: Color transfer results for $c \in \{1, 1.25, 1.5, 1.75, 2, 4, 8, 16\}$ using SS-Entropic.

## C   Appendix: Comparison with Semi-Relaxed Formulations

We analyze the variation of subset-selection mass assignments $\nu_j^* = \sum_{i=1}^m P_{ij}^*$, for $j \in [n]$ as a function of $c$ for source and target points ($m = 100, n = 80$) given in Figure 2. We use SS-Entropic and SS-Bregman and compare to solutions for the subset selection linear program using CVXPY (Diamond & Boyd, 2016; Agrawal et al., 2018). We also use CVXPY to solve following semi-relaxed unbalanced optimal transport problems of the form

$$\min_{\boldsymbol{P} \succcurlyeq 0} \quad \langle \boldsymbol{P}, \boldsymbol{M} \rangle + \frac{1}{\rho} \mathcal{D}_\varphi(\boldsymbol{P}^\top \mathbf{1}_m \| \boldsymbol{\nu})$$
$$\text{s.t.} \quad \boldsymbol{P}\mathbf{1}_n = \boldsymbol{\mu},$$

where $\mathcal{D}_\varphi \in \{\text{TV}, \ell_2, \ell_\infty\}$ can be either TV distance (TV), squared Euclidean ($\ell_2$), or $\ell_\infty$-norm based distance, and $\rho > 0$. It can be observed from Figure 18 that the sparsity patterns of mass assignments obtained using SS-Bregman closely match the linear program solutions across a range of $c$, whereas mass assignments obtained using SS-Entropic are more dense with less mass assignments equal to 0. The sparsity patterns for penalty-based relaxations differ from the linear program solutions. By comparing the mass assignments of different solutions in terms of number of non-zeros, i.e., the cardinality of the support of solution's marginal $\boldsymbol{\nu}^*$, denoted as (card), we note that the assignments obtained using subset selection have the largest entropy, which implies that subset selection tends to assign uniform masses to the selected subset of points. In comparison, the TV-based penalty, corresponding to the $\ell_1$-norm tends to have sparse deviations from uniform $\frac{1}{n}$ with many points at exactly this value. Entropies of $\boldsymbol{\nu}^*$ are plotted in Figure 19.

To further elaborate our observations, we plot the mass assignments to unlabeled data points $\boldsymbol{\nu}$, across different values of scaling parameter $c$ for MNIST/EMNIST PU learning in Figure 20 and Figure 21 for CIFAR-10 PU learning, as discussed in 3.3, and compared that with the mass assignments using semi-relaxed formulations ($\ell_2$ and TV) with different values of regularization parameter $\rho$. We observe that paths taken by our formulation differs from both $\ell_2$ and TV . But the solutions correspond at the extreme limits: as $\rho \to 0$ and $c \to 1$ all solutions approach optimal transport solution, similarly as $\rho \to \infty$ and $\forall \, c \geq c^*$ mass assignments correspond to nearest neighbor solutions discussed in the Section 2.1. For PU learning experiments on MNIST/EMNIST and CIFAR10, we used the POT toolbox Flamary et al. (2021) function optim.semirelaxed_cg to obtain the solutions for semi-relaxed optimal transport problem with TV and squared-$\ell_2$ penalties via the Frank-Wolfe conditional gradient algorithm.

## D   Appendix: Relation between Entropically Regularized Subset Selection and Unbalanced Optimal Transport

Using the notation from our paper, we have adapted the formulations for the unbalanced optimal transport with asymmetric penalties from the work by Séjourné et al. (2019). For $\varphi_1, \varphi_2$, the generating functions for the divergence penalty functions on the marginals, and entropic regularization parameter $\gamma > 0$, the unbalanced optimal transport for discrete probability measures $\mu$, $\nu$, and cost $\mathbf{M}$ is

$$\begin{aligned}
\mathcal{UOT}_\gamma^{(\varphi_1,\varphi_2)}(\mu,\nu) &= \min_{\boldsymbol{P} \in \mathbb{R}_{\geq 0}^{m \times n}} \quad \langle \boldsymbol{P}, \mathbf{M} \rangle + D_{\varphi_1}(\boldsymbol{P}\mathbf{1} \| \boldsymbol{\mu}) + D_{\varphi_2}(\boldsymbol{P}^\top \mathbf{1} \| \boldsymbol{\nu}) + \gamma \mathrm{KL}(\boldsymbol{P} \| \boldsymbol{\mu}\boldsymbol{\nu}^\top) \\
&= \min_{\boldsymbol{P} \in \mathbb{R}_{\geq 0}^{m \times n}} \quad \langle \boldsymbol{P}, \mathbf{M} \rangle + \sum_{i=1}^m \mu_i \varphi_1\Big(\frac{\sum_j P_{ij}}{\mu_i}\Big) + \sum_{i=1}^n \nu_i \varphi_2\Big(\frac{\sum_i P_{ij}}{\nu_i}\Big) \\
&\quad + \gamma\Big(\sum_{ij} P_{ij}(\log(P_{ij}) - \log(\mu_i \nu_j)) - \sum_{ij} P_{ij} + \sum_{ij} \mu_i \nu_j\Big) \\
&= \min_{\boldsymbol{P} \in \mathbb{R}_{\geq 0}^{m \times n}} \langle \boldsymbol{P}, \mathbf{M} \rangle + \sum_{i=1}^m \mu_i \varphi_1\Big(\frac{\sum_j P_{ij}}{\mu_i}\Big) + \sum_{i=1}^n \nu_i \varphi_2\Big(\frac{\sum_i P_{ij}}{\nu_i}\Big) \\
&\quad + \gamma\left(\langle \boldsymbol{P}, \mathbf{log}(\boldsymbol{P}) - \mathbf{1}_{m \times n}\rangle - \sum_i \mu_i \log(\mu_i) + \sum_j \sum_i P_{ij} \log(\nu_i) + 1\right).
\end{aligned}$$

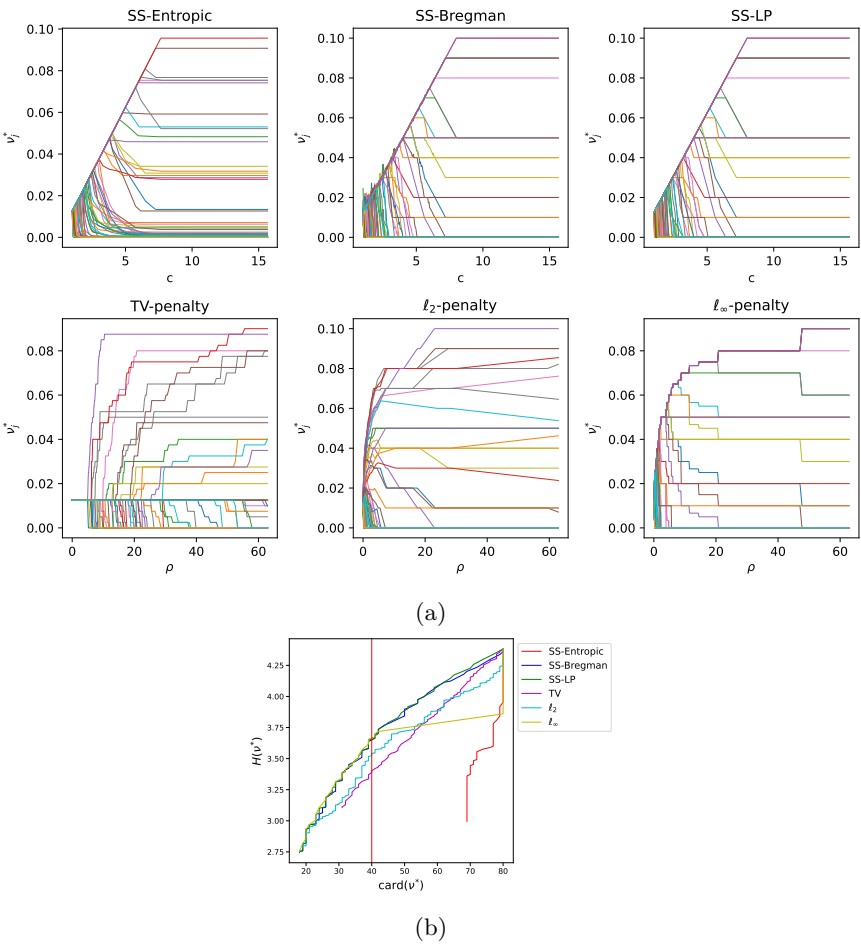

(a)

(b)

Figure 18: (a) The variation of mass assignment vector $\boldsymbol{\nu}^*$ with $c$ for toy problem in Figure 2, using `SS-Entropic`, `SS-Bregman`, and the subset selection linear program 2 is solved using CVXPY Agrawal et al. (2018); Diamond & Boyd (2016). All divergence regularized semi-relaxed problems are also solved using CVXPY. (b) Entropy versus cardinality selected set of points. It can be observed that `SS-Bregman` and $\ell_\infty$ solutions match each other until a support cardinality of around 40. This is due to fact that subset selection solution matches a unique $\ell_\infty$ penalized solution for all values of $c > 2$.

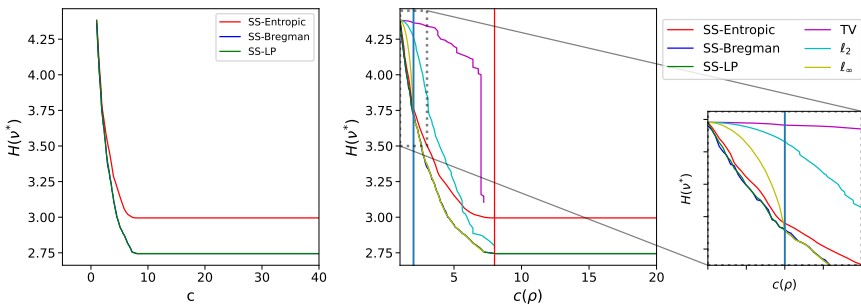

Figure 19: (a) Entropy of mass assignment vector $\boldsymbol{\nu}^*$ with $c$ for toy problem in Figure 2. (b) Entropy as function of scaling parameter for penalized relaxed-OT problems $c(\rho)$, obtained using formula $c(\rho) = \max_i \frac{\nu_i^*(\rho)}{\nu_i}$, which implies that for sufficiently large values of $\rho$, $c(\rho) = c^*$.

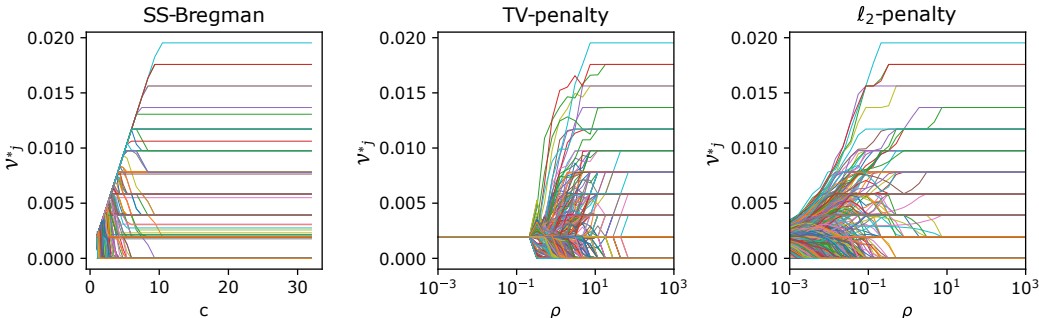

Figure 20: Mass assignments $\boldsymbol{\nu}^*$ to unlabeled data points in PU learning on MNIST/EMNIST discussed in Section 3.3.1 across $c$ for `SS-Bregman` and across $\rho$ for TV and $\ell_2$ penalties.

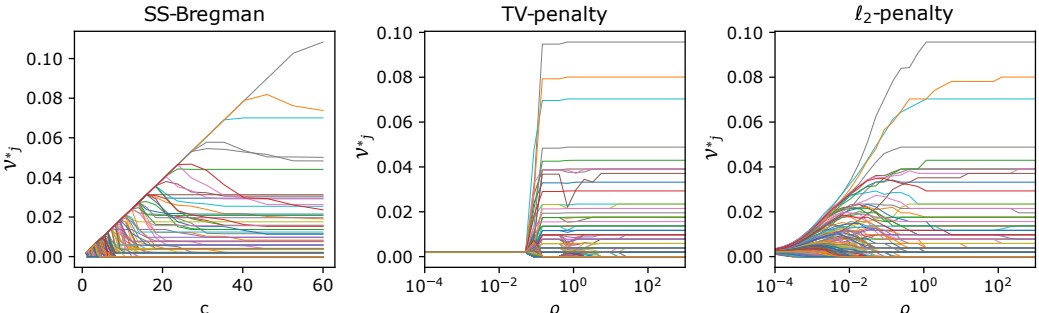

Figure 21: Mass assignments $\boldsymbol{\nu}^*$ to unlabeled data points in PU learning on CIFAR-10 neural network representations, discussed in Section 3.3.2 across $c$ for `SS-Bregman` and across $\rho$ for TV and $\ell_2$ penalties.

When $\varphi_1(x) = \imath_{\{1\}}(x) = \begin{cases} 0, & x = 1 \\ \infty, & \text{otherwise} \end{cases}$ is the indicator function for the ratio of the target marginal and

$\varphi_2(x) = \imath_{[0,c]}(x) = \begin{cases} 0, & x \in [0,c] \\ \infty, & \text{otherwise} \end{cases}$ is a range constraint to the interval of $[0,c]$ for the ratio of the source marginal, then

$$\mathcal{UOT}_\gamma^{(\varphi_1, \varphi_2)}(\mu, \nu) = \min_{\boldsymbol{P} \in \mathbb{R}_{\geq 0}^{m \times n}} \langle \boldsymbol{P}, \mathbf{M} \rangle + \gamma \left( \langle \boldsymbol{P}, \log(\boldsymbol{P}) - \mathbf{1}_{m \times n} \rangle - \langle \boldsymbol{\mu}, \log(\boldsymbol{\mu}) \rangle + \langle \boldsymbol{P}^\top \mathbf{1}_m, \log(\boldsymbol{\nu}) \rangle + 1 \right)$$

$$\text{s.t.} \quad \boldsymbol{P} \mathbf{1}_n = \boldsymbol{\mu}, \ \boldsymbol{P}^\top \mathbf{1}_m \preccurlyeq c\boldsymbol{\nu}.$$

With these choices of $\varphi_1, \varphi_2$ and a uniformly weighted source $\nu_i = \frac{1}{n} \quad \forall i \in [n]$, we can relate $\mathcal{S}_p^{(\gamma)}(\mu, c\nu)$ to $\mathcal{UOT}_\gamma^{(\varphi_1, \varphi_2)}(\mu, \nu)$, since $\langle \boldsymbol{P}^\top \mathbf{1}_m, \log(\boldsymbol{\nu}) \rangle = \sum_j \sum_i P_{ij} \log(\nu_i) = -\log(n)$, then $\mathcal{UOT}_\gamma^{(\varphi_1, \varphi_2)}(\mu, \nu) = \mathcal{S}_p^{(\gamma)}(\mu, c\nu) + \gamma(H(\mu) - \log(n) + 1)$, where $H(\mu) = -\langle \boldsymbol{\mu}, \log(\boldsymbol{\mu}) \rangle = -\sum_i \mu_i \log(\mu_i)$. The uniformly weighted source is necessary to eliminate the bias towards $\boldsymbol{\nu}$ for the marginal that the Kullback-Leilber divergence penalty induces compared to our entropy penalty, which only considers $\nu$ through the constraint.

# E   PU Learning Toy Comparison Test

In order to show evidence that subset selection, as a semi-relaxed partial optimal transport, can consistently outperform fully-relaxed partial optimal transport on PU learning, we plot the mean accuracy for a range of examples using of the same design as those in Figure 22.

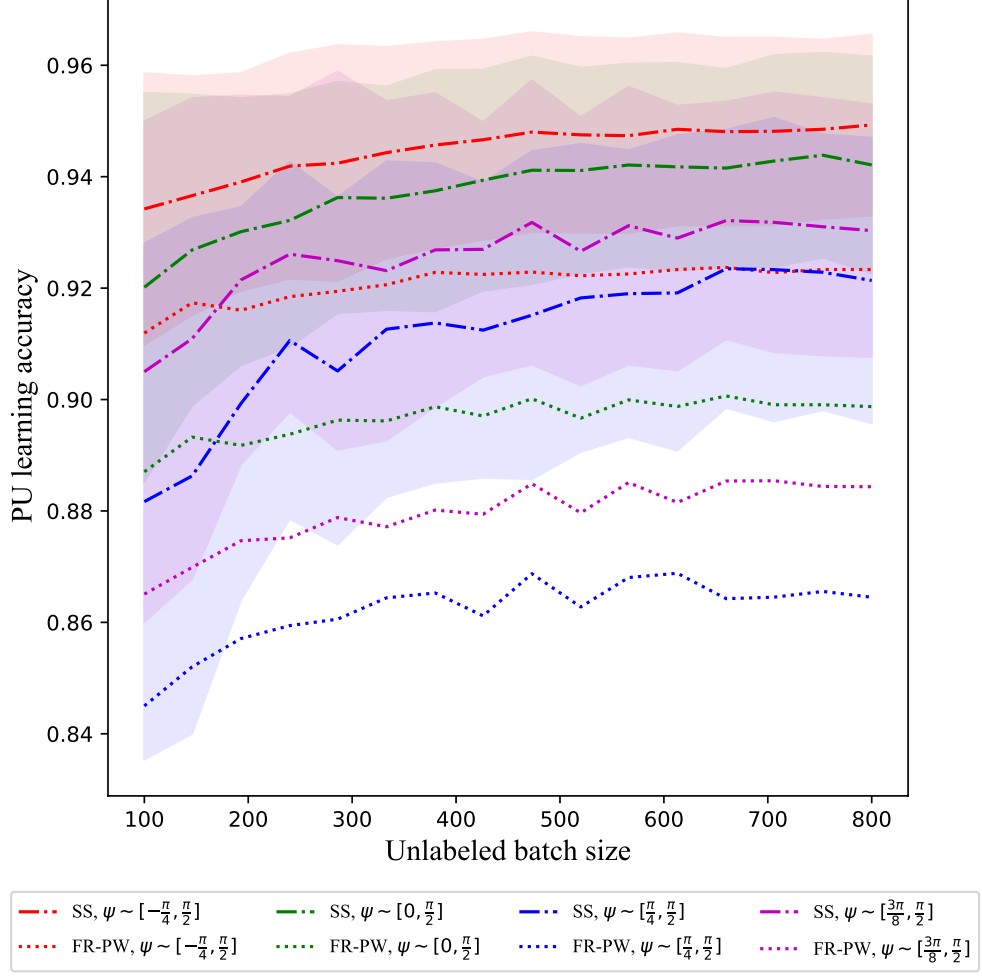

Figure 22: PU learning accuracy on toy data using subset selection and fully-relaxed partial optimal transport. Subset set consistently outperforms fully-relaxed on this toy data. Data is generated as $[r\cos(\psi),\ r\sin(\psi)]$ where the radius $r$ is drawn from a truncated exponential distribution with density $2\exp(-r)\imath_{[\log 2,\infty)}(r)$ and $\psi$ is uniform over the set of angles $\left[-\frac{\pi}{2},\frac{\pi}{2}\right]$ for the source and different subsets for the target as indicated in the legend. For each run $m = 100$ positive points are drawn and the size of the unlabeled sample is varied from $n = 100$ to $n = 800$ while are all from the third class. The solutions are obtained using the known number of positives $n_+$ out of the $n$ source points, setting $c = \frac{n}{n_+}$ for our semi-relaxed approach and $s = \frac{n_+}{n}$ for the fully-relaxed partial optimal transport. Confidence intervals show $\pm 1$ standard deviations of the accuracy across 100 runs.

## F  Appendix: Neural Network Model Architectures

For semi-supervised learning, we used neural networks to both perform the classification and provide a learning representation space in which to perform the optimal transport to assign pseudo-labels to both unlabeled points. For the CIFAR-10 data set, we used the ResNet-18 (He et al., 2016) architecture. For MNIST and Fashion-MNIST, we used custom, but simple, model architectures. Both architectures contain two convolutional layers, followed by three fully-connected layers. The model used to train Fashion-MNIST (FMNIST) classifier contains additional batch-normalization layer between the convolutional layers. Optimization algorithms along with related hyper-parameters are in Section 3.4. The code below details the exact architectures along with types and shapes of all transformations.

```python
import torch
import torch.nn as nn
import torch.nn.functional as F

class MNIST_classifier(nn.Module):
    def __init__(self):
        super().__init__()
        self.conv1 = nn.Conv2d(in_channels=1, out_channels=5,kernel_size=(5,5))
        self.conv2 = nn.Conv2d(in_channels=5, out_channels=1,kernel_size=(5,5))
        self.fc1 = nn.Linear(400, 128)
        self.fc2 = nn.Linear(128, 64)
        self.fc3 = nn.Linear(64, 10)

    def forward(self, x):
        x = F.relu(self.conv1(x))
        x = F.relu(self.conv2(x))
        x = torch.flatten(x, 1)
        x = F.relu(self.fc1(x))
        x_rep = F.relu(self.fc2(x))
        x = self.fc3(x_rep)
        return x, x_rep

class FMNIST_classifier(nn.Module):
    def __init__(self):
        super().__init__()
        self.conv1 = nn.Conv2d(in_channels=1, out_channels=32, kernel_size=(5, 5))
        self.batchN1 = nn.BatchNorm2d(num_features=32)
        self.conv2 = nn.Conv2d(in_channels=32, out_channels=64, kernel_size=(5, 5))
        self.fc1 = nn.Linear(in_features=64*4*4, out_features=128)
        self.fc2 = nn.Linear(in_features=128, out_features=64)
        self.fc3 = nn.Linear(in_features=64, out_features=10)

    def forward(self, x):
        x = self.conv1(x)
        x = F.relu(F.max_pool2d(input=x, kernel_size=2, stride=2))
        x = self.batchN1(x)
        x = self.conv2(x)
        x = F.relu(F.max_pool2d(input=x, kernel_size=2, stride=2))
        x = torch.flatten(x, 1)
        x = F.relu(self.fc1(x))
        x_rep = self.fc2(x)
        x = self.fc3(x_rep)
        return x, x_rep
```

Listing 1: Models used for training classifiers for MNIST and Fashion-MNIST

