# OpenReview forum: "Partial Optimal Transport for Support Subset Selection"
_TMLR — Accepted by TMLR_

### Review · Reviewer_4Dfp · 2023-08-21

**Summary Of Contributions:**

The paper deals with the problem of solving an instance of a semi-relaxed optimal transport problem, which differs from the original OT problem by relaxing the constraint on the marginal of one of the distributions. Algorithms to solve the problem are described, and a set of experiments, ranging from toy examples that illustrate the behavior of the solution, to color transfer, PU learning, and semi-supervised learning, is provided.

**Audience:**

Yes

**Broader Impact Concerns:**

There seems to be no ethical implications of the work.

**Claims And Evidence:**

Yes

**Requested Changes:**

See weaknesses about the position of the paper with respect to the literature and justification of the formulation (entropic formulation of a partial OT problem).

**Strengths And Weaknesses:**

**Strengths of the paper**

The paper is mostly comprehensive in the formulation of the problem and on the description of the algorithms. The experiments provided showcase the behavior of the proposed method in a wide variety of contexts, from toy examples that illustrate the behavior of the solution (in particular, the impact of the parameter that tunes the "partialness" of the solution) to experiments in image and machine learning contexts, showing that the problem that is addressed is useful in a number of applications.


**Weaknessess of the paper**

While relaxing one of the two marginals' constraints has previously been proposed in existing literature, there is a lack of discussion regarding how the proposed approach positions itself in relation to the current state of the art. For instance, [1], p. 2, paragraph "Relaxed Weighted OT," relaxes the matching constraint towards one distribution by introducing a set of relaxation parameters to increase (or decrease) the capacity value of each sample; [2], eq. (14), defines the "semi-relaxed OT," in which one of the marginals is free.

In the context of support subset selection, one advantage of using partial OT is that it defines a set of active regions, that is to say a subset of the samples. As stated in section 2.3, the entropic formulation of the problem yields denser couplings, which may be detrimental to the identification of active regions. A Bregman Proximal-point method is provided in Algorithm 2; as mentioned in the paper, it can result in infeasible solutions and/or high computational time. One might then wonder why not utilize a different yet related problem for active region identification that would inherently leads to sparse solutions, such as unbalanced OT with an $\ell_2$ penalization rather than an $\ell_1$ one, as in [2].

Furthermore, the experimental section lacks comparison with competitors.


[1]  RABIN, Julien, FERRADANS, Sira, et PAPADAKIS, Nicolas. Adaptive color transfer with relaxed optimal transport. IEEE international conference on image processing (ICIP), 2014.

[2] BLONDEL, Mathieu, SEGUY, Vivien, et ROLET, Antoine. Smooth and sparse optimal transport. International conference on artificial intelligence and statistics, 2018.


*Other comments*
- In the introduction, the second paragraph discussing partial OT might be misleading. When $\mu$ is discrete, the TV norm is equivalent to the $\ell_1$ norm of vectors, hence it comes down to a partial OT problem.
- On page 4, the claims regarding extreme points and breakpoints lack supporting arguments.
- Regarding the PU learning experiments, it remains unclear why the relative ranking provided by the entropic formulation outperforms the unconstrained one. Could you please elaborate on this point?

---

> ### Author Response · Authors · 2023-10-14
> **Reply to Reviewer 4Dfp 1/2**
>
> We are thankful to the reviewer for the thorough reading  and constructive criticism of the manuscript, which was helpful for us to both improve the quality of the work. We agree that the relation to prior work was not complete. We have added a new subsection 2.2 following the methodology that relates prior work, including the papers listed:
>
> Regarding the work of Rabin, Ferradans, and Papadakis (2014), our constraint do match the feasible set. However, the more general convex optimization problem in that work is distinct (besides the additional costs involved) since it treats the capacity defining the constraint as an parameter of the optimization and regularizes it using the $\ell_1$-norm of the difference from ones. We detail the vertices of the marginal's feasible set showing the solutions are distinct.
>
> Regarding the work of Blondel, Seguy, and Rolet (2018), the semi-relaxed optimal transport with an $\ell_\infty$-norm based divergence will yield an equivalent problem to ours for $c\ge2$. In contrast, the solutions for $\ell_2$-norm or $\ell_1$-norm yields different results. As mentioned, employing an $\ell_{2}$ penalty results may also yield sparse solutions, but that the work that we follow Xie et al. uses inexact proximal mappings with entropic regularization can also render sparse solutions close to original linear program.
>
> Generally, the problem formulations (linear and entropically regularized) we consider are similar (or subsumed) by the extensive prior work semi-relaxed  balanced or unbalanced OT problems. Our contribution is a scalable algorithm (Algorithm 2 $\texttt{SS-Bregman}$) that obtains solutions close to the linear problem formulation for the subset selection. We explore the effect of the constraint parameter on the solution, which has been explored previously explored in the semi-relaxed case with $\ell_2$ penalty in the work by Chapel et al. (2021). Recent work of Phatak et al. (2023) have considered the regularization path for partial OT, pointing toward promising future directions of our work situated between these two.
>
> In the revision, we compare results to competitors in terms of semi-relaxed partial optimal transport with different penalties.
> Through the solution paths of mass assignments on toy data and PU-Learning examples, we demonstrated that solution obtained by our formulation differ from the ones obtained using semi-relaxed optimal transport with total variation or squared Euclidean norm penalties. Toy and PU learning examples demonstrating differences in solutions are included in the revised Appendix.

---

> ### Author Response · Authors · 2023-10-14
> **Reply to Reviewer 4Dfp 2/2**
>
> # Other comments:
>
> 1.  We agree that TV-norm based penalties are equivalent to an $\ell_1$ norm; we also agree that for fully-relaxed partial OT problem there is the equivalence between using TV-norm based penalties; but we disagree that these are related to the semi-relaxed partial OT problem. The semi-relaxed formulation corresponds to a range constraint (as in the works by Chizat et al. 2018 and Séjourné et al., 2019) and in some cases an $\ell_\infty$ constraint/penalty.
>
> 2.  Regarding the remarks on page 4, the extreme points for the marginal are deduced from combinatoric arguments.  For specific value of $L$ and simplex $\Delta_{n}$, we can find extreme points of  $\Xi_{n}^{(L)}$  by creating a vector $\mathbf{x}$ by greedily filling $\frac{1}{L}$ in first $\lfloor L \rfloor$ elements, then the next element is $1-\frac{\lfloor L \rfloor}{L}$, and the remaining elements are 0. After that one can compute number of unique permutations of the entries of $\mathbf{x}$, which correspond to the number of extreme points (vertices) of $\Xi_{n}^{(L)}$.   For $L \leq1 $ one can see if we fill greedily $\mathbf{x}$, only one element of $\mathbf{x}$ can be 1, therefore $\forall L\leq1 $, whole simplex is feasible with natural basis vectors as its extreme points. We agree that the breakpoint claims are not proved yet. They have been empirically verified, and we hope to derive an algorithm to identify the breakpoints in an efficient manner in future work, similar to Chapel et al. (2021).
>
> 3.  The relative ranking provided by our Algorithm 2, which approaches the original linear program formulation for semi-relaxed optimal transport, is better in terms of AUC for PU learning than method in the work of Chapel et al. (2020), because the latter's solution is constrained such that the marginal has uniform mass amongst the selected support. Thus, the solution provides no relative ordering amongst points in the support (non-zero mass). In comparison, we select a value of $c$ such that were the mass be uniformly distributed amongst the support of the same cardinality, but our solution may have a higher cardinality and non-uniform non-zero mass, resulting in source points (likely positives) nearby targets positive points getting higher mass. As those nearby points are most likely positive points, the AUC is higher.
>
> # References:
> 1.  RABIN, Julien, FERRADANS, Sira, et PAPADAKIS, Nicolas. Adaptive color transfer with relaxed optimal transport. IEEE international conference on image processing (ICIP), 2014.
>
> 2.  BLONDEL, Mathieu, SEGUY, Vivien, et ROLET, Antoine. Smooth and sparse optimal transport. International conference on artificial intelligence and statistics, 2018.
> 3. Phatak, Abhijeet, et al. "Computing all Optimal Partial Transports." The Eleventh International Conference on Learning Representations. 2023.
> 4. Séjourné et al., 2019. "Sinkhorn divergences for unbalanced optimal transport." arXiv preprint arXiv:1910.12958. https://doi.org/10.48550/arXiv.1910.12958
> 5. Chapel et al., 2021. "Unbalanced optimal transport through non-negative penalized linear regression." Advances in Neural Information Processing Systems 34: 23270-23282.
> 6. Xie et al., 2020. "A Fast Proximal Point Method for Computing Exact Wasserstein Distance."  Proceedings of The 35th Uncertainty in Artificial Intelligence Conference, PMLR 115:433-453.

---

### Review · Reviewer_BRfh · 2023-09-01

**Summary Of Contributions:**

This paper considers (a special case of) the _partial optimal transport_ (POT) problem, as introduced by Figalli (2010). Namely, in this work, authors consider two probability measures $\mu,\nu$ (typically discrete),

$$ \min_{P} \braket{P,M}, $$

for some cost matrix $M$, provided that the first marginal $1^T P$ of $P$ should be $\mu$ (so the total mass of $P$ must be $1$), but its second marginal is only required to satisfy $P 1 \leq c \nu$ for some prescribed $c \geq 1$ (hyper-parameter).
The main idea is that the second marginal of the optimal solution to this linear program, denoted by $\nu^*_c \leq c \nu$ is a probability measure that can be seen as a "subset selection" problem.

Even though this linear program can be solved by standard (simplex, interior points) methods, better computational properties can be achieved by considering its entropic regularization, adding a term $+ \gamma \braket{P, \log(P) - 1}$. Standard duality considerations yields a Sinkhorn-like algorithm with respect to two dual variables $(\alpha,\beta)$, where the standard log-sum-exp iteration holds for $\alpha$ while for $\beta$ a projection step on the non-negative ortant is required (as a consequence of the submarginal constraint).

As typical in entropic OT, the resulting (partial) transport plan has a density with respect to $\mu \otimes \nu$, which is somewhat in contradiction with the "subset selection" motivation of the problem.
To mitigate this effect, authors follow the work of (Xie et al., 2020) and rely on an additional Bregman projection scheme , which can also be solved using a Sinkhorn like algorithm, yielding a sequence of transportation plans $P^{(t)}$ which are guaranteed to converge toward a solution of the _unregularized_ problem (with sparse support in general).

Authors support the feasibility and usefulness of their approach on an extensive set of experiments.

**Audience:**

Yes

**Broader Impact Concerns:**

No broader impact concerns as far as I can tell.

**Claims And Evidence:**

Yes

**Requested Changes:**

- [Critical] Having a better understanding of the benefits of the proposed setting (having one exact marginal, and one submarginal ; working with discrete measures). And if there is no such benefits, re-write the theoretical part of the paper in a more general setting.
- [Critical] A better presentation of related works to make the contribution of the present paper more salient.
- [Enhancement] Improving the clarity of the paper (see "Minor remarks and suggestion" above).

**Strengths And Weaknesses:**

# Strengths

- Fairly well motivated problem / solution.
- Theory and experiments seem sounded overall.
- Large set of experiments, and performances seem convincing as far as I can tell.


# Weaknesses

- As far as I can tell, from a theoretical perspective, the paper only plugs together known tools (Partial OT from Figalli, entropic OT a la Cuturi, Bregman projection from Xie et al). In particular the first two points of the contributions (page 2) are disputable as, unless I am missing something, there is no real novel problem/method introduced in this paper from a theoretical perspective.
- Related to the previous point, I would have appreciated more insights on the problem considered. Namely, the paper consider a subcase of Figalli's partial OT problem (transporting the whole $\mu$ and considering discrete measures with mostly uniform mass), but as far as I can tell the specificity of this formulation is never used (aside from the experimental intuition); everything developed in this paper could be adapted to POT faithfully, isn't it?
- A more detailed comparison with state-of-the-art theoretical and/or computational approach would be important. For instance, [this work](https://arxiv.org/pdf/1910.12958.pdf) by Séjourné et al., already propose a Sinkhorn scheme for Partial OT (which correspond to taking the total variation as marginal divergence) and I think that the dual problem / Sinkhorn algorithm derived is substantially equivalent to their algorithm. The work of Chapel et al. is not described either while being used as a major competitor in experiments. Right now, it is hard to assess how "fair" the comparison is.
- The choice of the parameter $c$ is not discussed while it is showcased to play an important role in terms of experimental results. Intuitively, it interpolates between standard OT (possibly with entropic regularization) and "nearest-neighbor Transport", the latter being simpler to compute than the former. As such, I think that a natural question (that would be an interesting contribution as it would depend on the specific context of the paper) would be to know whether the computational complexity decreases as $c$ increases (similarly to the entropic regularization parameter, see (Feydy et al., 2019)).
- Clarity can be improved in some places, see the "Minor remarks" below.


# Minor remarks and suggestions

- While the Remark in page 4 feels interesting, it is never leveraged later on as far as I can tell. If not, I think it could be deferred to the Appendix. If it plays an important role (e.g. as a way to somewhat bound the support of $\nu^*_c$ or so), this should be highlighted.
- [clarity] I'm a bit failing to see the point of introducing the two settings $P1 \leq c  \nu$ and $P 1 \leq \zeta$, given that we could always write the later $P 1 \leq \|\zeta\| \underbrace{\frac{\zeta}{\|\zeta\|}}_{=: \nu}$. I think that it would make things clearer to stick with one or another formulation, and only briefly mention the equivalence between these two settings.
- [clarity] The variable $\xi^{(k)}$ is only defined in the Algorithm 1; which makes equation (13) hard to parse at first glance.
- A more extensive presentation of the work of (Xie et al., 2020) would be useful given the role it plays in the current work.
- [clarity-ish] In Section 2.4, I think the presentation would be enhanced by immediately saying that we have access to a close form for $\theta$, but that relying on mini-batch gradient descent has merits (I was first really confused about the need of running a gradient descent for a simple weighted linear regression).
- [clarity] In the experiments (Figure 3 and 4), what are the 500 iterations on the $x$-axis? The caption only mentions 200 inter and outer iterations. From page 26, it seems to be a comparison point between Alg. 1 and 2. Please clarify this.
- The overhead (if any) of using Algorithm 2 instead of Algorithm 1 is not discussed as far as I can tell (even in the supplementary material).

---

> ### Author Response · Authors · 2023-10-14
> **Reply to Reviewer BRfh 1/2**
>
> We thank the reviewer for constructive feedback which we have taken into account in the revision.
>
> We have added a prior work section based largely on the responses to the reviewers. In particular, the work by  Séjourné et al. (2019) is very extensive and mentions the adjustments necessary to perform asymmetric marginal penalty case, which almost matches our discrete, semi-relaxed entropically regularized formulation.  Using the notation from our paper, we have adapted the formulations for the unbalanced optimal transport with asymmetric penalties, and added it to Appendix D. With slight differences, the Sinkhorn-like algorithm would be the same (without the acceleration) for uniform source masses.
>
> A key difference in Algorithm 1 ($\texttt{SS-Entropic}$) in our work is that it uses an accelerated proximal gradient method. Additionally, our work's Algorithm 2 ($\texttt{SS-Bregman}$) uses insights from the work by Xie et al. (2020) to achieve solutions closer to the solution to the linear program.
>
> Regarding related work of Chapel et al. (2020) in the context of positive-unlabeled learning, we will add a better description of that work in relation to ours. That work considers the constrained, fully-relaxed partial optimal transport where only a fraction $s$ of the total mass of the target.
>
> Due to the constraint set $\\{0, \frac{1}{n}\\}$ this problem is not a linear program, and only has a non-empty feasible set when $s \mod 1/n = 0$. The solution to this problem will create a uniform distribution amongst the selected subset of the source marginal, which can be renormalized to unit mass. (The solution is obtained from the solution to another optimization problem involving group Lasso regularization and additional dummy points to account for dropped mass.) However, the main difference with this relaxation as compared to our approach (or other semi-relaxed approaches) is that points in the target may lose mass or be completely dropped.
>
> Based on the observation that the new source mass is a piecewise linear function of $c$ (for the case of uniform masses), we propose to examine the specific relationships: 1) \hl{As source points exit the support with increasing $c$, they never re-enter the support. Thus, it is possible to remove these points (and the corresponding columns of the matrix).} 2) Starting from a solution at $c^*$, which can be obtained by a greedy algorithm, as $c$ decreases there are some points where the marginal constraint is active $\sum_{i} P_{ij} = c\nu_j$, once a point is active it will remain active as $c$ decreases. 3) Returning to the case where $c$ increases, there are multiple types of breakpoints: (i) when source points leave the active set but maintain mass ; (ii) when source points leave the active set and lose mass; (iii) when points with constant mass start to loss mass; (iv) when points loss all of their mass.

---

> ### Author Response · Authors · 2023-10-14
> **Reply to Reviewer BRfh 2/2**
>
> # Minor updates:
> 1. Regarding the remark (on page 4) in the introduction. We have better incorporated this remark with an understanding of the path of solutions as a function of $c$. Also this remark helps to illustrate that the extreme points of the set of possible marginals are proportional to the solutions $ \mathrm{PUW}_p^p(s)$ in Chapel et al. (2020). However, because of the equality constraint these extreme points are not always obtained.
>
> 2. The goal of having both the constraint written with bounds $c \boldsymbol{\nu}$ and $\boldsymbol{\zeta}$ was to illustrate solutions as a function of $c$, but state the algorithms in a general way that would allow for elementwise scaling of the initial mass $\boldsymbol{\zeta}=\mathbf{c} \boldsymbol{\nu}, \mathbf{c}\in[1,\infty)^n$. Thanks to another reviewer we note that Rabin, Ferradans, and Papadakis (2014) explored the elementwise scaling in the context of color transfer. We have clarified this in our notation also.
>
> 3.  We now introduce the extrapolated point in the accelerated proximal gradient method $\xi^{(k)}$ along with the update of the variable.
>
> 4.  We have introduced the work of Xie et al. (2020) discussing the IPOT approach more thoroughly in the introduction.
>
> 5. We have reorganized Section 2.4 regarding the discussion of the point selection with affine transforms.
>
> 6.  Regarding the iteration count, we agree that it was a mistake, the figure axis should refer to iterations in alternating algorithm for updating the affine parameters.    We have made appropriate changes to the figures.
>
> 7.  The work introducing IPOT (Xie et al., 2020) states that the complexity of $inexact$ proximal point method is comparable to the Sinkhorn algorithm. In contrast, an $exact$ proximal point method would be much costlier as compared to simple gradient steps involved in the Sinkhorn algorithm. We note this in the introduction.
>
> # References:
> 1. S ́ejourn ́e et al., 2019. ”Sinkhorn divergences for unbalanced optimal transport.” arXiv preprint arXiv:1910.12958.
> https://doi.org/10.48550/arXiv.1910.12958
>
> 2.  Xie et al., 2020. ”A Fast Proximal Point Method for Computing Exact Wasserstein Distance.” Proceedings
> of The 35th Uncertainty in Artificial Intelligence Conference, PMLR 115:433-453
>
> 3. Chapel, Laetitia, Mokhtar Z. Alaya, and Gilles Gasso. "Partial optimal tranport with applications on positive-unlabeled learning." Advances in Neural Information Processing Systems 33 (2020): 2903-2913.
>
> 4. RABIN, Julien, FERRADANS, Sira, et PAPADAKIS, Nicolas. Adaptive color transfer with relaxed optimal transport. IEEE international conference on image processing (ICIP), 2014.

---

### Review · Reviewer_sJdW · 2023-09-20

**Summary Of Contributions:**

The article deals with partial optimal transport, that is, when the target and source measures do not maintain the entirety of their mass over the transport. In particular, the authors focus on a specific case where the measures are discrete and uniform, to propose a methodology for a particular case where it is only for the source that only part of the mass is transported.

The paper is well presented with sufficient experimental validation

**Audience:**

Yes

**Claims And Evidence:**

Yes

**Requested Changes:**

i) Please refer to my comments above

ii) Sec 2.2: *Cuturi applied entropic regularization* please refer to works, not persons

iii) Secs 2.2, 2.3, 2.4 seem to propose a methodology and then the next section refutes them to propose something "better". In my opinion (if I am correct) this weakens the proposal of the paper. If possible, please reformulate.

iv) Possibly rename "algorithm 1 / algorithm 2" so that in the experiments they are not referred to as such. Idea: SS-entropic and SS-Bregman (or other appropriate names)

v) Sec 3.2 reads *It can be observed that the results for larger values of c are smoother and sharper as compared to the optimal transport case c = 1.* Can something be **smoother and sharper** at the same time?

**Strengths And Weaknesses:**

The paper is very well written; it is extensive, but that is because the authors give a great amount of detail, which I consider to be positive in this type of article, blending concepts with applications.

The conceptual part is very clearly presented, the authors re-derive the Sinkhron approach for the particular case of partial OT, and the procedure is presented in a detailed manner to the reader.

The experimental validation is superb. The treatment and explanation of the experimental aspects are very deep and informative.

Some points can be improved in the paper, but they are mainly related to the presentation and not methodology (so listed in the next section of the review) **except the following ones**:

- how can the scaling factor be chosen? The article provides multiple instances to compare how the methodology performs for different choices of $c$. However, in real applications, $c$ is not known and needs to be chosen. By arbitrarily choosing this factor, the method will leave parts of the support unmatched to the target (as shown in the first experiment), so a naive choice can lead to misleading conclusions. Can $c$ be chosen in a way that is more principled than just the ratio between the points one wants to match and all the points? (because one could not know how many points should be matched)

- Also, why is only the uniform case considered? In the toy example, the application of the method to this kind of distribution appears to result in removing the parts of the support that are far away from the target, to then rescale the source and apply complete OT

---

> ### Author Response · Authors · 2023-10-05
> **Reply to Reviewer sJdW**
>
> We thank the reviewer for providing constructive feedback that we have incorporated in revision.
>
> 1.  Regarding selecting the optimal value of the constraint parameter $c$, we adapt the knee-based method from the recent work by Phatak et al. (2023), which examines the derivative of cost of the subset solution as a function of $\frac{1}{c}$ to find a value that well separates inliers and outliers. The work by Phatak et al. (2023) shows that the OT-profile is a piece-wise linear convex function, and can be computed efficiently, and changes in the slope of the smoothed derivative can be find a mass fraction where the partial transport separates inliers and outliers.  We hope to investigate if the former properties hold for our cost.  Note that the quantity we compute is modified from the cost of the partial optimal transport as a function of mass transported (OT-profile) as in the work by Phatak et al. (2023), since partial optimal transport uses fixed constraints, whereas our proposed support subset selection varies the constraints along with the mass. A description of the relationship is added in the new prior work subsection 2.2 and results of the knee-based method are in PU learning figures for MNIST/EMNIST and CIFAR10.
>
> The uniform case is adopted in many examples (besides color transform), because of the nature of sampling the points in the source and target independently from two distributions. In setting such as color transfer, where clustering is first applied yielding unequal masses from initially uniformly weighted points, the presented methodology is still applicable for general masses unless specifically noted.
>
>  2.  Regarding in-text references, we interpret the suggestion to mean to use prepositions such as ``Work by Author (Year) applied'' instead of ``Author (Year) applied''. We have tried to apply this throughout.
>
> 3.  We have restructured the wording regarding Section 2.2 and 2.3 and their relation.  Section 2.2 concerns the entropic regularization as a scalable alternative to solving the linear program for support subset selection. Section 2.3 defines an algorithm using the inexact proximal point method that relies on function calls to the algorithm proposed in Section 2.2 in its inner loop to yield solutions closer to the original linear program.
>
> 4. We have changed the two algorithm names to be $\texttt{SS-Entropic}$ and $\texttt{SS-Bregman}$.
>
> 5.  We agree that the word choices of smoother and sharper seem contradictory. Our observation was that the color transitions within an object or area of similar color are ${smoother}$ (the colors in the dark backdrop behind the peppers) and the color transitions between different objects are ${sharper}$, i.e., more distinct, (the colors in the orange versus red pepper on the right side of the photograph).
>
> # Reference:
> Phatak, et al. 2023. "Computing all Optimal Partial Transports." in The Eleventh International Conference on Learning Representations.

---

### Author Response · Authors · 2023-10-14
**Reply to All Reviewers.**

We are thankful to all the reviewers for their suggestions to make our manuscript better. Answers to the questions of each reviewer are provided individually. We have revised the paper noted deletions with strikedthrough text and additions with blue text. Other than minor changes suggested by reviewers to the text along with naming convention for algorithms, we made the following changes to the paper:
*  We have updated the literature review to properly place our work in context of more recent contributions in computational optimal transport and its applications.

      *  We have a section after the introduction of the methodology to briefly survey the work done on semi-relaxed optimal transport and partial optimal transport and discuss similarities and differences between our approach and previously done work.
      *  We compare the solution path for the linear inequality-based optimization in our approach to the solution paths for divergence-based penalties discussed in literature for the semi-relaxed formulation.  We note that the inclusion of the divergence-based penalties change the underlying solution.  Results are included in both the main body and Appendix of the revision.

We will upload supplementary material along with final version of the paper.

---

### Decision · Action_Editor_VGtq · 2023-11-13

**Recommendation:** Accept with minor revision

**Comment:**

All reviewers agreed that while this work might not be totally original, it provides a new formulation for partial OT and interesting experiments, they all lean toward accept.

Some minor comments from the final discussion with reviewers that should be taken into account in the camera ready:

- p2 "Recent works have proposed algorithms for computing the entire scaling or regularization path of solution [...] semi-relaxed unbalanced optimal transport (Chapel et al., 2021)": the work of (Chapel et al., 2021) also computes the regularization path of the UOT problem, not only the semi-relaxed version.
- in Fig. 2, I would recommend scaling the point size wrt the amount of mass points to better illustrate the creation of mass.
- Fig 3 and 4 : the points are not visible (for all c>1 and theta opdate) in the pdf reader from google chrome (they are visible in local PDF reader) please solve this problem for final version.
- It is claimed "However, the main difference with this relaxation [(Chapel et al 2020)] as compared to our approach (or other semi-relaxed approaches) is that points in the target may lose mass or be completely dropped". It is still unclear why the proposed method and the one of Chapel et al, 2020 should give different accuracy results in a PU learning context as "accuracy is evaluated by assigning label 1 to the largest mass assignments to and label 0 to the remaining mass assignment". Please discuss and give more details about  this.
- Fig. 9 (a): It is not clear why the-sampling is not the same.

**Audience:**

This is of interest to the TMLR audience especially the OT for ML community.

**Claims And Evidence:**

The claims in the submission are accurate, the formulation have been validated by the reviewers. The novelty is a bit limited since many equivalent formulations already exist but updated positioning wrt state of the art better inform the readers about existing work. The approach is illustrated on several numerical experiments.

---

> ### Author Response · Authors · 2023-12-06
> **Thanks to Action Editor and Reviewers.**
>
> We are thankful to the action editor and anonymous reviewers for their thoughtful reviews and comments, which were very helpful for us to improve the quality of paper. We have incorporated all the requested changes in the camera ready version of the paper. Link for the code has also been provided in the paper.
> * We have fixed the issues with the figures and suggested mass scaling. We have also clarified the references to fully and semi-relaxed prior work on page 2 and in the related work sections.
> * Our claim in that context was regarding differences between fully-relaxed and semi-relaxed partial optimal transport. We have added a toy example to illustrate how in the context of PU learning the fully-relaxed solution can drop target (positive) points resulting in lower accuracy. We have also added the fully-relaxed partial optimal transport to the table of results for PU learning.
> The difference is subtle between our solutions and those for the PU Wasserstein (PUW) problem, proposed in the work by Chapel, et al. (2020), as both are semi-relaxed and maintain equality constraints on the target distribution. The small, but consistent, differences in the accuracy between our method and PUW may be due to the additional constraints in PUW on the new masses being uniform (achieved through group LASSO), which may yield a solution that has higher transport cost than the solutions for our subset selection which are not uniform. The largest mass assignments in the solution to our more relaxed problem may be more reliable than constrained solutions to PU Wasserstein. We have added a paragraph that clarifies this claim in the PU learning section.